# FEDERATED LEARNING WITH PROFILE MAPPING UNDER DISTRIBUTION SHIFTS AND DRIFTS

**Mohan Li**[*], **Dario Fenoglio**[*], **Martin Gjoreski & Marc Langheinrich**
Università della Svizzera italiana, Switzerland

## ABSTRACT

Federated Learning (FL) enables decentralized model training across clients without sharing raw data, but its performance degrades under real-world data heterogeneity. Existing methods often fail to address distribution shift across clients and distribution drift over time, or they rely on unrealistic assumptions such as known number of client clusters and data heterogeneity types, which limits their generalizability. We introduce FEROMA, a novel FL framework that explicitly handles both distribution shift and drift without relying on client or cluster identity. FEROMA builds on client distribution profiles—compact, privacy-preserving representations of local data—that guide model aggregation and test-time model assignment through adaptive similarity-based weighting. This design allows FEROMA to dynamically select aggregation strategies during training, ranging from clustered to personalized, and deploy suitable models to unseen, and unlabeled test clients without retraining, online adaptation, or prior knowledge on clients' data. Extensive experiments show that compared to 10 state-of-the-art methods, FEROMA improves performance and stability under dynamic data heterogeneity conditions—an average accuracy gain of up to 12 percentage points over the best baselines across 6 benchmarks—while maintaining computational and communication overhead comparable to FedAvg. These results highlight that distribution-profile-based aggregation offers a practical path toward robust FL under both data distribution shifts and drifts. Code is available at https://github.com/dariofenoglio98/FEROMA.

## 1 INTRODUCTION

Federated Learning (FL) (McMahan et al., 2017) has become a promising paradigm for training models collaboratively across distributed clients without sharing their private data. However, one of the central challenges in FL is the presence of heterogeneous client data, which can significantly degrade model performance if not properly handled (Kairouz et al., 2021). In real-world deployments, clients rarely hold independent and identically distributed (IID) data (Zhao et al., 2018; Zhu et al., 2021). They often exhibit two forms of heterogeneity: *distribution shift*, where different clients possess distinct data distributions (Sattler et al., 2021; Deng et al., 2020b; Guo et al., 2024), and *distribution drift*, where a single client's data distribution also evolves over time (Jothimurugesan et al., 2023; Lu et al., 2019b; Gama et al., 2014). These dynamics challenge the notion of a single global model that performs uniformly well across all clients throughout training and deployment.

Vanilla FedAvg (McMahan et al., 2017) struggles to converge efficiently due to its uniform aggregation of all client updates, regardless of their data distributions. Under distribution shift (Figure 1 Left), clients whose data distributions are underrepresented in the global model receive limited benefit, resulting in persistently low accuracy and slow convergence. This issue is exacerbated under distribution drift (Figure 1 Right), where local data distributions evolve over time (In Figure 1, solid lines denote average accuracy across clients, while transparent lines correspond to individual client trajectories). Such drift may occur during training—causing instability and divergence in the global model—or during test time, degrading performance as clients encounter data that no longer aligns with the distribution observed during training.

Existing methods for handling heterogeneous data in FL typically fall into three categories: Clustered FL (CFL) (Sattler et al., 2021; Guo et al., 2024; Jothimurugesan et al., 2023; Ghosh et al., 2020;

---
[*]Equal contribution.

Marfoq et al., 2021; Long et al., 2023), Personalized FL (PFL) (Deng et al., 2020b; T. Dinh et al., 2020; Tan et al., 2023a; Kulkarni et al., 2020), and Test-time Adaptive FL (TTA-FL) (Bao et al., 2023; Deng et al., 2020a; Wang et al., 2019). CFL methods group clients with similar data distributions based on model parameters or training metrics, which can effectively address distribution shift. However, they often lack adaptability under training and/or test-time drifts, require prior knowledge on the number of clusters or assumptions about data distribution(Guo et al., 2024; Ghosh et al., 2020), and incur significant training overhead due to the use of computationally intensive clustering techniques, as well as the transmission, evaluation, or training of multiple models per client (Sattler et al., 2021; Jothimurugesan et al., 2023). PFL methods optimize a personalized model for each client using its local data distribution, thereby mitigating the negative effects of inter- and intra-client distribution dissimilarity. While this yields strong performance locally—particularly when ample training data is available per client—the resulting models often become highly client-specific, sacrificing robustness in favour of personalization and offering limited generalization to unseen distributions or new clients. TTA-FL approaches are designed to handle test-time drifts, but usually rely on online adaptation or additional client interaction, which limits their practicality and efficiency in deployment. In addition, they often overlook training-time drift and shift, which can lead to unstable or slowed convergence and and degraded model performance across clients. Despite their strengths, these existing approaches are often tailored to specific non-IID types (Sattler et al., 2021; Deng et al., 2020b; Guo et al., 2024; Jothimurugesan et al., 2023; Ghosh et al., 2020; Marfoq et al., 2021; Long et al., 2023; T. Dinh et al., 2020) and may struggle to balance generalization, adaptability, and efficiency—factors that are increasingly important for practical FL deployments under real-world heterogeneous conditions.

To bridge this gap, we introduce **Fe**derated Learning with Distribution P**ro**file **Ma**pping (**FEROMA**), an FL framework that moves the focus from client or cluster identity to the underlying data *distribution profile*. To the best of our knowledge, FEROMA is the first general-purpose FL framework explicitly designed to address both distribution shift and distribution drift, during training as well as test time. FEROMA extracts a lightweight, differentially private statistical profile from each client's local data and maps it to previously observed profiles from last training round. This mapping guides model aggregation through similarity-based weighting.

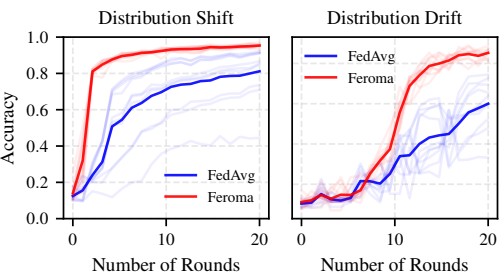

Figure 1: Comparison between FedAvg and FEROMA under (Left) distribution shift across clients, and (Right) under distribution drift every 2 rounds.

Based on these profiles, FEROMA automatically selects the most suitable aggregation strategy for each round through adaptive similarity thresholds. This design enables FEROMA to remain both flexible and scalable under dynamic, real-world FL settings, without relying on any prior knowledge. Moreover, FEROMA assigns trained models to unseen clients based on profile similarity, enabling robust model selection during test-time. We summarize the key contributions of FEROMA as follows:

- **A unified and adaptive aggregation framework.** FEROMA dynamically selects the best aggregation strategy—ranging from clustered to personalized or global—based on client distribution profiles, and naturally extends to test-time adaptation without retraining.
- **Effective under both distribution shift and drift.** FEROMA handles static and dynamic data heterogeneity by leveraging round-wise distribution mapping, and scales efficiently to a large number of clients. We evaluate FEROMA on four standard and two real-world datasets, showing consistent gains over 10 SOTA baselines across a wide range of scenarios: including four types of distribution shift (with low, medium, and high severity) and varying drift frequencies.
- **Lightweight and efficient design.** FEROMA introduces minimal communication and computation overhead, with both server- and client-side costs comparable to FedAvg, enabling practical deployment in resource-constrained environments. We provide both theoretical bounds and empirical measurements to validate its practicality in resource-constrained federated settings.

## 2 BACKGROUND

**FL under IID assumption.** FL systems (McMahan et al., 2017) consist of a collection of $K \in \mathbb{N}$ distributed clients, denoted as $\mathcal{K} = \{1, 2, \ldots, K\}$, coordinated by a central server. These clients

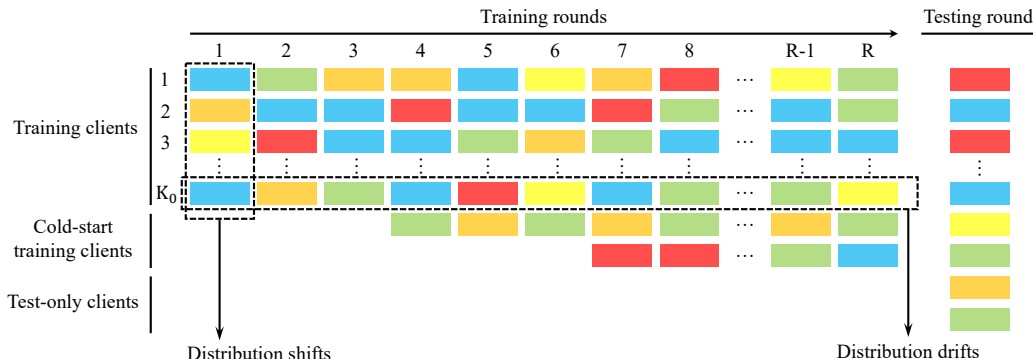

Figure 2: **Distribution shifts and drifts in FL.** Colors indicate distinct local data distributions. Changes across clients reflect *distribution shifts*; changes over rounds reflect *distribution drifts*.

collaboratively train a shared machine learning model while keeping their local data private. Under IID assumptions, FL typically assumes that each client $k \in \mathcal{K}$ holds an IID dataset sampled from a common joint distribution $P(X, Y)$. Throughout, we use $P$ to denote joint distributions over $(X, Y)$. Specifically, client $k$ draws a finite dataset $D^{(k)} = \{(x_i^{(k)}, y_i^{(k)})\}_{i=1}^{s^{(k)}}$ of size $s^{(k)} \in \mathbb{N}$; once sampled, $D^{(k)}$ is treated as fixed throughout training. We write $x^{(k)} \in \mathbb{R}^{s^{(k)} \times z}$ and $y^{(k)} \in \{0, 1\}^{s^{(k)} \times u}$ for the matrices collecting the inputs and labels in $D^{(k)}$, where $z$ is the number of features per sample and $u$ the number of output classes. During each communication round $t$, every client $k$ performs local training by optimizing its model parameters $\theta_t^{(k)}$ to maximize the likelihood over its private dataset $D^{(k)}$, as shown in Equation 1. Once local updates are computed, clients transmit their parameters to the central server. The server then aggregates the received updates—typically via simple weighted averaging (McMahan et al., 2017)—to produce a new global model $\theta_{t+1}$, as defined in Equation 2. This updated global model is broadcast back to the clients to initialize the next training round.

$$\text{(Local training)} \quad \theta_t^{(k)} = \arg\max_{\theta} \mathcal{L}(\theta \mid x^{(k)}, y^{(k)}) \tag{1}$$

$$\text{(Aggregation)} \quad \theta_{t+1} = \sum_{k=1}^{K} p_k \cdot \theta_t^{(k)} \quad \text{where} \quad p_k = \frac{s^{(k)}}{\sum_{j=1}^{K} s^{(j)}}, \ \sum_{k=1}^{K} p_k = 1. \tag{2}$$

**FL under distribution shifts.** In practical FL scenarios, clients commonly face *inter-client distribution shift*, where data distributions vary across clients—referred to here as distribution shift. As illustrated in Figure 2 (vertical variation within each round), for any two clients $k_1, k_2 \in \mathcal{K}$, it may hold that $P^{(k_1)} \neq P^{(k_2)}$, violating the standard IID assumption $P^{(k)} = P$ for all $k$. Such shifts may result from variations in user behavior, environment, or adversarial activity, and are typically grouped into four types (Kairouz et al., 2021):

- *Feature distribution skew (covariate shift)*: Marginal distributions $P(X)$ vary across clients.
- *Label distribution skew (prior probability shift)*: Marginal distributions $P(Y)$ vary across clients.
- *Concept shift (same $X$, different $Y$)*: Conditional distributions $P(Y \mid X)$ vary across clients.
- *Concept shift (same $Y$, different $X$)*: Conditional distributions $P(X \mid Y)$ vary across clients.

While distribution shift is well-studied in centralized learning (Lu et al., 2019b; Koh et al., 2021; Li et al., 2022c; Tahmasbi et al., 2021b), it remains underexplored in FL—where most works target specific shift types and demand extra communication/computation costs (Sattler et al., 2021; Deng et al., 2020b; Guo et al., 2024; Ghosh et al., 2020; Marfoq et al., 2021; Long et al., 2023; T. Dinh et al., 2020; Tan et al., 2023a; Kulkarni et al., 2020), due to the server's limited visibility into decentralized client data. This challenge is further compounded by resource constraints on client devices, which limit both the training capacity and the complexity of deployable models. As a result, global models often fail to generalize across non-identically distributed client populations.

**FL under distribution drifts.** In addition to distribution shift, FL systems may also encounter *intra-client distribution drift*, which we refer to as distribution drift. As illustrated in Figure 2 (horizontal variation across rounds), this occurs when the local data distribution of a single client changes over

time. Formally, for any client $k$ that participates at both rounds $t_1$ and $t_2$ (i.e., $k \in \mathcal{K}_{t_1} \cap \mathcal{K}_{t_2}$), the local distribution may drift over time: $P_{t_1}^{(k)} \neq P_{t_2}^{(k)}$. Drift may manifest during training—due to changes in user behavior, sensor conditions, or data collection environments—or during testing. In the latter case, drift arises either because the same client exhibits evolving behavior at inference time or due to the presence of entirely new clients (e.g., *test-only* clients in Figure 2) whose data distributions were not observed during training and for which label information is unavailable. Such temporal drift significantly increases training and testing complexity and necessitates adaptive strategies that can respond to evolving data. While distribution drift has been extensively studied in centralized settings (Lu et al., 2019a; Gao et al., 2022; Li et al., 2022b; Tahmasbi et al., 2021a), only a few studies have investigated training-time drift (Jothimurugesan et al., 2023; Chen et al., 2024b) or test-time drift (Bao et al., 2023; Tan et al., 2023c) in FL. However, to the best of our knowledge, the joint presence of training and test-time drift has never been explored in existing FL literature—despite its critical impact on model performance and accuracy in real-world deployments.

## 3 FEROMA

To tackle the challenges posed by dynamic and heterogeneous federated environments, we propose FEROMA, a lightweight framework that adapts to both distribution shift and drift. This section begins by formalizing the problem setting (section 3), then outlines the two core components of the FEROMA pipeline, illustrated in Figure 3: *distribution profile extraction* (subsection 3.1) and *distribution profile mapping* (subsection 3.2). In addition, section 3.2.1 details how FEROMA dynamically selects the optimal aggregation strategy for each client based on the inferred distribution structure. The full implementation and algorithmic details are provided in Appendix B.

**Problem definition.** We consider a dynamic FL system with $K_t$ clients at round $t$, denoted as $\mathcal{K}_t = \{1, \dots, K_t\}$, where each client holds local data $D_t^{(k)} = (x_t^{(k)}, y_t^{(k)})$ and trains a local model $\theta_t^{(k)}$. Each $D_t^{(k)}$ is drawn from an underlying local distribution $P_t^{(k)}$ over $(X, Y)$ and treated as fixed at round $t$ once collected. The number of clients may vary across rounds due to client arrivals (*cold-start*) or departures, i.e., $K_{t-1} \neq K_t$. In practical FL settings, data distributions are both *non-identical across clients* and *non-stationary over time*. Formally, for any $t_1, t_2$ and any $k_1 \in \mathcal{K}_{t_1}$, $k_2 \in \mathcal{K}_{t_2}$, the local distributions may differ: $P_{t_1}^{(k_1)} \neq P_{t_2}^{(k_2)}$. This unified formulation captures both *inter-client shift* (when $k_1 \neq k_2$) and *intra-client drift* (when $k_1 = k_2$). Our goal is to design a lightweight FL framework that explicitly addresses both distribution shift and drift during training and testing, without relying on client identities or prior knowledge of underlying distributions.

### 3.1 DISTRIBUTION PROFILE EXTRACTION

When a client's local distribution shifts between consecutive rounds—e.g., from $P_t^{(k)}$ to $P_{t+1}^{(k)}$—adapting a model trained on its own earlier distribution may require significant local computation and communication. A more efficient alternative is to reassign a model (or an aggregation of models) previously trained on distributions that closely resemble the client's current distribution. To enable such model selection under distribution drift and to guide aggregation across heterogeneous clients under distribution shift, we require a distribution profile, i.e., a low-dimensional, stable summary that quantifies distribution similarity without exposing raw data or labels. Our use of compact distribution summaries builds on the descriptor-based perspective introduced in (Fenoglio et al., 2025), while targeting drift-aware temporal matching across rounds.

**Definition 3.1** (Distribution–Profile Extractor). *Let $z \in \mathbb{N}$ denote the feature dimension, $u \in \mathbb{N}$ the label dimension, and let $d$ be the desired profile dimension. A Distribution–Profile Extractor (DPE) is a stochastic mapping $\phi_\psi : \mathbb{R}^{v_t^{(k)} \times (z+u)} \to \mathbb{R}^d$, with $v_t^{(k)} \leq s_t^{(k)}$, parametrized by $\psi$, that maps the dataset $D_t^{(k)}$ to a distribution profile $d_t^{(k)} := \phi_\psi(D_t^{(k)})$. Intuitively, the DPE compresses a client's dataset into a compact distributional signature that captures how the data is distributed, rather than the data itself, enabling reliable similarity estimation without exposing fine-grained information. For any two client–round pairs $(k_1, t_1)$ and $(k_2, t_2)$, the extractor must satisfy the requirements below.*

(R1) **Distribution fidelity.** Profile distances should approximate a reference distance $\Delta$ (e.g., Jensen–Shannon or Wasserstein) between the corresponding underlying distributions. Concretely,

letting the expectation be taken over the internal randomness of $\phi_\psi$,

$$\mathbb{E}\Big[\big|\,\|d_{t_1}^{(k_1)} - d_{t_2}^{(k_2)}\|_2 - \Delta\big(P_{t_1}^{(k_1)}, P_{t_2}^{(k_2)}\big)\,\big|\Big] \leq \xi.$$

In other words, the extractor $\phi_\psi$ should map similar distributions to nearby profiles and dissimilar distributions to distant profiles.

(R2) **Label agnosticism.** The extractor $\phi_\psi$ must support profile generation *without* labels:

$$d_t'^{(k)} := \phi_\psi\big(x_t^{(k)}, \mathbf{0}\big) \in \mathbb{R}^p, \quad \text{with } p \leq d,$$

where $d_t'^{(k)}$ is a subvector of the full profile $d_t^{(k)} \in \mathbb{R}^d$. This enables profile extraction and similarity matching at test time. Here, $d_t'^{(k)}$ captures marginal distributional characteristics based solely on the input features $x_t^{(k)}$.

(R3) **Controlled stochasticity.** The extractor $\phi_\psi$ is a stochastic mapping: for a fixed input dataset, the profile $d_t^{(k)} = \phi_\psi(D_t^{(k)})$ is a random vector. Let the expectation be taken over the internal randomness of $\phi_\psi$, we have $\mathbb{E}[d_t^{(k)} \mid D_t^{(k)}] = \bar\phi_\psi(D_t^{(k)})$, with bounded conditional covariance $\text{Cov}\big(d_t^{(k)} \mid D_t^{(k)}\big) \preceq \rho^2 \mathbf{I}_d$. This controlled stochasticity prevents exact fingerprinting of client distributions across rounds while maintaining reliable inter-profile distances in expectation.

(R4) **Differential-privacy guarantee.** The mapping $\phi_\psi$ satisfies $(\varepsilon, \delta)$-differential privacy at the *sample* level: for any two datasets $D_t^{(k)}$ and $D_t'^{(k)}$ that differ in exactly one example and for every measurable set $\mathcal{S} \subseteq \mathbb{R}^d$,

$$\Pr\big[\phi_\psi(D_t^{(k)}) \in \mathcal{S}\big] \;\leq\; e^\varepsilon \,\Pr\big[\phi_\psi(D_t'^{(k)}) \in \mathcal{S}\big] + \delta.$$

For example, this guarantee can be realized with a Gaussian (or Laplace) mechanism: $d_t^{(k)} = \bar\phi_\psi(D_t^{(k)}) + \mathcal{N}\big(0, \sigma^2 \mathbf{I}_d\big)$, where $\sigma$ is calibrated to the $\ell_2$-sensitivity of $\bar\phi_\psi$. The resulting many-to-one mapping both limits information leakage and obfuscates client identity.

(R5) **Compactness.** Profile extraction introduces minimal overhead compared to vanilla FL: its computational cost is negligible relative to a local training epoch, and the profile dimension $d$ satisfies $d \ll |\theta|$ (typically $d/|\theta| \leq 10^{-2}$), ensuring that the additional communication cost remains marginal compared to the transmitted model update.

We implemented our DPE using a four-step statistical moment extraction from latent space with differential privacy. We provide details of the implemented DPE in our FEROMA in Appendix B.4, which satisfies all five requirements: (R1) with a mapping $\phi_\psi$ provably bi-Lipschitz equivalent to the 2-Wasserstein metric, showing $\xi < 1.1$ on MNIST (and $< 0.54$ under Jensen–Shannon); (R2) by consistently providing a label-free subvector $d_t'^{(k)}$ that approximates the marginal data distribution; (R3) with bounded covariance $\rho^2 = \big(\frac{\tau^2}{M\gamma v^{(k)}} + 2b_{\max}^2\big) \leq 2.2 \times 10^{-3}$ depending on a Monte-Carlo subsampling $(M, \gamma)$ plus Laplace noise $(b_{\max})$; (R4) by ensuring $(\varepsilon, 0)$-DP for each profile vector $d_t^{(k)}$ with added variance $\leq 2.2 \times 10^{-5}$; and (R5) by introducing negligible computation and a communication cost of $d/|\theta| \leq 3.5 \times 10^{-3}$. Full implementation details and theoretical justifications are in Appendix B.4, with privacy calibration in Appendix F.

## 3.2 DISTRIBUTION PROFILE MAPPING

The core idea of FEROMA is to decouple model identity from specific clients or clusters, and instead associate each model with a data distribution characterized by its distribution profile $d_t^{(k)}$. Once profiles are extracted, we employ two complementary mapping strategies: during *training*, we enable model sharing by matching current and past profiles to derive weighted aggregations across clients; during *testing*, we extract a label-free profile for each unseen client and assign the closest model from the final round for direct inference.

### 3.2.1 TRAINING DISTRIBUTION MAPPING

After extracting all distribution profiles for the current round $t$, i.e., $\{d_t^{(k)}\}_{k=1}^{K_t}$, we map them to the last round profiles $\{d_{t-1}^{(k)}\}_{k=1}^{K_{t-1}}$ to define the weights for model aggregation and assignment. The

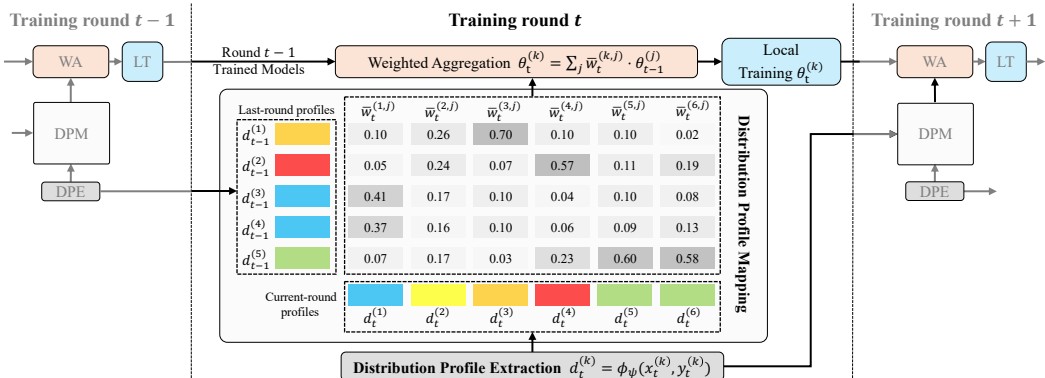

Figure 3: **FEROMA pipeline.** In each round $t$, clients extract distribution profiles (DPE), map them to previous-round profiles (DPM), and compute weighted aggregation (WA) for local training (LT).

mapping can be done with a normalized distance function:

$$w_t^{(k,j)} = \frac{\exp\left(-\mathcal{D}(d_t^{(k)}, d_{t-1}^{(j)})\right)}{\sum\limits_{j' \in \mathcal{A}_{t-1}} \exp\left(-\mathcal{D}(d_t^{(k)}, d_{t-1}^{(j')})\right)} \tag{3}$$

where $\mathcal{D}(\cdot, \cdot)$ is a chosen distance function (e.g., Euclidean distance), $w_t^{(k,j)}$ the association weight between current client $k$ and previous-round client $j$, and $\mathcal{A}_{t-1}$ the set of clients active in round $t-1$.

Although weakly similar profiles receive small weights, their aggregation can still introduce noise. To regulate when collaboration is beneficial, a threshold $\tau$ can be applied to discard any $w_t^{(k,j)}$ below $\tau$, ensuring that model updates are shared only among sufficiently similar profiles and preventing both over-collaboration and unnecessary fragmentation, in a manner similar to clustered FL approaches:

$$\tilde{w}_t^{(k,j)} = \begin{cases} w_t^{(k,j)} & \text{if } w_t^{(k,j)} \geq \tau \\ 0 & \text{otherwise} \end{cases} \tag{4}$$

After thresholding, the weights $\tilde{w}_t^{(k,j)}$ are renormalized to ensure they sum to 1 across $j \in \mathcal{A}_{t-1}$. This weight can be combined with weighting based on data size as in FedAvg. More details on the distance function $\mathcal{D}(\cdot, \cdot)$ and ablation studies on the utility of $\tau$ are in Appendices I.9 and I.10.

**Automatic aggregation strategy selection.** At each training round $t$, we compute the association weights $\{w_t^{(k,j)}\}_{k \in \mathcal{K}_{\sqcup}, \, j \in \mathcal{A}_{t-1}}$ via equation 3 or the thresholded $\tilde{w}_t^{(k,j)}$ version via equation 4, depending on the desired level of selectivity, and then aggregate as

$$\theta_t^{(k)} = \sum_{j \in \mathcal{A}_{t-1}} \bar{w}_t^{(k,j)} \cdot \theta_{t-1}^{(j)} \tag{5}$$

where $\bar{w}_t^{(k,j)}$ denotes either $w_t^{(k,j)}$ or $\tilde{w}_t^{(k,j)}$. After aggregation, each client proceeds with its local training step as in Equation 1. By inspecting the support of $\{\bar{w}_t^{(k,j)}\}_{j \in \mathcal{A}_{t-1}}$ for each client $k$ at round $t$, FEROMA independently and dynamically recovers the most suitable FL aggregation strategy:

- *Clustered FL.* ($d_t^{(1)}$ *in Figure 3*) If multiple weights $\bar{w}_t^{(k,j)} > 0$ survive thresholding, the aggregation for client $k$ aggregates those models trained on similar data distribution, akin to CFL.
- *Personalized FL.* ($d_t^{(3)}, d_t^{(4)}, d_t^{(5)}, d_t^{(6)}$ *in Figure 3*) If exactly one thresholded weight is nonzero, client $k$ simply inherits that single most similar model, yielding a personalized specialization.
- *Global FL fallback.* ($d_t^{(2)}$ *in Figure 3*) If no sufficiently similar profile is identified, it falls back to global aggregation ($\bar{w}_t^{(k,j)} \approx 1/|\mathcal{A}_{t-1}|$ if not thresholded), thus combining all models before the next adaptive round.

### 3.2.2 TESTING DISTRIBUTION MAPPING

At test time, we aim to assign each unseen client $k$ to the best-matching model learned in the final training round $R$ based on its feature distribution, without any further optimization. First, we extract the label-free profile $d_{\text{test}}'^{(k)} = \phi_\psi(x_{\text{test}}^{(k)}, \mathbf{0})$, as required by (R2). Then we match $d_{\text{test}}'^{(k)}$ against the set of round-$R$ profiles $\{d_R^{(j)}\}_{j \in \mathcal{A}_R}$ by selecting the nearest neighbor in profile space:

$$ j^* \;=\; \arg \min_{j \in \mathcal{A}_R} \mathcal{D}\big(d_{\text{test}}'^{(k)}, d_R'^{(j)}\big), \qquad \theta_{\text{test}}^{(k)} \;=\; \theta_R^{(j^*)}. $$

This one-shot assignment requires no gradient steps, leverages the DP-protected distribution profiles, and naturally generalizes to unseen, unlabeled clients. As discussed in Appendix E, pure label-free matching cannot inherently capture concept shift with identical $X$ but different $Y$. However, for addressing this problem, we show that a small, test-time labeled validation set can seamlessly refine the associations and substantially improve performance. In addition, by assigning the most appropriate pre-trained model to each test client, FEROMA enables the integration of unsupervised test-time adaptation methods by offering a distribution-aware initialization point (see subsection 5.2).

## 4 EXPERIMENTS

This section presents our experimental setup and results from two primary scaling studies. We evaluate FEROMA under varying drift frequencies, non-IID types, severity levels, and numbers of clients. These experiments assess the scalability, robustness, and efficiency of FEROMA compared to 10 SOTA baselines across a wide range of real-world heterogeneity scenarios.

### 4.1 EXPERIMENT SETTINGS

**Drifting datasets generation.** We employ six publicly available datasets for our experiments: MNIST (LeCun & Cortes, 2005), Fashion-MNIST (FMNIST) (Xiao et al., 2017), CIFAR-10, CIFAR-100 (Krizhevsky), and two real-world datasets, CheXpert (Irvin et al., 2019) and Office-Home (Venkateswara et al., 2017). To construct distribution shift and drift datasets under different non-IID conditions, we use *ANDA*[1], a toolkit supporting operations such as class isolation and label swapping. For the real-world datasets, we preserve their intrinsic characteristics without modification. Detailed dataset information is provided in Appendix I.1.

**Baseline algorithms.** We evaluate our approach against baseline methods summarized in Table 3, including: FedAvg (McMahan et al., 2017), FedRC (Guo et al., 2024), FedEM (Marfoq et al., 2021), FeSEM (Long et al., 2023), CFL (Sattler et al., 2021), IFCA (Ghosh et al., 2020), FedDrift (Jothimurugesan et al., 2023), pFedMe (T. Dinh et al., 2020), APFL (Deng et al., 2020b), and ATP (Bao et al., 2023). The baseline methods are detailed further in Appendix A. The experimental environment, models, and hyperparameter configurations are described in Appendices B.2 and B.3.

### 4.2 RESULTS

**Scaling the drifting frequency, non-IID types, and non-IID levels.** We first evaluate the robustness of FEROMA by comparing it against baseline methods across three drift frequencies: each client's dataset drifts every four rounds, every two rounds, and at every round. Additionally, we simulate four types of distribution shifts—$P(X)$, $P(Y)$, $P(Y|X)$, and $P(X|Y)$—each under three levels of non-IID severity: low, medium, and high. This setup enables us to thoroughly assess the adaptability of FEROMA in highly dynamic and heterogeneous FL environments. Detailed experimental setups and results are provided in Appendices I.2 and I.4, and summarized in Table 1. Ablation study on training rounds is summarized in Appendices J and K.

Table 1 shows that FEROMA consistently outperforms all baselines across six benchmark datasets, with results computed as unweighted averages over all drift frequencies, non-IID types, and severity levels, without case selection. Notably, FEROMA improves accuracy by up to 14.1, 14.3, and 10.3 percentage points (pp) over the best-performing baseline CFL on MNIST, FMNIST, and CIFAR-10, respectively. On CIFAR-100, it achieves an accuracy of 39.9%, surpassing FedEM by 8.2 pp. On

---

[1] https://github.com/alfredoLimo/ANDA

the real-world datasets CheXpert and Office-Home, FEROMA demonstrates consistent robustness, exceeding the strongest baselines while maintaining lower variance. Table 2 further shows that FEROMA remains robust under varying scales of distribution shift and drift (see Appendix I.2 for detailed results of additional benchmarks). These improvements are achieved under realistic FL conditions with no prior knowledge of distribution modes or test-time labels. While baselines are often specialized for either shift or drift, FEROMA adapts to both with significantly lower variance. The results underscore the effectiveness of distribution-profile-based aggregation and highlight FEROMA as a generalizable solution for dynamic non-IID FL conditions.

**Scaling the number of clients.** In real-world FL deployments, the number of participating clients can be large, requiring FL methods to remain scalable with minimal computational and communication overhead—even under highly dynamic conditions. To evaluate scalability, we assess the performance of FEROMA and baseline methods with MNIST as the number of clients increases from 10 to 20, 50, and 100. Notably, FedDrift could not be evaluated with 50 or 100 clients due to excessive computational requirements (see Appendix B.2 for details). A summary of the results is presented in Figure 4, with detailed experimental setup and results provided in Appendix I.3.

| Dataset | MNIST | FMNIST | CIFAR-10 | CIFAR-100 | CheXpert | Office-Home |
|---|---|---|---|---|---|---|
| FedAvg | 71.8 ± 5.5 | 63.7 ± 6.4 | 33.0 ± 5.3 | 28.2 ± 4.7 | 59.1 ± 3.0 | 41.0 ± 1.3 |
| FedRC | 30.9 ± 6.9 | 45.1 ± 6.9 | 23.2 ± 4.5 | 30.6 ± 4.2 | 55.1 ± 1.9 | 14.6 ± 3.5 |
| FedEM | 30.7 ± 7.0 | 46.1 ± 7.0 | 23.0 ± 4.8 | 31.7 ± 3.6 | 53.3 ± 2.2 | 15.5 ± 2.9 |
| FeSEM | 69.6 ± 5.7 | 59.0 ± 5.6 | 31.1 ± 4.8 | 26.2 ± 3.7 | 61.0 ± 1.9 | 33.8 ± 0.9 |
| CFL | 76.6 ± 3.9 | 65.6 ± 4.8 | 33.9 ± 4.7 | 28.9 ± 3.5 | 62.3 ± 2.2 | 34.8 ± 1.0 |
| IFCA | 44.1 ± 9.9 | 36.9 ± 9.2 | 27.4 ± 4.7 | 15.7 ± 4.6 | 52.8 ± 2.8 | 33.5 ± 4.5 |
| pFedMe | 53.2 ± 7.4 | 43.6 ± 6.8 | 24.2 ± 4.1 | 15.8 ± 2.2 | 58.2 ± 1.1 | 34.6 ± 1.6 |
| APFL | 70.0 ± 5.8 | 56.9 ± 5.8 | 31.4 ± 4.5 | 29.7 ± 3.1 | 54.4 ± 0.9 | 39.4 ± 2.2 |
| FedDrift | 57.0 ± 7.7 | 47.6 ± 7.2 | 29.2 ± 4.9 | 20.2 ± 3.5 | 72.3 ± 0.8 | 42.1 ± 2.4 |
| ATP | 72.1 ± 10.5 | 61.1 ± 12.1 | 28.7 ± 5.1 | 16.7 ± 3.8 | N/A | 40.8 ± 4.3 |
| FEROMA | 90.7 ± 1.8 | 79.9 ± 2.8 | 44.2 ± 3.8 | 39.9 ± 2.5 | 72.4 ± 0.6 | 42.4 ± 1.4 |

Table 1: Mean accuracy and standard deviation across different datasets, comparing FEROMA and baselines under varying drifting frequency, non-IID types and levels.

| Non-IID Level | Low | | | Medium | | | High | | |
|---|---|---|---|---|---|---|---|---|---|
| # Drifting | 5 / 20 | 10 / 20 | 20 / 20 | 5 / 20 | 10 / 20 | 20 / 20 | 5 / 20 | 10 / 20 | 20 / 20 |
| FedAvg | 72.1 ± 8.0 | 76.5 ± 4.7 | 79.3 ± 2.6 | 71.5 ± 4.2 | 73.6 ± 5.3 | 75.8 ± 2.6 | 63.5 ± 6.6 | 65.1 ± 6.6 | 68.6 ± 6.5 |
| FedRC | 40.5 ± 9.7 | 41.4 ± 7.3 | 44.8 ± 9.7 | 23.8 ± 4.2 | 26.6 ± 6.6 | 28.3 ± 4.1 | 21.1 ± 7.0 | 22.9 ± 7.2 | 28.6 ± 2.4 |
| FedEM | 40.3 ± 11.2 | 43.2 ± 7.3 | 44.5 ± 10.5 | 21.7 ± 4.8 | 25.8 ± 5.2 | 28.7 ± 5.1 | 20.2 ± 4.9 | 23.7 ± 6.2 | 28.4 ± 3.5 |
| FeSEM | 72.4 ± 5.7 | 75.8 ± 4.0 | 77.8 ± 3.8 | 69.6 ± 6.0 | 71.5 ± 3.6 | 72.0 ± 8.7 | 60.2 ± 6.2 | 61.6 ± 6.8 | 65.4 ± 4.3 |
| CFL | 79.1 ± 4.3 | 78.1 ± 4.0 | 82.5 ± 3.0 | 76.6 ± 4.8 | 78.3 ± 3.6 | 78.1 ± 3.9 | 69.5 ± 4.4 | 72.4 ± 3.7 | 74.5 ± 2.9 |
| IFCA | 51.0 ± 9.9 | 52.9 ± 10.9 | 51.1 ± 13.0 | 46.1 ± 12.1 | 45.3 ± 8.7 | 40.5 ± 4.4 | 40.6 ± 9.8 | 35.8 ± 8.4 | 33.4 ± 8.8 |
| pFedMe | 54.9 ± 7.3 | 59.3 ± 7.0 | 58.5 ± 9.9 | 53.3 ± 9.0 | 54.3 ± 7.2 | 53.6 ± 5.0 | 47.3 ± 5.2 | 47.8 ± 6.2 | 49.9 ± 8.4 |
| APFL | 69.0 ± 6.4 | 72.4 ± 4.7 | 76.1 ± 7.2 | 70.0 ± 7.4 | 71.8 ± 5.6 | 71.9 ± 4.4 | 64.5 ± 4.2 | 64.8 ± 5.3 | 69.3 ± 6.0 |
| FedDrift | 58.8 ± 9.0 | 63.1 ± 6.4 | 58.4 ± 11.1 | 58.1 ± 9.0 | 60.8 ± 7.8 | 56.6 ± 6.0 | 52.3 ± 6.0 | 52.7 ± 6.8 | 52.5 ± 5.5 |
| ATP | 70.8 ± 12.8 | 78.5 ± 10.7 | 83.9 ± 6.5 | 65.8 ± 14.2 | 77.6 ± 5.3 | 78.0 ± 8.7 | 59.1 ± 12.8 | 65.1 ± 11.5 | 70.4 ± 8.4 |
| FEROMA | 90.6 ± 2.9 | 91.4 ± 1.0 | 92.1 ± 1.1 | 90.0 ± 2.7 | 90.6 ± 1.8 | 91.0 ± 1.8 | 90.2 ± 1.2 | 89.8 ± 1.6 | 90.8 ± 0.8 |

Table 2: Performance comparison across three non-IID levels and three drifting levels of all non-IID types on the MNIST dataset. **5 / 20, 10 / 20, 20 / 20**: Drifting 5 / 10 / 20 times in overall 20 rounds.

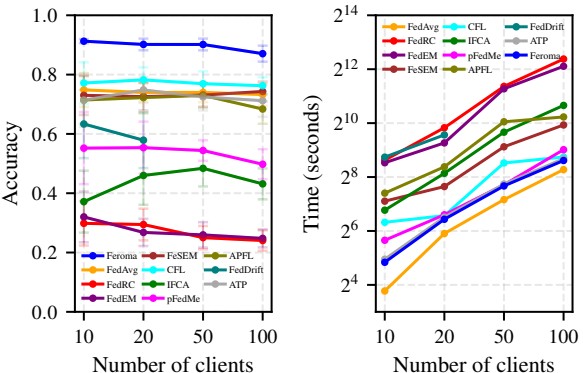

Figure 4: Performance comparison across varying numbers of clients. Left: Mean accuracy and standard deviation. Right: Training time per 20 rounds.

Despite the increased data heterogeneity and reduced data size introduced by scaling up the client number, Figure 4 shows that FEROMA consistently achieves the highest accuracy across all settings—exceeding 90% with even 50 clients, and maintaining over 85% accuracy with 100 clients. It outperforms the best baseline CFL by more than 10 pp at the largest scale. In addition to accuracy, FEROMA demonstrates strong computational and communication efficiency, with training times comparable to FedAvg. In contrast, most other baselines experience a sharp increase in runtime as the number of clients grows. This efficiency is attributed to the lightweight nature of distribution profiles which introduce minimal overhead throughout the training process. The results underscore the robustness and scalability of FEROMA in large-scale, dynamic FL scenarios.

| Algorithm | Cat. | D. shift | D. drift | T. adapt. | Low comm. | Low comp. (Server) | Low comp. (Client) | Scalability |
|---|---|---|---|---|---|---|---|---|
| FedAvg | N/A | | | | ✓ | ✓ | ✓ | ✓ |
| FedRC | CFL | ✓ | | | | ✓ | | ○ |
| FedEM | CFL | ✓ | | | | ✓ | | ○ |
| FeSEM | CFL | ✓ | | | ✓ | | ✓ | |
| CFL | CFL | ✓ | | | ✓ | | ✓ | |
| IFCA | CFL | ✓ | | | | ✓ | | ○ |
| pFedMe | PFL | ✓ | | | ✓ | ✓ | ○ | ✓ |
| APFL | PFL | ✓ | | | ✓ | ✓ | | ✓ |
| FedDrift | CFL | ✓ | ✓ | | | ✓ | | |
| ATP | TTA-FL | | | ✓ | ✓ | ✓ | ○ | ✓ |
| FedProto | PFL | ✓ | | | ✓ | ✓ | ✓ | ✓ |
| FedGPS | PFL | ✓ | | | ✓ | ○ | ✓ | ○ |
| FOOGD | PFL | ✓ | | | ✓ | ○ | ✓ | ○ |
| ShiftEx | PFL | ✓ | ✓ | | ○ | ○ | ✓ | ○ |
| CoLA | TTA-FL | N/A | N/A | ✓ | N/A | N/A | N/A | N/A |
| **FEROMA** | N/A | ✓ | ✓ | ✓ | ✓ | ✓ | ✓ | ✓ |

Table 3: **Qualitative comparison among FEROMA and baselines**. **Cat.**: FL category. **D. shift/drift**: Designed to tackle distribution shift/drift. **T. adapt.**: Designed to adapt test time distribution. **Low comm.**: Low communication cost (comparable to FedAvg). **Low comp. (Server/Client)**: Low computational cost on server/client side (comparable to FedAvg). **Scalability**: Scales efficiently to large client number. **CFL**: Clustered FL. **PFL**: Personalized FL. **TTA-FL**: Test time adaptive FL. ✓: Property satisfied. ○: Property conditionally satisfied.

**Key Observations.** FEROMA demonstrates consistent robustness across datasets, non-IID severities, and drifting frequencies. By dynamically selecting aggregation behavior at each round, it effectively recovers clustered or personalized FL when client distributions are similar, and falls back to global aggregation when distributions diverge sharply. This adaptability allows FEROMA to maintain strong performance in highly heterogeneous and unstable settings where fixed-strategy baselines degrade.

## 5 DISCUSSION

### 5.1 RELATED WORKS

FEROMA relates to three major lines of work in FL designed to address data heterogeneity: CFL, PFL, and TTA-FL. CFL methods (Sattler et al., 2021; Guo et al., 2024; Jothimurugesan et al., 2023; Ghosh et al., 2020; Marfoq et al., 2021; Long et al., 2023) group clients with similar data distributions and aggregate models accordingly. While effective under distribution shift, they typically assume a fixed number of clusters (Guo et al., 2024; Ghosh et al., 2020), require computationally intensive cluster procedures, and involve transmitting or maintaining multiple models per client (Jothimurugesan et al., 2023), limiting scalability. In contrast, FEROMA avoids explicit clustering by leveraging continuous distribution profiles for soft, data-driven association. PFL methods (Deng et al., 2020b; T. Dinh et al., 2020; Tan et al., 2023a; Kulkarni et al., 2020; Zhang et al., 2020a; Marfoq et al., 2022; Tan et al., 2022; Yang et al., 2025; Liao et al., 2024; Bhope et al., 2025) personalize models to each client's local distribution, improving performance when sufficient local data is available. However, they incur higher computation and storage costs on the client side and lack mechanisms for model assignment in cold-start or test-time scenarios. FEROMA achieves a similar personalization effect when required, by matching profiles without per-client optimization, and with significantly lower system overhead. TTA-FL methods (Bao et al., 2023; Deng et al., 2020a; Wang et al., 2019; Liang et al., 2025; Rajib et al.; Chen et al., 2024a) are designed to handle test-time drift via online adaptation, which requires additional client interaction or retraining. FEROMA, by contrast, supports test-time adaptation by matching profiles to observed ones, requiring no further updates or communication. We summarize the qualitative comparison in Table 3 and provide more details in Appendix A.

### 5.2 LIMITATIONS AND FUTURE WORKS

**Extractor dependence.** While prior works represent client distributions using model parameters, gradients, or training metrics—which exhibit intrinsic limitations (see Appendix A)—the effectiveness of FEROMA similarly depends on the quality of its DPE, which must generate reliable representations of client data distributions. In our implementation, the DPE relies on a few-round pretrained model to embed sampled data into a latent space. This approach may be limited in two scenarios: (1) if the model is undertrained—e.g., due to a difficult task or limited data—the resulting latent space may not

adequately reflect the underlying distribution; (2) if the model is overly simplistic or overly complex, the extracted representations may be uninformative or unstable. In both cases, suboptimal profiles may impair the accuracy of distribution mapping and reduce the overall robustness of FEROMA.

**Unseen Distributions.**    FEROMA associates models with training-time data distributions, but it does not explicitly address two challenging scenarios: (1) distributions that were seen during training but are absent in the final round, and thus not retained; and (2) entirely unseen distributions at test time. (See Appendix C for further discussion) However, FEROMA demonstrates strong generalization, as it assigns the most relevant model based on profile similarity—even for distributions not directly observed during training. Moreover, the models associated with final-round profiles offer strong initialization for downstream personalization or unsupervised adaptation. Unlike methods that begin personalization from a generic global model, FEROMA provides a well-trained, distribution-aware starting point. Future work could periodically checkpoint model in sparsely populated regions of the descriptor space, ensuring that rare or transient distributions are retained for test deployment.

## 6   CONCLUSIONS

In this work, we proposed FEROMA, an FL framework that explicitly addresses both distribution shift and drift across all four major types of data heterogeneity. By leveraging lightweight, differentially private distribution profiles to represent client data, FEROMA enables adaptive model aggregation based on distributional similarity without relying on any prior knowledge. This profile-based design supports both training and test-time adaptation, allowing FEROMA to generalize across dynamic client populations and unseen distributions, without requiring retraining or personalization from scratch. Through extensive experiments, we demonstrated that FEROMA consistently improves robustness and performance across a wide range of non-IID scenarios, with minimal overhead. Unlike prior methods that specialize in clustered, personalized, or adaptive FL, FEROMA unifies these strategies under a single framework—scalable, adaptable, and suitable for real-world heterogeneous deployments. This work lays the foundation for distribution-driven FL, and opens new directions for profile-based personalization, distribution tracking, and generalization to unseen client data in future systems.

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

APPENDIX CONTENTS

## A    RELATED APPROACHES TO DISTRIBUTION SHIFT AND DRIFT IN FEDERATED LEARNING

A rich body of research has emerged to mitigate data heterogeneity in FL. Existing approaches can be broadly grouped into three categories: *clustered FL (CFL)*, *personalised FL (PFL)*, or *test-time adaptive FL (TTA-FL)*. Below we summarise their core ideas and limitations and contrast them with FEROMA.

**Clustered Federated Learning (CFL)**    CFL assumes that the federation comprises $M$ data distributions and aims to learn one model per distribution. Two variations are common:

- *Hard-CFL.* Each client is assigned to *exactly one* cluster $c_m \subseteq \mathcal{K}_t$ based on predefined criteria or similarity measures, such that clusters are disjoint and collectively exhaustive: $\bigcup_{m=1}^{M} c_m = \mathcal{K}_t$ and $c_m \cap c_n = \emptyset$ for all $m \neq n$. In principle, each cluster $c_m$ contains a distinct subset of clients with similar data distributions, enabling the training of specialized models. Algorithms typically alternate between updating cluster models and re-assigning clients, using clustering on model parameters (Sattler et al., 2021; Gong et al., 2024) or performance-based metrics (Ghosh et al., 2020; Jothimurugesan et al., 2023; Morafah et al., 2023; Cai et al., 2023). Hard assignments simplify optimisation but struggle when distributions overlap, and they generally require prior knowledge of the number of clusters $M$—which is generally unavailable in practice.
- *Soft-CFL.* This approach allows clients to belong to multiple clusters with certain probabilities, accommodating scenarios where client data may exhibit overlapping distributions. Accordingly, each client holds a probability vector $\pi^{(k)} \in \mathbb{R}^M$ and trains an ensemble of $M$ models; examples include FedEM (Marfoq et al., 2021) and FedRC (Guo et al., 2024). The joint optimisation of $\{\pi^{(k)}\}_{k=1}^K$ and model parameters $\{\theta_m\}_{m=1}^M$ is typically tackled via Expectation–Maximization or alternating minimisation (Zhou et al., 2011), which incurs additional communication and may suffer from convergence issues. Soft-CFL also requires per-client labeled data to estimate $\pi^{(k)}$, making cold-start and unseen-client scenarios problematic.

**Personalized Federated Learning (PFL).**    PFL reframes FL as a client-centric optimization problem, where each client $k$ learns its own personalized model $f^{(k)}$ that minimizes its local loss. This approach explicitly addresses the presence of heterogeneous non-IID data distributions across clients, rather than enforcing a single global model for all participants. Broadly, PFL methods fall into three classes:

- *Fine-tuning.* A global model is first trained and then locally adapted through additional gradient steps or meta-learning techniques to simplify personalization (Chen et al., 2019; Fallah et al., 2020; Jiang et al., 2023). While simple, fine-tuning requires careful hyperparameter selection (e.g. learning rates, number of steps) and sufficient per-client data to avoid overfitting.
- *Model decoupling.* The network is partitioned into shared (global) and private (local) components. A common strategy is to jointly train a shared backbone while equipping it with separate global and personalized heads (Arivazhagan et al., 2019; Deng et al., 2020b; Collins et al., 2021; Jiang & Lin, 2022; Marfoq et al., 2022). Others personalize only batch-norm statistics (Li et al., 2021b) or allow heterogeneous encoder architectures (Diao et al., 2020). These approaches improve representational capacity at the cost of increased on-device model size and computation.
- *Regularization-based.* These methods introduce a regularization term to balance local and global objectives. For example, a popular strategy is to augment each client's loss with a penalty that ties the personalized model $\phi^{(k)}$ to the global model $\theta$: $\min_{\phi^{(k)}} \mathcal{L}^{(k)}(\phi^{(k)}) + \frac{\lambda}{2} \left\| \phi^{(k)} - \theta \right\|^2$, where $\lambda$ balances local versus global objectives (Fallah et al., 2020; Li et al., 2021a). Such bi-level formulations yield smooth personalization but introduce per-client hyperparameters, and nested optimization loops.

Despite their effectiveness when large amount of labeled data are available, PFL methods degrade with limited local samples, incur additional client-side compute and memory costs, and—being inherently supervised—are ill-suited for unseen, unlabeled clients or distribution drift at test time.

**Test-time adaptive federated learning (TTA-FL).**    TTA-FL addresses *post-deployment* distribution shift: after global training has concluded, each client adapts the received model to its own (unlabeled) test data. Most methods rely on *unsupervised* objectives such as entropy minimisation (Jiang & Lin,

2022; Bao et al., 2023; Rajib et al.) or self-supervised contrastive losses (Tan et al., 2023b; Chen et al., 2022; Liang et al., 2025). Since gradients must be estimated without labels, the optimization landscape is often ill-conditioned (e.g., in the presence of concept shift): entropy minimization can drive the model toward over-confident but incorrect predictions, while contrastive objectives may collapse when test batches are small or unbalanced—a common situation in on-device FL. To mitigate this instability, recent works restrict adaptation to a few parameters (e.g., batch-norm statistics or a single gating weight that interpolates between global and personalized heads (Jiang & Lin, 2022)). However, these approaches still require labeled data during training to learn the personalized head, making them unsuitable for unseen and unlabeled clients at test time.

**Contrast with FEROMA.**  While PFL, CFL, and TTA-FL each address aspects of client heterogeneity, FEROMA unifies their strengths in a lightweight, privacy-preserving framework:

- **vs. PFL.** Instead of learning a separate model $\phi^{(k)}$ for each client—incurring per-client hyperparameters, inner-loop optimization, and fine-tuning overhead (Arivazhagan et al., 2019; Deng et al., 2020b)—FEROMA optimizes a *single* model per observed distribution. Each client then receives its best-matching distribution slice based on a DP-protected profile, eliminating the need for client-specific adaptation or supervision. In doing so, FEROMA shifts the objective from client-centric to distribution-centric optimization.
- **vs. CFL.** Unlike hard- or soft-CFL methods that assume $M$ disjoint distributions known a priori (Guo et al., 2024; Jothimurugesan et al., 2023; Ghosh et al., 2020; Marfoq et al., 2021; Long et al., 2023), FEROMA imposes no such assumptions. Its distribution profiles adapt continuously to overlapping or drifting distributions and naturally support unseen clients via the label-agnostic component (R2). Moreover, FEROMA avoids the need to train multiple models per client or transmit multiple updates per round, thereby reducing both computation and communication costs compared to CFL approaches (see Table 3).
- **vs. TTA-FL.** Rather than relying on costly and unstable unsupervised fine-tuning at test time (e.g., entropy or contrastive minimisation (Jiang & Lin, 2022; Chen et al., 2022)), FEROMA performs a one-shot profile extraction followed by nearest-distribution association. This eliminates the risk of overconfident predictions or collapsed representations, incurs negligible overhead (R5), and generalizes seamlessly to unseen, unlabeled clients.

## B  IMPLEMENTATION DETAILS

### B.1  ALGORITHM PSEUDO-CODE

We provide the pseudo-code for the training phase (Algorithm 1) and inference phase (Algorithm 2) of the proposed FEROMA framework. Both algorithms detail the server-side and client-side operations involved at each round. During training, FEROMA extracts distribution profiles from client data, maps them to previous-round profiles, and assigns models accordingly for local updates (Line 6 and 11 in Algorithm 1). During inference, FEROMA matches test clients to the closest available model based on distribution profile similarity, without requiring retraining (Line 9 in Algorithm 2).

### B.2  CODE, LICENSES AND HARDWARE

Our experiments were implemented using Python 3.12 and open-source libraries including Scikit-learn 1.5 (Pedregosa et al., 2011) (BSD license), Flower 1.11 (Beutel et al., 2022) (Apache License), and PyTorch 2.4 (Paszke et al., 2019) (BSD license). For visualization, we use Matplotlib 3.9 (Hunter, 2007) (BSD license) and Seaborn 0.13 (Waskom, 2021) (BSD license), For data processing, we use Pandas 2.2 (Wes McKinney, 2010) (BSD license). The datasets used in our experiments—MNIST (GNU license), FMNIST (MIT license), CIFAR-10, CIFAR-100, CheXpert (Irvin et al., 2019), and Office-Home—are freely available online. To ensure reproducibility, our code, along with detailed instructions for reproducing the experiments, is publicly accessible on GitHub[2] under the MIT license. We implement publicly available codes for our baselines (except FedAvg). All experiments were conducted on a workstation equipped with four NVIDIA RTX A6000 GPUs (48 GB each), two AMD EPYC 7513 32-Core processors, and 512 GB of RAM.

---

[2]https://github.com/dariofenoglio98/FEROMA.git

---

**Algorithm 1** The FEROMA Framework - Training Phase

---

**Require:** initial set of clients $\mathcal{K}_0 = \{1, 2, \ldots, K_0\}, \mathcal{A}_0 = \mathcal{K}_0$, initial global models $\{\theta_0^{(k)}\}_{k \in \mathcal{K}_1}$, initial
profiles $\{d_0^{(k)}\}_{k \in \mathcal{K}_1}$, number of rounds $R$, DPE $\phi_\psi$, distance function $\mathcal{D}(\cdot, \cdot)$.
1: **for** $t = 1$ **to** $R$ **do**
2:      // Server-side
3:      $\mathcal{K}_t \leftarrow$ UPDATECLIENTPOOL$(\mathcal{K}_{t-1})$ {# Update set of available clients}
4:      $\mathcal{A}_t \leftarrow$ CLIENTSELECT$(\mathcal{K}_t)$ {# Sample clients for training round}
5:      // Client-side
6:      **for** each client $k \in \mathcal{A}_t$ **in parallel do**
7:          $d_t^{(k)} \leftarrow \phi_\psi(x_t^{(k)}, y_t^{(k)})$ {# Def. 3.1}
8:          Send $d_t^{(k)}$ to the server
9:      **end for**
10:     // Server-side
11:     **for** each client $k \in \mathcal{A}_t$ **do**
12:         $\{\bar{w}_t^{(k,j)}\}_{j \in \mathcal{A}_{t-1}} \leftarrow$ PROFILEMAP$(d_t^{(k)}, \{d_{t-1}^{(j)}\}_{j \in \mathcal{A}_{t-1}})$ {# Eq. equation 3/equation 4}
13:         $\theta_t^{(k)} = \sum_{j \in \mathcal{A}_{t-1}} p_k \cdot \bar{w}_t^{(k,j)} \cdot \theta_{t-1}^{(j)}$ {# Eq. equation 5}
14:         Send $\theta_t^{(k)}$ to client $k$
15:     **end for**
16:     // Client-side
17:     **for** each client $k \in \mathcal{A}_t$ **in parallel do**
18:         $\theta^{(k)} \leftarrow \theta_t^{(k)}$
19:         $\theta^{(k)} \leftarrow$ LOCALUPDATE$(\theta^{(k)}, x_t^{(k)}, y_t^{(k)})$
20:         Send $\theta^{(k)}$ to the server
21:     **end for**
22: **end for**

---

**Algorithm 2** The FEROMA Algorithm - Inference Phase

---

**Require:** set of test clients $\mathcal{K}_{\text{test}} = \{1, 2, \ldots, K_{\text{test}}\}$, last-round client participation $\mathcal{A}_R$, last-round models
$\{\theta_R^{(k)}\}_{k \in \mathcal{A}_R}$ with profiles $\{d_R^{(k)}\}_{k \in \mathcal{A}_R}$, DPE $\phi_\psi$, distance function $\mathcal{D}(\cdot, \cdot)$.
1: // Client-side
2: **for** each client $k \in \mathcal{K}_{\text{test}}$ **in parallel do**
3:      $d_{\text{test}}'^{(k)} \leftarrow \phi_\psi(x_{\text{test}}^{(k)}, \mathbf{0})$ {# (R2)}
4:      Send $d_{\text{test}}'^{(k)}$ to the server
5: **end for**
6: // Server-side
7: $\{d_R'^{(j)}\}_{j \in \mathcal{A}_R} \leftarrow$ GETPRIME$(\{d_R^{(j)}\}_{j \in \mathcal{A}_R})$
8: **for** each client $k \in \mathcal{K}_{\text{test}}$ **do**
9:      $\{\bar{w}_{\text{test}}^{(k,j)}\}_{j \in \mathcal{A}_R} \leftarrow$ PROFILEMAP$(d_{\text{test}}'^{(k)}, \{d_R'^{(j)}\}_{j \in \mathcal{A}_R})$ {# Eq. equation 3}
10:     $j^* \leftarrow \arg\max_{j \in \mathcal{A}_R} \bar{w}_{\text{test}}^{(k,j)}$
11:     $\theta_{\text{test}}^{(k)} \leftarrow \theta_R^{(j^*)}$ {# Best-matching model}
12:     Send $\theta_{\text{test}}^{(k)}$ to client $k$
13: **end for**
14: // Client-side
15: **for** each client $k \in \mathcal{K}_{\text{test}}$ **in parallel do**
16:     $\theta^{(k)} \leftarrow \theta_{\text{test}}^{(k)}$
17:     $\hat{y}^{(k)} = f(x_{\text{test}}^{(k)}; \theta^{(k)})$
18: **end for**

---

## B.3 MODELS AND HYPER-PARAMETER SETTINGS

We employ a 5-fold cross-validation strategy to evaluate the model's performance, using random
seeds (from 42 to 46) for all experiments to ensure reproducibility. For datasets MNIST, FMNIST,
and CIFAR-10, we use the LeNet-5 (Lecun et al., 1998) as the base model. For datasets CIFAR-100,
CheXpert and Office-Home, we use the ResNet-9 (He et al., 2016) as the base model. Specifically,
we use CheXpert-v1.0-small, a subset from CheXpert, and use the first 120,000 samples for our
experiments. We train the model with a batch size of 64 for both training and testing. Each client

allocated 20% of their local data for evaluation. The FL process is conducted over 20 communication rounds, with 100% participation rate and each client performing 2 local epochs per round. The learning rate is set to 0.005, and a momentum value of 0.9 is applied to optimize the training process.

### B.4 DISTRIBUTION PROFILE EXTRACTION

This section details the implementation of the distribution profile extractor (DPE) $\phi$, which is used to capture the local data distribution of each client in a privacy-preserving yet consistent manner across the federation. In addition, we provide a requirement-by-requirement justification that it fulfills Definition 3.1.

#### B.4.1 DPE IMPLEMENTATION.

Let the last hidden layer of the global model produce, for client $k$ in round $t$, a matrix of latent vectors $s_t^{(k)} \in \mathbb{R}^{v^{(k)} \times z}$. Our extractor maps these latents to a $d$-dimensional profile $d_t^{(k)}$ in four privacy-preserving steps:

- S1 *Global alignment (one shot, no raw data).* Each client computes the element-wise minimum and maximum of its latents and sends only the two $z$-dimensional vectors ($2z$ floats) to the server. The server aggregates by coordinate-wise minimum and maximum, obtaining global bounds $[m^-, m^+]$, and broadcasts these bounds together with the current model weights. No gradients, labels, or raw examples leave the devices.
- S2 *Shared PCA on synthetic reference points.* Using an agreed-upon random seed, every client draws 200 points uniformly in the range $[m^-, m^+]$ and fits a PCA map $g : \mathbb{R}^z \to \mathbb{R}^l$ (with $l = 10$) on this *synthetic* dataset only. [3] Because both the seed and the data are identical, the resulting linear projector $g$ is the same on every client, ensuring that Euclidean geometry in the reduced space is comparable across the federation.
- S3 *Monte-Carlo moment computation.* Each client projects its latents $z_t^{(k)} = g\big(s_t^{(k)}\big) \in \mathbb{R}^{v^{(k)} \times l}$ and draws $M = 3$ independent Bernoulli($\gamma = 0.5$) masks (see subsection I.8 for the ablation study on $M$). For each mask $m$, it computes $(\mu_x^{(k,m)}, \Sigma_x^{(k,m)})$ and $\big\{(\mu_u^{(k,m)}, \Sigma_u^{(k,m)})\big\}_{u=1}^{U}$, then averages over $m$ to obtain $(\mu_x^{(k)}, \Sigma_x^{(k)})$ and $\big\{(\mu_u^{(k)}, \Sigma_u^{(k)})\big\}_{u=1}^{U}$. Here, $\mu_x^{(k)}, \Sigma_x^{(k)}$ are the mean and covariance of the reduced latents (approximating the marginal $P(X)$ of client data), and $(\mu_u^{(k)}, \Sigma_u^{(k)})$ are the corresponding class-conditional moments (approximating $P(Y \mid X)$). These averaged moments can be written coordinate-wise as $h_i(z_t^{(k)}, y_t^{(k)}) = \frac{1}{M} \sum_{m=1}^{M} h_i^{(m)}$, where each $h_i^{(m)}$ is an unbiased estimate of the corresponding full-sample statistic. Under sub-Gaussianity with proxy variance $\tau^2$, the variance of this estimator is bounded by $\tau^2/(M \gamma v^{(k)})$ (see B.4.4 for derivation).
- S4 *Differential-privacy sanitization & profile assembly.* The concatenated statistics are each perturbed with an independent Laplace mechanism ($\delta = 0$) (Dwork et al., 2006):

$$\eta_i \overset{\text{iid}}{\sim} \text{Laplace}(0, b_i), \quad b_i = \Delta_{1,i}/\varepsilon,$$

where $\Delta_{1,i}$ is the $\ell_1$-sensitivity of statistic $h_i$. For a mean or standard-deviation coordinate we conservatively bound $\Delta_{1,i} \leq \frac{\text{Range}(h_i)}{v^{(k)}}$. The released profile is therefore

$$d_t^{(k)} = [h_1, \dots, h_d]^\top + \eta, \quad \eta \sim \text{Laplace}\big(0, \text{diag}(b_1, \dots, b_d)\big).$$

Across the pipeline, the only raw, example-level information ever transmitted is the $2z$-float (i.e., min/max pair from *S1*), ensuring compliance with FL privacy constraints. Moreover, with $\varepsilon = 10.0$ and typical $v^{(k)} > 300$ and $\text{Range}(h_i) < 10$, the worst-case variance $2b_i^2 \leq 2.2 \times 10^{-5}$ is negligible relative to inherent data variability, yet it guarantees $(\varepsilon, 0)$-DP at the profile level.

---

[3] Because the synthetic points are fully determined by the shared seed and the broadcast bounds $[m^-, m^+]$, the same projector $g$ could equivalently be fitted once on the server and broadcast to clients. We compute it locally to avoid broadcasting PCA parameters (projection + mean, $(zl + z)$ floats) and to simplify updates when latent ranges evolve across rounds; this choice does not affect privacy.

### B.4.2 DISTRIBUTION FIDELITY (R1)

To meet the *distribution fidelity* requirement (R1), we design our DPE so that Euclidean distances in the profile space provably track the 2-Wasserstein distance $W_2$ between client data distributions. After aligning all clients with a globally consistent, privacy-preserving PCA (see Paragraph B.4.1), each profile concatenates the first two moments of (i) the marginal latent distribution $P(X)$ and (ii) the class-conditional latents $\{P(X\,|\,Y=u)\}_{u=1}^{U}$.

Consider the following assumptions:

(A1) (Latent Gaussianity) Each latent distribution can be approximated by a Gaussian: $P_i \approx \mathcal{N}(\mu_i, \Sigma_i)$. This is a standard assumption for deep features.

(A2) (Spectral bounds) There exist constants $0 < \lambda_{\min} \le \lambda_{\max} < \infty$ such that the spectra of all covariances lie in $[\lambda_{\min}, \lambda_{\max}]$. This is enforced in our case by Step S1 of the DPE, which ensures that all latent representations lie within a global bounding box $[m^-, m^+]$.

(A3) (Approximate commutation) Post-PCA, the covariances are (close to) diagonal; we use commutation to obtain tight identities and can otherwise rely on standard operator bounds.

We can define the following proposition for marginals, which then extends to class-conditionals. This result applies to the noise-free moment map (i.e., $\bar{\phi}$); the stochastic DPE output $d_t^{(k)}$ concentrates around $\bar{\phi}$ as quantified by (R3).

**Proposition 1** (Bi-Lipschitz equivalence to $W_2$ for marginals). *Define the profile distance for two clients as*

$$\Delta^2 \;=\; \|\mu_1 - \mu_2\|_2^2 \;+\; \|\Sigma_1 - \Sigma_2\|_F^2. \tag{1}$$

*If* (A1)–(A3) *hold, then for constants*

$$c_- \;=\; \min\{1, (2\sqrt{\lambda_{\max}})^{-1}\}, \qquad c_+ \;=\; \max\{1, (2\sqrt{\lambda_{\min}})^{-1}\},$$

*we have the two-sided bound*

$$c_-^2\, \Delta^2 \;\le\; W_2^2\big(\mathcal{N}(\mu_1, \Sigma_1), \mathcal{N}(\mu_2, \Sigma_2)\big) \;\le\; c_+^2\, \Delta^2.$$

*Consequently $W_2$ and $\Delta$ are bi-Lipschitz equivalent [4] on the set of admissible covariances.*

*Proof.* For notational simplicity, we write each client's marginal profile as $d^{(k)} = [\mu_k, \text{vec}(\Sigma_k)]$. Then the squared $\ell_2$ distance between two client profiles is

$$\|d^{(1)} - d^{(2)}\|_2^2 = \|\mu_1 - \mu_2\|_2^2 + \|\text{vec}(\Sigma_1 - \Sigma_2)\|_2^2.$$

For any matrix $A$, $\|\text{vec}(A)\|_2^2 = \|A\|_F^2$. With $A = \Sigma_1 - \Sigma_2$, we obtain Appendix equation 1:

$$\Delta^2 = \|d^{(1)} - d^{(2)}\|_2^2 = \|\mu_1 - \mu_2\|_2^2 + \|\Sigma_1 - \Sigma_2\|_F^2.$$

For Gaussians, $W_2^2 = \|\mu_1 - \mu_2\|_2^2 + B^2(\Sigma_1, \Sigma_2)$, where $B^2(\Sigma_1, \Sigma_2) = \text{Tr}\big(\Sigma_1 + \Sigma_2 - 2(\Sigma_1^{1/2}\Sigma_2\Sigma_1^{1/2})^{1/2}\big)$ is the squared Bures distance (Villani et al., 2008). The mean terms coincide; it remains to compare $B$ to $\|\Sigma_1 - \Sigma_2\|_F$.

Under (A3) (covariances commute $\Sigma_1\Sigma_2 = \Sigma_2\Sigma_1$—e.g., are diagonal in the shared PCA basis),

$$\Sigma_1^{1/2}\Sigma_2\Sigma_1^{1/2} = \Sigma_1^{1/2}\Sigma_2^{1/2}\Sigma_2^{1/2}\Sigma_1^{1/2} = \big(\Sigma_1^{1/2}\Sigma_2^{1/2}\big)^2 \quad \Rightarrow \quad \big(\Sigma_1^{1/2}\Sigma_2\Sigma_1^{1/2}\big)^{1/2} = \Sigma_1^{1/2}\Sigma_2^{1/2}.$$

Plugging this into the Bures formula:

$$B^2(\Sigma_1, \Sigma_2) = \text{Tr}(\Sigma_1) + \text{Tr}(\Sigma_2) - 2\,\text{Tr}\Big(\big(\Sigma_1^{1/2}\Sigma_2\Sigma_1^{1/2}\big)^{1/2}\Big) = \text{Tr}(\Sigma_1) + \text{Tr}(\Sigma_2) - 2\,\text{Tr}(\Sigma_1^{1/2}\Sigma_2^{1/2}).$$

Using the identity for the squared Frobenius norm of the difference between two symmetric matrices:

$$\big\|\Sigma_1^{1/2} - \Sigma_2^{1/2}\big\|_F^2 = \text{Tr}(\Sigma_1) + \text{Tr}(\Sigma_2) - 2\,\text{Tr}(\Sigma_1^{1/2}\Sigma_2^{1/2}) \quad \Rightarrow \quad B^2(\Sigma_1, \Sigma_2) = \big\|\Sigma_1^{1/2} - \Sigma_2^{1/2}\big\|_F^2.$$

---

[4]We use *bi-Lipschitz equivalence* in the standard metric-geometry sense: two distances $d_1, d_2$ are bi-Lipschitz equivalent on a set $\mathcal{B}$ if there exist constants $c_-, c_+ > 0$ such that for all $a, b \in \mathcal{B}$, $c_-\, d_1(a, b) \le d_2(a, b) \le c_+\, d_1(a, b)$. In our case, letting $\Delta(d_{t_1}^{(k_1)}, d_{t_2}^{(k_2)}) := \|d_{t_1}^{(k_1)} - d_{t_2}^{(k_2)}\|_2$, Proposition 1 establishes constants $c_-, c_+$ such that $c_-\, \Delta \le W_2\big(P_{t_1}^{(k_1)}, P_{t_2}^{(k_2)}\big) \le c_+\, \Delta$ over the admissible set considered in the proposition.

| Non-IID Level | Max | Min | Mean | Std |
|---|---|---|---|---|
| low | 1.060 | 0.194 | 0.526 | 0.106 |
| medium | 1.028 | 0.159 | 0.479 | 0.113 |
| high | 1.016 | 0.136 | 0.458 | 0.108 |

(a) 2-**Wasserstein**

| Non-IID Level | Max | Min | Mean | Std |
|---|---|---|---|---|
| low | 0.477 | 0.030 | 0.288 | 0.074 |
| medium | 0.517 | 0.009 | 0.332 | 0.084 |
| high | 0.535 | 0.007 | 0.352 | 0.078 |

(b) **Jensen–Shannon**

Table 4: Absolute error $\delta$ between profile distance and the true inter-client distance under three non-IID Levels on MNIST under $P(Y \mid X)$ concept shift.

By the mean value theorem for $f(x) = \sqrt{x}$ on $[\lambda_{\min}, \lambda_{\max}]$, applied entrywise,

$$\frac{1}{2\sqrt{\lambda_{\max}}} \, \|\Sigma_1 - \Sigma_2\|_F \;\leq\; B(\Sigma_1, \Sigma_2) \;\leq\; \frac{1}{2\sqrt{\lambda_{\min}}} \, \|\Sigma_1 - \Sigma_2\|_F.$$

Let $a = \|\mu_1 - \mu_2\|_2^2$, $b = \|\Sigma_1 - \Sigma_2\|_F^2$, and $k \in [k_{\min}, k_{\max}]$ with $k_{\min} = 1/(4\lambda_{\max})$, $k_{\max} = 1/(4\lambda_{\min})$. Then

$$\min\{1, k\}(a + b) \leq a + kb \leq \max\{1, k\}(a + b),$$

which yields $c_-^2 \, \Delta^2 \leq W_2^2 \leq c_+^2 \, \Delta^2$ with the stated $c_-$ and $c_+$.

$\square$

**Empirical validation.** We validate this guarantee on $44\,850$ client–round pairs generated from MNIST under three levels of non-IID concept shift that perturb $P(Y \mid X)$ (low, medium, high; see Appendix I.1 for the protocol). Table 4 reports the absolute error $\delta = \big| \|d_{t_1}^{(k_1)} - d_{t_2}^{(k_2)}\|_2 - D(P_{t_1}^{(k_1)}, P_{t_2}^{(k_2)}) \big|$. For the target 2-Wasserstein metric, the worst-case error never exceeds 1.1, with a mean of $0.49 \pm 0.11$. We also compute the gap w.r.t. the Jensen–Shannon (JS) distance to demonstrate robustness to the choice of $D$; the maximum JS error is $< 0.54$ and the mean is $0.32 \pm 0.08$. These results confirm that our extractor satisfies (R1) with $\xi < 1.1$ across a broad spectrum of distribution shifts while preserving local-data privacy.

### B.4.3 Label agnosticism (R2)

In a deployed FL system the clients encountered at test-time rarely possess reliable labels. Requirement (R2) therefore asks for a sub-vector of every profile that can be computed with *features only*. Our implementation already provides such a component:

- *Training phase.* Step *S3* produces *both* the marginal moments $(\mu_x^{(k)}, \Sigma_x^{(k)})$ and the class-conditional moments $\{(\mu_u^{(k)}, \Sigma_u^{(k)})\}_{u=1}^U$. The resulting profile splits naturally into

$$d_t^{(k)} = \big[ \underbrace{\mu_x^{(k)}, \Sigma_x^{(k)}}_{d_t'^{(k)} \in \mathbb{R}^p}, \ \underbrace{\mu_1^{(k)}, \Sigma_1^{(k)}, \ldots, \mu_U^{(k)}, \Sigma_U^{(k)}}_{d_t''^{(k)} \in \mathbb{R}^{d-p}} \big].$$

- *Test phase (labels unavailable).* The client repeats *S1–S2* unchanged, then executes the *label-free* part of *S3*, yielding only $(\mu_x^{(k)}, \Sigma_x^{(k)})$. Reasonably, the conditional distribution $P(Y \mid X)$ cannot be approximated at test time in the absence of labels. These statistics form the sub-vector $d_t'^{(k)} := \phi_\psi\big(x_t^{(k)}, \mathbf{0}\big) \in \mathbb{R}^p$, fully satisfying the formal condition in the main text. Step *S4* applies the same Laplace mechanism coordinate-wise, so $d_t'^{(k)}$ enjoys the same $(\varepsilon, \delta{=}0)$ differential-privacy guarantee as the full profile.

Because the PCA projector is shared (*S2*) and the noise calibration in *S4* is data-independent, Euclidean distances between two label-agnostic profiles, $\|d_{t_1}'^{(k_1)} - d_{t_2}'^{(k_2)}\|_2$, remain a meaningful proxy for the marginal Wasserstein distance between $P(X)$ distributions. Consequently, our DPE allows us to match, at test time, unseen and unlabeled clients to the closest marginal distributions fitted during training—satisfying Requirement (R2).

### B.4.4 CONTROLLED STOCHASTICITY (R3)

To thwart exact fingerprinting of a client whose distribution remains static across rounds—which would otherwise cause it to be matched with certainty at each round, potentially suppressing the contribution of other clients with similarly relevant distributions during aggregation—the extractor must output similar but not identical profiles for the same input—while preserving the true geometry in expectation. We achieve this with two independent randomness sources.

1. *Monte-Carlo subsampling.* Given $v^{(k)}$ examples, the client draws $M = 3$ independent Bernoulli($\gamma = 0.5$) masks and computes the moments of each subsample. Averaging these estimates yields the profile statistic $\tilde{h} := \frac{1}{M}\sum_{m=1}^{M} h^{(m)}$, where each $h^{(m)}$ is unbiased for the corresponding full-sample statistic $h = \bar{\phi}(z_t^{(k)}, y_t^{(k)})$. If every reduced latent coordinate is sub-Gaussian with proxy variance $\tau^2$, then

$$\mathrm{Var}(\tilde{h}) \leq \frac{\tau^2}{M\gamma v^{(k)}},$$

giving a data-dependent variance that shrinks both with sample size and with the number of Monte-Carlo replicas.

2. *Laplace mechanism (Step S4).* The zero-mean noise $\eta_i \sim \mathrm{Laplace}(0, b_i)$ adds fixed variance $2b_i^2$ per coordinate. Because the noise is independent of the subsampling, the total covariance is diagonal and bounded:

$$\mathrm{Cov}(d_t^{(k)} \mid z_t^{(k)}, y_t^{(k)}) \preceq \left(\frac{\tau^2}{M\gamma v^{(k)}} + 2b_{\max}^2\right)\mathbf{I}_d = \rho^2 \mathbf{I}_d. \tag{2}$$

Hence $\mathbb{E}[d_t^{(k)} \mid z_t^{(k)}, y_t^{(k)}] = \bar{\phi}(z_t^{(k)}, y_t^{(k)})$ and $\mathrm{Cov}(d_t^{(k)} \mid z_t^{(k)}, y_t^{(k)}) \preceq \rho^2 \mathbf{I}_d$. This exactly matches Requirement (R3) with $\rho^2 = \frac{\tau^2}{M\gamma v^{(k)}} + 2b_{\max}^2$. In our experiments ($v^{(k)} > 300$, $\tau^2 < 1.0$, $b_{\max} = 0.003$) this gives $\rho^2 \leq 2.2 \times 10^{-3}$, yielding profile distances that are stable across draws yet impossible to replicate perfectly—providing the desired controlled stochasticity.

### B.4.5 DIFFERENTIAL-PRIVACY GUARANTEE (R4)

Because FEROMA transmits client-side profiles $\{d_t^{(k)}\}_{k\in\mathcal{K}_t} \subset \mathbb{R}^d$—which, by design, capture each client's data distribution—an adversary could, in principle, combine them with model updates to mount stronger data reconstruction or membership inference attacks (Shokri et al., 2017; Zari et al., 2021; Li et al., 2022a; Hitaj et al., 2017; Zhu et al., 2019; Yin et al., 2021). For this reason, Requirement (R4) mandates an $(\varepsilon, \delta)$-differential privacy guarantee at the *sample* level. The coordinate-wise Laplace mechanism (implemented in *S4*) ensures $(\varepsilon, 0)$-DP for the entire profile vector $d_t^{(k)}$, since each coordinate is perturbed with noise scaled to the same $\varepsilon$, and sensitivities are computed in the $\ell_1$ norm. Thus, for any neighbouring datasets $(x, y)$ and $(x', y')$ that differ in one example, and for any measurable set $\mathcal{S} \subseteq \mathbb{R}^d$,

$$\Pr[\phi_\psi(x, y) \in \mathcal{S}] \leq e^\varepsilon \Pr[\phi_\psi(x', y') \in \mathcal{S}],$$

with $\delta = 0$. In our experiments we set $\varepsilon = 10.0$; with typical client sizes $v^{(k)} > 300$, the added variance $2(\Delta_{1,i}/\varepsilon)^2 \leq 2.2 \times 10^{-5}$ is negligible compared to natural data variability, so inter-profile distances remain reliable. Combined with the many-to-one nature of $\phi_\psi : \mathbb{R}^{v^{(k)} \times (z+u)} \to \mathbb{R}^d$—where infinitely many distinct datasets map to the same profile, making inversion information-theoretically impossible—this DP mechanism bounds any additional leakage to an $\varepsilon$-limited factor beyond what is already exposed by model parameters, thereby fully satisfying Requirement (R4).

### B.4.6 COMPACTNESS (R5)

Requirement (R5) demands that profile extraction adds only marginal computation and communication overhead relative to standard FL. Our implementation meets this target in both aspects.

- *Computation.* The only non-trivial operation is fitting a rank-$l$ PCA on $s^{\mathrm{PCA}} = 200$ synthetic latent vectors of dimension $z$ (Step *S2*). Using an SVD solver, the cost is $O(\min(s^{\mathrm{PCA}} z^2, (s^{\mathrm{PCA}})^2 z))$. In practice $z > s^{\mathrm{PCA}}$, so the second term dominates: $(s^{\mathrm{PCA}})^2 z =$

$200^2 z \approx 4 \times 10^4 z$ floating-point operations. For typical latent sizes ($z \in [128, 2048]$), this results in at most $8.2 \times 10^7$ FLOPs—negligible compared to the cost of a single local training epoch. For reference, a single forward pass (no backward, no optimization) with our smallest network on MNIST exceeds $6.5 \times 10^8$ FLOPs, while the largest network used on CIFAR-100 requires over $6 \times 10^{11}$ FLOPs per epoch. Moment computation (Step *S3*) involves simple summations and products, and is therefore negligible, while Laplace noise injection (Step *S4*) has complexity $O(d)$.

- *Communication.* Each profile transmits $d = (l + l) \times (1 + U)$ floats, i.e. a mean and a (diagonal) covariance entry per latent dimension for the marginal distribution plus the same for each of the $U$ classes. With $l = 10$ this yields $d = 220$ floats for MNIST ($U = 10$) and $d = 2\,020$ for CIFAR-100 ($U = 100$). By contrast, the model updates in our experiments range from 62 006 parameters (MNIST) to 6 775 140 parameters (CIFAR-100). Hence the profile occupies at most $d/|\theta| \leq 3.5 \times 10^{-3}$ of the uplink payload, comfortably satisfying the compactness criterion $d/|\theta| \leq 10^{-2}$.

These results demonstrate that the DPE introduces only minimal overhead while maintaining the communication efficiency expected in FL, thereby fully satisfying Requirement (R5).

### B.5 IMPACT OF DPE QUALITY

FEROMA relies on DPEs to summarize client data distributions, and its effectiveness therefore depends on the quality of the extracted profiles. This dependency is not unique to FEROMA: many personalized or clustered FL methods implicitly rely on representation quality through gradients, model updates, or learned embeddings. FEROMA makes this dependence explicit through compact distribution profiles, but does not introduce a fundamentally new requirement beyond what is already assumed in modern FL systems.

Importantly, FEROMA is designed to remain stable even when the DPE is imperfect or noisy. The profiles produced by the DPE are intentionally low-dimensional and capture coarse, distribution-level statistics rather than fine-grained features. As a result, aggregation decisions depend only on relative profile distances, not on exact representations. Noise or approximation error therefore affects FEROMA only through these distances. In practice, the thresholding mechanism further filters out uninformative or noisy associations, and stable aggregation strategies emerge as long as relative similarities are approximately preserved. This behavior is confirmed by the sensitivity analysis which shows that moderate degradation in profile quality—e.g., through increased differential privacy noise—leads to only mild performance impact.

While our implementation instantiates the DPE using a few-round warm-up model, FEROMA itself does not require a pretrained encoder. The DPE can be realized using a variety of mechanisms, including lightweight public encoders, simple statistical descriptors computed directly on the data (e.g., feature moments or sketching), or any mapping that satisfies Definition 3.1. Moreover, FEROMA does not require profiles to be available from the first training round: early rounds can follow standard FedAvg, with profile-based aggregation activated once a minimally reasonable encoder or statistic becomes available. This design makes FEROMA applicable even in low-resource settings where no strong pretrained model exists.

Overall, FEROMA does not require task-optimal representations. It only assumes that the DPE provides consistent embeddings or statistics that approximately preserve distributional differences across clients. Under this mild assumption, FEROMA remains robust and effective across a wide range of heterogeneity settings.

## C DISCUSSION: HANDLING SEEN-ONCE OR UNSEEN DISTRIBUTIONS

As illustrated in Figure 5 and Figure 6, there are two drifting scenarios that FEROMA does not explicitly target: **(1) Seen-once distributions**, which appear during intermediate training rounds but are absent in the final round; and **(2) Unseen distributions**, which never occur during training but appear during testing. These two conditions may also coexist.

While FEROMA retains only final-round profiles and models by default, it can be extended to address both cases effectively. For seen-once distributions, the server can optionally store their corresponding

profiles and models during training, provided they yield acceptable validation performance. This enables FEROMA to retain models for all distributions encountered during training, not just those from the final round. For entirely unseen distributions, which are fundamentally unpredictable, FEROMA still assigns the closest available model based on distribution profile similarity. Unlike standard baselines (e.g., FedAvg) that rely on a single global model, FEROMA selects from a diverse set of models trained on different distributional modes. This leads to better initial performance and provides a stronger starting point for downstream personalization or test-time adaptation. In this sense, FEROMA complements and can be naturally integrated with test-time adaptive FL methods to further improve robustness in dynamic environments.

**Empirical validation.** Importantly, both seen-once and unseen distribution scenarios are inherently present in our experimental setup. Our dynamic FL experiments with varying drift frequencies (every 2-4 rounds) naturally create seen-once distributions as client data evolves over time. Similarly, our test-time evaluation on cold-start and test-only clients directly evaluates performance on unseen distributions. The consistent performance gains of FEROMA across all experimental conditions—achieving up to 12 percentage points improvement over baselines—demonstrate that our framework effectively handles these challenging scenarios in practice. This empirical evidence validates that distribution-profile-based model selection provides robust generalization even when exact distributional matches are unavailable.

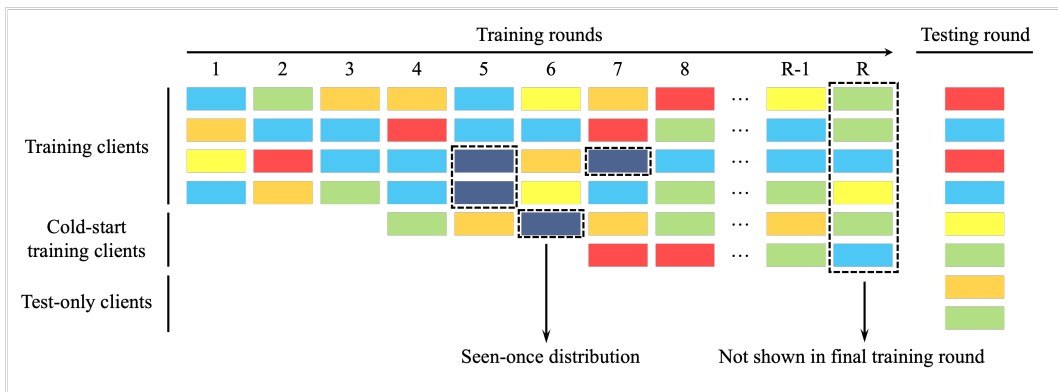

Figure 5: **Seen-once distribution in training stage.**

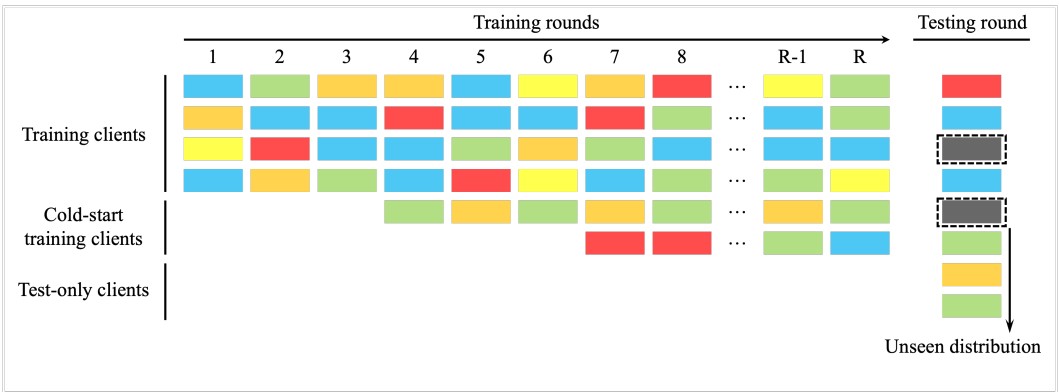

Figure 6: **Unseen distribution in testing stage.**

## D   FEROMA ROBUSTNESS AGAINST MALICIOUS CLIENTS

In this section, we discuss the robustness of FEROMA against malicious or dishonest clients that attempt to manipulate the training process (Xia et al., 2023; Fenoglio et al., 2024). While FEROMA

is not explicitly designed as a Byzantine-robust FL algorithm, its profile-based design introduces several natural protections that significantly limit the effectiveness and impact of adversarial behavior.

**Threat model.** We consider adversarial clients that may (i) arbitrarily manipulate their local training updates, (ii) deliberately distort their extracted distribution profiles, or (iii) attempt to infer or target the data distributions of other clients. We do not assume the presence of secure hardware on the client side, but we assume standard authenticated communication with the server, as in most FL systems.

**Limited information exposure.** In FEROMA, clients do not have access to the distribution profiles, similarity weights, or aggregation decisions of other clients. The server never broadcasts the global set of profiles, nor does it reveal the weights assigned to a client in previous rounds. Profiles are not shared among clients, and only the aggregated model (or assigned model) is returned. As a result, malicious clients lack the information required to meaningfully infer, target, or impersonate other clients' data distributions, even across multiple rounds.

**Minimal influence of manipulated profiles.** Even under an unusually strong adversary that can observe all profiles across rounds, deliberately manipulating a profile yields limited influence. Aggregation weights in FEROMA are derived from profile distances: a profile that deviates substantially from the population receives negligible weight and is removed entirely by the thresholding mechanism. Conversely, an adversary attempting to craft a profile that mimics a specific target distribution can only affect the corresponding local neighborhood in profile space—equivalent to influencing a small clustered subgroup or, in the extreme case, a single personalized branch. The remainder of the system remains unaffected, preventing global poisoning effects.

**Lightweight profiles and secure transmission.** The above strong adversary scenario—full access to all profiles—is highly unlikely in practice. Nevertheless, FEROMA is designed to remain robust even under such assumptions. Because distribution profiles are intentionally lightweight, they can be efficiently protected using standard encryption or secure-channel techniques. In contrast to traditional FL, where encrypting full model updates is often prohibitively expensive due to their size and frequency, encrypting FEROMA's compact profiles introduces negligible computational and communication overhead. This makes secure transmission practical and substantially reduces the likelihood of profile compromise.

Taken together, these properties imply that FEROMA naturally limits the attack surface available to malicious clients. While adversarial manipulation can still affect a small subset of closely matched distributions, the system as a whole remains stable and resilient. We view this robustness as a consequence of decoupling model aggregation from explicit client or cluster identities and instead relying on soft, similarity-based associations over compact distribution profiles. Formal defenses against fully Byzantine adversaries are complementary to FEROMA and represent an interesting direction for future work.

# E   TEST-TIME MODEL ASSOCIATION UNDER $P(Y|X)$ CONCEPT SHIFT

In practical settings, class-conditional distributions $P(X \mid Y = y)$ cannot be estimated at test time, as labels are typically unavailable. As a result, detecting $P(Y \mid X)$ concept shift during inference is a known impossibility: different label distributions can induce identical feature distributions, leaving no observable signal for detecting the shift based solely on input features. In real-world applications, $P(Y \mid X)$ shifts are often linked to evolving user preferences, and the only feasible solution is to query a small number of labeled examples at test time to infer the underlying preference.

To assess the feasibility of this approach, we evaluated FEROMA under varying amounts of labeled test samples. Specifically, we measured test-time model association accuracy on MNIST with $P(Y \mid X)$ shifts of increasing severity (low, medium, high non-IID), using between 1 and 50 labeled samples per class. Importantly, these labels were used solely for model selection and not for any model retraining. Results are reported in Figure 7. As expected, increasing the number of labeled samples improves the association quality, approaching the upper bound defined by the optimal association (shown as dashed lines). Moreover, stronger non-IID levels require more labeled examples to achieve a good association, reflecting the inherent difficulty of $P(Y \mid X)$ shift scenarios.

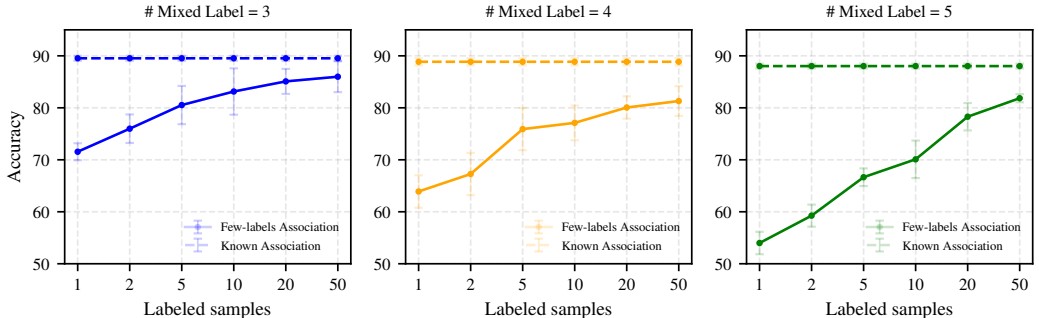

Figure 7: **Test accuracy of FEROMA on MNIST under $P(Y \mid X)$ concept shift, as a function of the number of labeled test samples** per class used for model association. Dashed lines indicate the performance under (optimal) known model assignment. Results are shown for three levels of non-IID severity.

**Practical instantiation.** From these results, we conclude that using 20 labeled samples per class strikes a good trade-off between performance and labeling cost. Accordingly, we adopt this configuration in all our experiments involving test-time association with FEROMA. To ensure a fair comparison, baseline methods are evaluated under (optimal) known association, since they do not natively address $P(Y \mid X)$ shifts or propose alternative strategies.

**Numerical experiments.** For clarity and completeness, we additionally tracked the performance of FEROMA under the (optimal) *known association* condition across all experiments. This comparison isolates the influence of test-time association errors from the effects of distribution drift and shift occurring during training. Tables 5–8 report the test accuracy of FEROMA when using either the known association or the automatically detected association (with 20 labeled test samples per class) across four datasets: MNIST, FMNIST, CIFAR-10, and CIFAR-100. As expected, the results show that under higher non-IID severity, the gap between known and detected association widens, reflecting the increasing difficulty of matching clients to appropriate models in the presence of stronger $P(Y|X)$ shifts. Nevertheless, FEROMA maintains strong performance even when relying on few labeled samples, demonstrating robustness to moderate test-time association errors.

In addition, these results confirm that when labels are available—as is the case during training—FEROMA consistently maintains high performance across all levels of non-IID severity and client drift rates. This stability highlights the robustness of the framework in handling both distribution shift and drift during training, demonstrating that FEROMA's adaptive aggregation remains effective even under highly heterogeneous and dynamic conditions.

| Non-IID Level | Low | | | Medium | | | High | | |
|---|---|---|---|---|---|---|---|---|---|
| # Drifting | 5 / 20 | 10 / 20 | 20 / 20 | 5 / 20 | 10 / 20 | 20 / 20 | 5 / 20 | 10 / 20 | 20 / 20 |
| Known Association | 89.58 ± 0.50 | 89.48 ± 0.45 | 89.80 ± 0.75 | 88.90 ± 0.53 | 88.87 ± 0.68 | 89.01 ± 0.64 | 88.15 ± 0.66 | 88.06 ± 0.51 | 88.41 ± 0.68 |
| Few labeled samples | 83.44 ± 3.24 | 85.04 ± 2.44 | 84.76 ± 1.47 | 78.15 ± 3.40 | 80.09 ± 2.14 | 80.33 ± 3.14 | 78.65 ± 2.45 | 78.38 ± 2.62 | 79.65 ± 1.26 |

Table 5: Test accuracy of FEROMA on MNIST under known test-time model association and automatic detection with few labeled samples (20 per class). Results are reported across low, medium, and high non-IID levels, with varying numbers of drifting clients.

# F PRIVACY IMPLICATIONS OF DISTRIBUTION PROFILE

Besides model parameters, FEROMA also transmits the client–side profiles $\{d_t^{(k)}\}_{k \in \mathcal{K}_t} \subset \mathbb{R}^d$. Because, by design, profile distances approximate distribution divergences (requirement R1 in 3.1), an honest-but-curious server (or an external eavesdropper) could leverage $d_t^{(k)}$ together with model updates to mount stronger data–reconstruction or membership–inference attacks (e.g. (Shokri et al., 2017; Zari et al., 2021; Li et al., 2022a; Hitaj et al., 2017; Zhang et al., 2020b; Ren et al., 2022;

| Non-IID Level | Low | | | Medium | | | High | | |
|---|---|---|---|---|---|---|---|---|---|
| **# Drifting** | **5 / 20** | **10 / 20** | **20 / 20** | **5 / 20** | **10 / 20** | **20 / 20** | **5 / 20** | **10 / 20** | **20 / 20** |
| Known Association | 75.37 ± 0.33 | 75.23 ± 0.42 | 75.25 ± 0.27 | 74.83 ± 0.38 | 74.64 ± 0.37 | 74.69 ± 0.36 | 74.29 ± 0.37 | 73.83 ± 0.42 | 73.93 ± 0.26 |
| Few labeled samples | 72.49 ± 1.69 | 70.62 ± 2.79 | 72.48 ± 1.64 | 67.14 ± 1.95 | 68.61 ± 1.40 | 66.74 ± 1.66 | 65.11 ± 1.15 | 63.47 ± 1.57 | 63.08 ± 2.01 |

Table 6: Test accuracy of FEROMA on FMNIST under known test-time model association and automatic detection with few labeled samples (20 per class). Results are reported across low, medium, and high non-IID levels, with varying numbers of drifting clients.

| Non-IID Level | Low | | | Medium | | | High | | |
|---|---|---|---|---|---|---|---|---|---|
| **# Drifting** | **5 / 20** | **10 / 20** | **20 / 20** | **5 / 20** | **10 / 20** | **20 / 20** | **5 / 20** | **10 / 20** | **20 / 20** |
| Known Association | 42.22 ± 0.54 | 42.47 ± 0.32 | 42.14 ± 0.32 | 41.18 ± 0.78 | 41.74 ± 0.25 | 41.47 ± 0.48 | 40.76 ± 0.61 | 41.01 ± 0.50 | 40.74 ± 0.64 |
| Few labeled samples | 37.19 ± 0.76 | 36.61 ± 1.61 | 37.37 ± 1.53 | 35.23 ± 0.45 | 33.43 ± 1.04 | 34.24 ± 0.81 | 31.96 ± 1.04 | 29.89 ± 1.42 | 29.87 ± 1.09 |

Table 7: Test accuracy of FEROMA on CIFAR-10 under known test-time model association and automatic detection with few labeled samples (20 per class). Results are reported across low, medium, and high non-IID levels, with varying numbers of drifting clients.

Zhu et al., 2019; Zhao et al., 2020; Yin et al., 2021)). Accordingly, we treat profile transmission as an additional leakage channel and explicitly constrain it via a multi-layer protection mechanism: (i) structural compression (low-dimensional, many-to-one profiles), (ii) intentional stochasticity in extraction to reduce linkability across rounds, and (iii) a formal $(\varepsilon, \delta)$-DP guarantee at the *sample* level for each released profile.

For this reason, it is necessary to satisfy requirement R4, which provides the formal DP guarantee for profile release. Requirement R4 (see subsection 3.1) endows the Distribution-Profile Extractor $\phi_\psi : \mathbb{R}^{v^{(k)} \times z} \to \mathbb{R}^d$ with $(\varepsilon, \delta)$-differential privacy at the *sample* level: for any neighbouring datasets $(x, y)$ and $(x', y')$ that differ in one example and every measurable $\mathcal{S} \subseteq \mathbb{R}^d$,

$$\Pr[\phi_\psi(x, y) \in \mathcal{S}] \leq e^\varepsilon \Pr[\phi_\psi(x', y') \in \mathcal{S}] + \delta.$$

First, $\phi_\psi$ is intentionally *many-to-one* and low-dimensional: it aggregates statistics (e.g., moments in latent space) into a vector in $\mathbb{R}^d$ with $d \ll$ the input data dimension. As a consequence, infinitely many distinct datasets can induce the same profile, so the mapping is not uniquely invertible in general and does not preserve sample-level detail. This structural compression does not, by itself, constitute a formal DP guarantee, but it reduces the granularity of information exposed and makes exact reconstruction highly underdetermined without strong auxiliary assumptions. Second, we enforce $(\varepsilon, \delta)$-DP at the *sample* level for each released profile, which bounds the incremental leakage attributable to profile transmission (beyond what may already be exposed by model updates).

**Numerical experiments.** We evaluated the impact of enforcing DP on FEROMA under varying privacy budgets $\epsilon$, focusing on the privacy–utility trade-off in realistic FL scenarios. As shown in Tables 9 and 10, we tested FEROMA across all four types of statistical heterogeneity—$P(X)$, $P(Y)$, $P(Y|X)$, and $P(X|Y)$—at three levels of non-IID severity (low, medium, high). Overall results are summarized in Figure 8. We observe that FEROMA maintains robust accuracy for moderate budgets commonly targeted in practice ($5 \leq \epsilon \leq 10$), achieving performance comparable to the non-private (No-DP) baseline. In contrast, for very strict budgets ($\epsilon \in \{1, 2\}$), the injected noise is intentionally

| Non-IID Level | Low | | | Medium | | | High | | |
|---|---|---|---|---|---|---|---|---|---|
| **# Drifting** | **5 / 20** | **10 / 20** | **20 / 20** | **5 / 20** | **10 / 20** | **20 / 20** | **5 / 20** | **10 / 20** | **20 / 20** |
| Known Association | 38.88 ± 0.28 | 39.62 ± 0.50 | 39.36 ± 0.30 | 35.63 ± 0.30 | 35.71 ± 0.29 | 35.72 ± 0.08 | 31.75 ± 0.14 | 32.46 ± 0.27 | 32.15 ± 0.25 |
| Few labeled samples | 30.42 ± 0.50 | 29.36 ± 1.08 | 30.05 ± 1.06 | 22.86 ± 0.71 | 21.88 ± 0.85 | 23.56 ± 1.58 | 16.73 ± 2.15 | 13.34 ± 1.99 | 15.98 ± 1.63 |

Table 8: Test accuracy of FEROMA on CIFAR-100 under known test-time model association and automatic detection with few labeled samples (20 per class). Results are reported across low, medium, and high non-IID levels, with varying numbers of drifting clients.

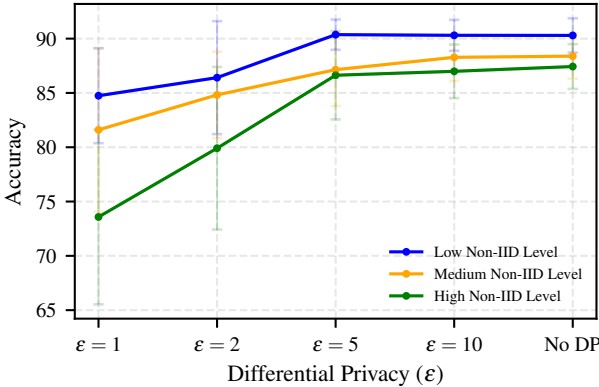

Figure 8: **Average test accuracy of FEROMA across different privacy budgets ($\epsilon$) and non-IID types.** Results are aggregated over $P(X)$, $P(Y)$, $P(Y|X)$, and $P(X|Y)$ scenarios.

large and we observe accuracy drops, especially under high non-IID severity—a behavior that is expected under strong privacy constraints. These results indicate that DP-protected distribution profiles can provide formal privacy guarantees with minimal utility loss at moderate $\epsilon$, supporting deployments subject to regulatory constraints such as GDPR (Regulation, 2016) or HIPAA (U.S. Congress, 1996).

| Non-IID Type | $P(X)$ | | | $P(Y)$ | | |
|---|---|---|---|---|---|---|
| **Non-IID Level** | **Low** | **Medium** | **High** | **Low** | **Medium** | **High** |
| $\epsilon = 1$ | 77.39 ± 6.23 | 77.78 ± 11.08 | 57.83 ± 13.07 | 98.48 ± 0.73 | 96.94 ± 3.06 | 97.72 ± 1.99 |
| $\epsilon = 2$ | 86.64 ± 2.10 | 83.32 ± 5.54 | 69.88 ± 11.25 | 98.25 ± 1.14 | 96.98 ± 2.98 | 97.21 ± 2.99 |
| $\epsilon = 5$ | 89.00 ± 0.38 | 85.15 ± 4.56 | 84.71 ± 1.71 | 98.69 ± 0.39 | 97.43 ± 2.85 | 97.21 ± 2.93 |
| $\epsilon = 10$ | 88.72 ± 0.50 | 86.46 ± 2.45 | 85.69 ± 0.43 | 98.67 ± 0.44 | 97.86 ± 2.02 | 97.29 ± 3.05 |
| No DP | 88.67 ± 0.30 | 86.50 ± 2.48 | 85.81 ± 0.36 | 98.66 ± 0.44 | 97.94 ± 2.01 | 97.24 ± 2.96 |

Table 9: Test accuracy of FEROMA under different privacy budgets $\epsilon$ for non-IID types based on feature distribution skew $P(X)$ and label distribution skew $P(Y)$ on the MNIST dataset. Results are reported for low, medium, and high non-IID levels.

| Non-IID Type | $P(Y|X)$ | | | $P(X|Y)$ | | |
|---|---|---|---|---|---|---|
| **Non-IID Level** | **Low** | **Medium** | **High** | **Low** | **Medium** | **High** |
| $\epsilon = 1$ | 79.15 ± 3.25 | 74.23 ± 3.20 | 71.63 ± 4.94 | 83.95 ± 5.14 | 77.45 ± 9.22 | 67.14 ± 7.73 |
| $\epsilon = 2$ | 82.92 ± 3.44 | 77.04 ± 2.77 | 75.10 ± 4.50 | 77.82 ± 9.52 | 81.93 ± 4.03 | 77.44 ± 8.28 |
| $\epsilon = 5$ | 84.50 ± 2.22 | 78.23 ± 3.43 | 78.32 ± 2.08 | 89.28 ± 1.59 | 87.75 ± 1.99 | 86.28 ± 7.10 |
| $\epsilon = 10$ | 84.56 ± 2.63 | 79.16 ± 2.74 | 78.02 ± 2.86 | 89.27 ± 0.87 | 89.63 ± 1.13 | 86.96 ± 2.58 |
| No DP | 85.04 ± 2.44 | 80.09 ± 2.14 | 78.38 ± 2.62 | 88.81 ± 1.94 | 89.02 ± 1.59 | 88.29 ± 1.12 |

Table 10: Test accuracy of FEROMA under different privacy budgets $\epsilon$ for non-IID types based on concept shifts $P(Y|X)$ and $P(X|Y)$ on the MNIST dataset. Results are reported for low, medium, and high non-IID levels.

## G QUANTIFICATION OF COSTS IN FEROMA

In this section, we quantify the communication and computation overhead introduced by FEROMA and compare it explicitly with standard FedAvg. While FEROMA introduces additional operations for extracting and transmitting distribution profiles, these costs are intentionally designed to be lightweight and independent of model size.

**Communication cost.** On the communication side, FEROMA transmits exactly the same model parameters as FedAvg, plus a single low-dimensional distribution profile per client per round. The size of this profile depends only on the descriptor dimension and the number of labels, and is independent of the underlying model size. As a result, the relative communication overhead decreases rapidly as models scale. Table 11 reports the per-client upload/download size and corresponding transmission time under a bandwidth of 20 MB/s for representative models of increasing scale. Across all settings, the additional communication overhead introduced by FEROMA remains below 0.4%, and becomes negligible (below 0.01%) for large models.

| Dataset / Model | Method | Upload / Download | Time (20MB/s) | Overhead |
|---|---|---|---|---|
| MNIST – LeNet5 (62K) | FedAvg | 248.0 KB | 0.496 s | – |
| | FEROMA | 248.88 KB | 0.498 s | +0.35% |
| CIFAR-100 – ResNet-9 (1.65M) | FedAvg | 6.60 MB | 13.200 s | – |
| | FEROMA | 6.61 MB | 13.216 s | +0.15% |
| CIFAR-100 – ViT-B/16 ($\sim$86M) | FedAvg | 344.0 MB | 688.000 s | – |
| | FEROMA | 344.01 MB | 688.016 s | $< 0.01\%$ |

Table 11: Per-round communication cost per client for FedAvg and FEROMA under different model scales.

**Computation cost.** On the computation side, the only non-trivial additional operation introduced by FEROMA is the computation of a rank-1 PCA over $s^{\text{PCA}} = 200$ synthetic latent vectors of dimension $z$ during distribution profile extraction (Step S2). This operation costs at most approximately $8.2 \times 10^7$ FLOPs for typical latent dimensions ($z \in [128, 2048]$). For comparison, even a single forward pass of the smallest MNIST model already exceeds $6.5 \times 10^8$ FLOPs, while training the largest CIFAR-100 model requires over $6 \times 10^{11}$ FLOPs per epoch. The remaining operations in the DPE, including moment computation and noise addition, are linear in the descriptor dimension and therefore negligible.

Overall, FEROMA introduces well below 1% additional computation relative to standard local training, while providing substantial robustness benefits under distribution shift and drift.

## H CONVERGENCE ANALYSIS ON STATIONARY WINDOWS

This section addresses the convergence behavior of FEROMA. Under unbounded and potentially adversarial concept drift, classical global convergence guarantees are in general unattainable, since the underlying objectives may change arbitrarily over time. We therefore analyze FEROMA on a *stationary time window* between two distribution-shift events, where local data distributions are approximately constant. This "piecewise-stationary" viewpoint is standard when formal guarantees under perpetual drift are not meaningful. For this analysis, we adopt the common idealized setting of a fixed set of clients with full participation in every round, although FEROMA supports partial participation in practice.

**Stationary window.** Fix a contiguous interval of communication rounds $\mathcal{T} = \{t_0, \ldots, t_0 + T - 1\}$ during which each client distribution is stationary:

$$P_t^{(k)} = P^{(k)} \qquad \forall t \in \mathcal{T}, \ k \in \{1, \ldots, K\}.$$

For each client $k$, define the local objective

$$f_k(\theta) := \mathbb{E}_{(x,y) \sim P^{(k)}} \big[ \ell(\theta; x, y) \big],$$

and the weighted global objective $F(\theta) := \sum_{k=1}^{K} p_k f_k(\theta)$, where $p_k > 0$ and $\sum_k p_k = 1$. Within $\mathcal{T}$, the local training step in FEROMA corresponds to standard SGD on $f_k$, with a fixed number $H \geq 1$ of local steps per communication round.

We adopt standard conditions from local-SGD / decentralized SGD analyses (e.g., (Koloskova et al., 2020; Stich, 2019; Gao et al., 2021; Pu et al., 2020)).

**Assumption A1 (Smoothness and bounded variance).** For each $k$, $f_k$ is $L$-smooth and lower bounded by $f_k^\star$. Stochastic gradients $g_{t,h}^{(k)}$ at local step $h \in \{0, \ldots, H-1\}$ are unbiased and have bounded variance:

$$\mathbb{E}\left[g_{t,h}^{(k)} \mid \theta_{t,h}^{(k)}\right] = \nabla f_k(\theta_{t,h}^{(k)}), \qquad \mathbb{E}\left[\|g_{t,h}^{(k)} - \nabla f_k(\theta_{t,h}^{(k)})\|^2\right] \leq \sigma^2,$$

for all $t \in \mathcal{T}$ and $k \in \{1, \ldots, K\}$.

**Assumption A2 (Profile-induced mixing matrices).** At each round $t \in \mathcal{T}$, FEROMA induces a mixing matrix $W_t \in \mathbb{R}^{K \times K}$ with entries $(W_t)_{kj} := \bar{w}_t^{(k,j)}$, where $\bar{w}_t^{(k,j)}$ denotes the (possibly thresholded and renormalized) association weight used in Eq. (5). We assume:

1. *Symmetry and double stochasticity:* $W_t = W_t^\top$, $\quad W_t \mathbf{1} = \mathbf{1}$, $\quad \mathbf{1}^\top W_t = \mathbf{1}^\top$.

2. *Expected consensus rate (time-varying topology):* there exist constants $\beta \in (0, 1]$ and an integer block length $B \geq 1$ such that, for all $X \in \mathbb{R}^{d \times K}$ and all integers $\ell$,

$$\mathbb{E}\left[\|XW_{\ell,B} - \bar{X}\|_F^2\right] \leq (1 - \beta)\|X - \bar{X}\|_F^2,$$

where $W_{\ell,B} := W_{(\ell+1)B-1} \cdots W_{\ell B}$, $\bar{X} := X \frac{1}{K} \mathbf{1}\mathbf{1}^\top$, and the expectation is over the randomness of $\{W_{\ell B}, \ldots, W_{(\ell+1)B-1}\}$.

Assumption A2 is standard in decentralized SGD with changing topology and implies that repeated mixing contracts disagreement towards the uniform average with factor $(1 - \beta)$ (Koloskova et al., 2020; Pu et al., 2020). In FEROMA, such mixing arises by building similarity weights from distribution profiles and applying a normalization step that yields (approximately) doubly-stochastic matrices. Moreover, given requirements (R1) and (R3), profile distances track underlying distribution distances up to small distortion and the injected profile noise has bounded covariance; under a stationary window, this makes it natural to assume that the resulting similarity-based matching can be chosen so that the induced $\{W_t\}$ satisfy the expected consensus condition in Assumption A2 (cf. Koloskova et al. (2020)).

**From FEROMA to decentralized local-SGD.** Let $\Theta_t := [\theta_t^{(1)}, \ldots, \theta_t^{(K)}] \in \mathbb{R}^{d \times K}$ denote the matrix of client parameters at the beginning of round $t$. Over a stationary window, one round of FEROMA can be written in the generic decentralized local-SGD form

$$\Theta_{t+1} = \Theta_t W_t - \eta_t G_t, \tag{3}$$

where $\Theta_t W_t$ is the profile-based aggregation (Eq. (5)) and $G_t$ stacks the stochastic gradients accumulated during the $H$ local SGD steps starting from the mixed parameters. Concretely, the $k$-th column of $\Theta_t W_t$ equals $\sum_j \bar{w}_t^{(k,j)} \theta_t^{(j)}$, matching Eq. (5). This recursion coincides with decentralized/local-SGD with time-varying mixing matrices studied in Koloskova et al. (2020), specialized to a star-shaped implementation where the server computes $W_t$ and broadcasts the mixed models, but the update is applied client-side.

**Implication (convergence within $\mathcal{T}$).** Under Assumptions A1–A2 and on a stationary window $\mathcal{T}$, FEROMA is a special case of decentralized local-SGD with changing topology. Therefore, existing results for Eq. equation 3 apply directly, yielding convergence guarantees to a stationary point of $F$ (for non-convex objectives) with an optimization error term and an additional consensus term that depends on the mixing quality (through $\beta$), the number of local steps $H$, and gradient noise $\sigma^2$ (see, e.g., (Koloskova et al., 2020)). Intuitively, better profile-induced mixing (larger $\beta$) reduces client disagreement and improves the rate, whereas very strict privacy/noise settings that perturb profile similarities can degrade $\beta$ and thus slow convergence. Formally characterizing how DPE hyperparameters affect the spectral gap of $W_t$ is an interesting direction for future work.

# I    ADDITIONAL EXPERIMENT RESULTS

## I.1    GENERATING DRIFTING DATASETS WITH ANDA, CHEXPERT AND OFFICE-HOME

For datasets MNIST, FMNIST, CIFAR-10, and CIFAR-100, we generate drifting datasets across clients with four non-IID types using *ANDA*. *ANDA* (**A N**on-IID **D**ata generator supporting **Any**

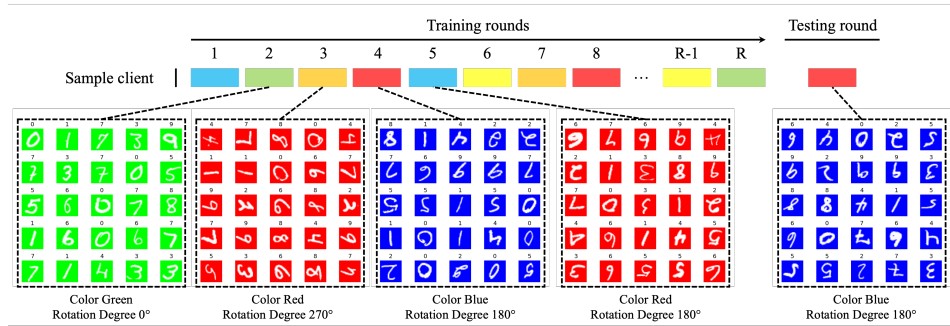

Figure 9: **Distribution drifting in $P(X)$ with MNIST.**

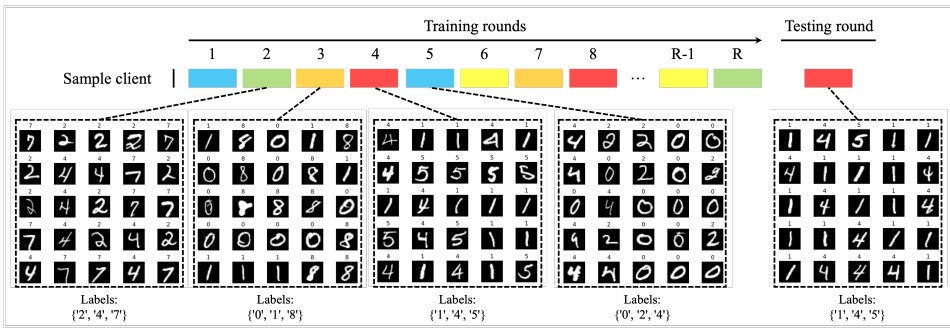

Figure 10: **Distribution drifting in $P(Y)$ with MNIST.**

kind) is a toolkit designed to create non-IID datasets for reproducible FL experiments. It supports datasets MNIST, EMNIST, FMNIST, CIFAR-10, and CIFAR-100, and facilitates five types of data distribution shifts:

- *Feature distribution skew (covariate shift)*: Marginal distributions $P(X)$ vary across clients.
- *Label distribution skew (prior probability shift)*: Marginal distributions $P(Y)$ vary across clients.
- *Concept shift (same X, different Y)*: Conditional distributions $P(Y|X)$ vary across clients.
- *Concept shift (same Y, different X)*: Conditional distributions $P(X|Y)$ vary across clients.
- *Quantity shift*: The amount of data vary across clients.

*ANDA* enables the generation of only shifting datasets or shifting with drifting datasets, allowing clients to possess datasets with varying distributions across training rounds or test round.

*ANDA* applies commonly used approaches (Sattler et al., 2021; Deng et al., 2020b; Guo et al., 2024; Jothimurugesan et al., 2023; Ghosh et al., 2020; Marfoq et al., 2021; Long et al., 2023; T. Dinh et al., 2020) to generate drifting heterogeneous datasets:

- $P(X)$ (Figure 9): Each image undergoes one of three color transformations (blue, green, or red) and one of four rotations ($0°$, $90°$, $180°$, or $270°$), with distinct distributions applied to each subset.
- $P(Y)$ (Figure 10): Each client receives data only from certain classes. For example, in the MNIST dataset, the dataset in training round 2 only has digits 2, 4, and 7, while the dataset in training round 3 has images of digits 0, 1, and 8.
- $P(Y|X)$ (Figure 11): Given identical feature distributions (e.g., image pixels), labels differ between clients. For instance, in the MNIST dataset, the dataset in training round 2 labels the digit '8' as '8', '1' as '5', and '5' as '1', whereas the dataset in training round 3 labels '8' as '1', '1' as '5', and '5' as '8'.
- $P(X|Y)$ (Figure 12): For the same label, different features are applied. For example, in the MNIST dataset, the dataset in training round 2 applies a blue hue to images labeled '0', while the dataset in training round 3 applies a red hue to the same label.

For the real-world dataset CheXpert, we do not apply any image augmentation or label modification in order to preserve the correctness and integrity of the data. To simulate different levels of data

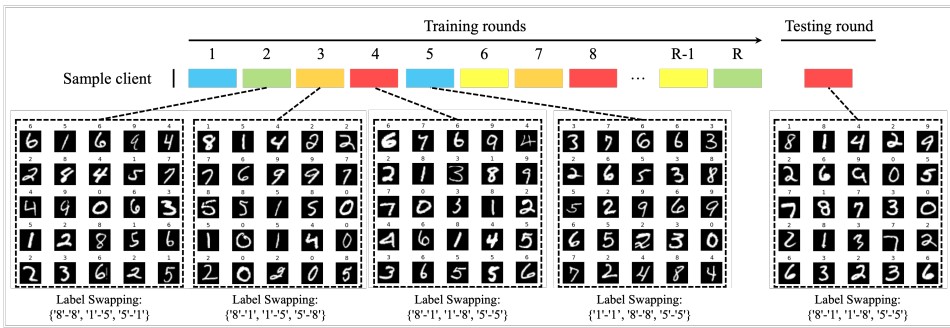

Figure 11: **Distribution drifting in** $P(Y|X)$ **with MNIST.**

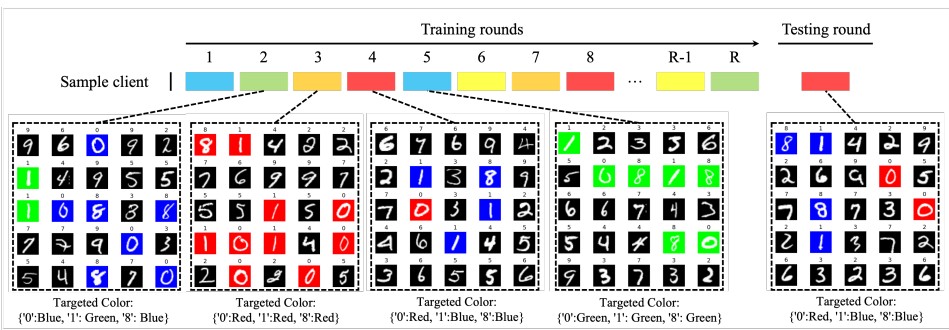

Figure 12: **Distribution drifting in** $P(X|Y)$ **with MNIST.**

heterogeneity, we partition the dataset into multiple distributions based on three metadata attributes available in the dataset: *ViewPosition*, *Age*, and *Sex*. Specifically, the dataset is split according to the following criteria for each non-IID level used in our experiments:

- **Low Non-IID Level**: 2 distributions based on *ViewPosition* (Frontal / Lateral).
- **Medium Non-IID Level**: 4 distributions based on *ViewPosition* (Frontal / Lateral) and *Age* ($\geq$50 / <50).
- **High Non-IID Level**: 8 distributions based on *ViewPosition* (Frontal / Lateral), *Age* ($\geq$50 / <50), and *Sex* (Male / Female).

For the real-world dataset Office-Home, we do not apply any image augmentation or label modification; instead, we directly use the dataset in its original form. For our experiments, we restrict the dataset to images with labels from 0 to 9. To model different levels of data heterogeneity, we leverage the four inherent domains of the dataset: *Art*, *Clipart*, *Product*, and *Real-World*. The domain shift across these categories naturally induces heterogeneity in the data distribution.

### I.2   SCALING THE DRIFTING FREQUENCY, NON-IID TYPES, AND NON-IID LEVELS

We evaluate the effects of data heterogeneity using four non-IID dataset types across 20 clients, generated with *ANDA* at three distinct levels of heterogeneity and three levels of drifting frequency.

**Drifting frequency.**    Client's local data change every round, and we scale the local data distribution drifting frequency. At level one, each client's local training data distribution drifts every four rounds. At level two, each client's local training data distribution drifts every two rounds. At level three, each client's local training data distribution drifts every round. The test time distributions are always under drift.

**Non-IID Types and Levels.**    We evaluate FEROMA under four common types of non-IID data distributions and define three levels of heterogeneity (low, medium, and high). The full configurations are summarized in Table 12 and detailed as follows:

- **Feature distribution skew ($P(X)$):** Each client is assigned a unique combination of data augmentations (e.g., rotation and color transformation), applied consistently to all local samples. The non-IID level controls the number of available augmentation choices. For example, at the Level Medium, a client may apply one rotation from $\{0°, 180°\}$ and one color from {Red, Green, Blue} to all images. At Level Low, the "Original" color indicates no color transformation.
- **Label distribution skew ($P(Y)$):** Each client retains samples from a subset of classes. For MNIST, FMNIST, and CIFAR-10, each client holds data from 2 classes; for CIFAR-100, from 20 classes. We define a bank of class subsets (e.g., 4 banks of {[0,4], [1,9], [3,5], [6,9]} at Level Low), and clients randomly sample from these banks. Increasing the number of banks increases the heterogeneity level.
- **Conditional label skew ($P(Y|X)$):** Clients receive relabeled versions of a subset of classes via label permutation. For example, at the medium level, a class pool $\{2, 3, 5, 8\}$ may be randomly permuted to $\{5, 8, 3, 2\}$, mapping images originally labeled as '2' to label '5'. The number of classes in the permutation pool increases with the non-IID level.
- **Conditional feature skew ($P(X|Y)$):** Clients apply different augmentations to the same class. For instance, Client A may apply a $0°$ rotation to class '5', while Client B applies a $180°$ rotation. Augmentations are limited to rotations ($\{0°, 90°, 180°, 270°\}$) and colors ({Red, Green, Blue}). For MNIST, FMNIST, and CIFAR-10, this applies to 8 classes; for CIFAR-100, to 80 classes. The level of heterogeneity is scaled in the same way as for $P(Y)$, using banks of class-specific transformations.

| non-IID type | $P(X)$ | $P(Y)$ | $P(Y|X)$ | $P(X|Y)$ |
|---|---|---|---|---|
| Low | Rotation $\{0°, 90°, 180°, 270°\}$, Color {Original} | #Bank = 4 | #Swapped Class = 3 (*40*) | #Bank = 4 |
| Medium | Rotation $\{0°, 180°\}$, Color {Red, Blue, Green} | #Bank = 6 | #Swapped Class = 4 (*60*) | #Bank = 6 |
| High | Rotation $\{0°, 90°, 180°, 270°\}$, Color {Red, Blue, Green} | #Bank = 8 | #Swapped Class = 5 (*80*) | #Bank = 8 |

Table 12: **Summary of non-IID data heterogeneity configurations across levels.** Each type of heterogeneity corresponds to a specific distribution shift. For $P(Y|X)$, the numbers of swapped class for CIFAR-100 are *40*, *60*, and *80*, accordingly.

We dynamically scale the size of training dataset based on the factor of drifting frequency and number of clients to ensure each subset preserves enough samples to train the local model. Tables 13 to Table 28 present detailed results of FEROMA and all baseline methods across various drifting frequency, non-IID types and levels. For experiments on $P(Y|X)$ concept shift, we keep the test time distribution the same as the last training round (see Appendix E for more details).

| Non-IID Level | Low | | | Medium | | | High | | |
|---|---|---|---|---|---|---|---|---|---|
| # Drifting | **5 / 20** | **10 / 20** | **20 / 20** | **5 / 20** | **10 / 20** | **20 / 20** | **5 / 20** | **10 / 20** | **20 / 20** |
| FedAvg | 66.55 ± 1.23 | 67.92 ± 1.51 | 72.12 ± 3.45 | 73.19 ± 4.03 | 73.66 ± 2.29 | 73.25 ± 1.84 | 54.77 ± 1.81 | 54.32 ± 2.68 | 58.54 ± 5.87 |
| FedRC | 25.36 ± 1.71 | 25.79 ± 1.39 | 26.42 ± 1.33 | 11.35 ± 0.00 | 11.35 ± 0.00 | 11.35 ± 0.00 | 11.35 ± 0.00 | 11.35 ± 0.00 | 11.35 ± 0.00 |
| FedEM | 25.36 ± 1.71 | 25.79 ± 1.38 | 26.42 ± 1.33 | 11.35 ± 0.00 | 11.35 ± 0.00 | 11.35 ± 0.00 | 11.35 ± 0.00 | 11.35 ± 0.00 | 11.35 ± 0.00 |
| FeSEM | 60.88 ± 4.47 | 67.26 ± 3.33 | 70.39 ± 4.80 | 67.78 ± 6.39 | 72.97 ± 4.69 | 71.52 ± 4.04 | 47.95 ± 3.42 | 49.26 ± 0.99 | 49.89 ± 2.91 |
| CFL | 74.28 ± 1.51 | 75.65 ± 1.31 | 78.69 ± 2.55 | 82.95 ± 1.32 | 80.94 ± 2.30 | 80.23 ± 3.33 | 68.03 ± 2.18 | 68.78 ± 1.94 | 70.65 ± 4.20 |
| IFCA | 38.68 ± 8.39 | 37.48 ± 11.31 | 42.53 ± 6.00 | 40.41 ± 15.49 | 39.49 ± 4.71 | 38.55 ± 5.59 | 30.00 ± 11.54 | 31.27 ± 15.08 | 22.01 ± 14.43 |
| pFedMe | 35.20 ± 2.54 | 43.45 ± 8.54 | 47.72 ± 4.03 | 43.29 ± 4.98 | 46.52 ± 5.68 | 50.22 ± 3.47 | 30.31 ± 5.43 | 34.66 ± 9.46 | 34.76 ± 3.16 |
| APFL | 55.29 ± 1.22 | 61.87 ± 5.42 | 65.83 ± 3.98 | 70.88 ± 5.59 | 71.42 ± 3.50 | 72.26 ± 2.80 | 55.03 ± 3.60 | 59.80 ± 5.79 | 61.35 ± 5.22 |
| FedDrift | 41.31 ± 2.08 | 46.08 ± 8.46 | 46.41 ± 4.26 | 50.94 ± 10.99 | 54.49 ± 8.04 | 59.06 ± 4.54 | 39.63 ± 5.76 | 39.47 ± 8.12 | 37.40 ± 5.80 |
| ATP | 78.74 ± 2.69 | 82.43 ± 0.72 | 84.03 ± 1.93 | 82.56 ± 1.50 | 76.59 ± 2.64 | 80.49 ± 1.75 | 60.55 ± 14.14 | 39.60 ± 19.31 | 51.14 ± 14.01 |
| FEROMA | **88.90 ± 0.33** | **88.67 ± 0.30** | **89.52 ± 0.42** | **87.14 ± 2.29** | **86.50 ± 2.48** | **86.08 ± 3.36** | **86.40 ± 0.19** | **85.81 ± 0.36** | **85.74 ± 1.17** |

Table 13: Performance comparison across three different non-IID Levels of $P(X)$ and three distribution drifting levels on the MNIST dataset.

### I.3 SCALING THE NUMBER OF CLIENTS

We evaluate the scalability and efficiency of the proposed approach across varying numbers of clients on MNIST dataset, with the non-IID Level Medium (see subsection I.2 for more details). To ensure each client has sufficient local training data, we adjust the dataset size based on the total number of clients. For 10 clients, we reduce the dataset to half the size used in the main experiments

| Non-IID Level | Low | | | Medium | | | High | | |
|---|---|---|---|---|---|---|---|---|---|
| # Drifting | 5 / 20 | 10 / 20 | 20 / 20 | 5 / 20 | 10 / 20 | 20 / 20 | 5 / 20 | 10 / 20 | 20 / 20 |
| FedAvg | 75.12 ± 12.47 | 83.77 ± 8.76 | 94.58 ± 1.61 | 74.46 ± 5.80 | 82.83 ± 9.38 | 90.77 ± 1.41 | 70.38 ± 11.91 | 74.70 ± 11.61 | 89.44 ± 5.75 |
| FedRC | 68.72 ± 12.82 | 76.24 ± 7.75 | 81.37 ± 8.20 | 56.44 ± 6.80 | 68.54 ± 12.47 | 72.72 ± 6.83 | 47.94 ± 13.45 | 56.24 ± 14.30 | 79.22 ± 3.89 |
| FedEM | 66.81 ± 14.63 | 78.29 ± 4.30 | 83.40 ± 2.76 | 48.68 ± 8.21 | 62.98 ± 6.57 | 73.85 ± 8.76 | 44.29 ± 8.81 | 59.28 ± 12.07 | 78.05 ± 6.27 |
| FeSEM | 80.00 ± 7.15 | 82.17 ± 4.80 | 86.79 ± 3.51 | 64.05 ± 5.32 | 75.46 ± 3.72 | 75.44 ± 13.45 | 54.07 ± 10.98 | 63.39 ± 11.30 | 81.49 ± 3.41 |
| CFL | 83.78 ± 7.14 | 83.78 ± 7.20 | 93.92 ± 2.13 | 77.18 ± 8.70 | 86.65 ± 5.89 | 87.97 ± 5.62 | 74.45 ± 8.32 | 83.15 ± 3.74 | 90.35 ± 2.31 |
| IFCA | 42.53 ± 12.68 | 47.09 ± 12.40 | 47.52 ± 22.21 | 39.16 ± 15.13 | 34.78 ± 9.81 | 23.50 ± 5.16 | 33.89 ± 9.71 | 22.06 ± 2.74 | 27.45 ± 5.85 |
| pFedMe | 45.59 ± 11.11 | 47.25 ± 8.41 | 42.03 ± 17.83 | 38.94 ± 13.87 | 37.17 ± 11.39 | 25.50 ± 5.17 | 30.80 ± 5.23 | 28.72 ± 7.36 | 29.50 ± 8.40 |
| APFL | 61.16 ± 11.47 | 63.98 ± 7.13 | 69.99 ± 10.55 | 51.64 ± 9.55 | 56.67 ± 6.98 | 54.33 ± 6.76 | 49.12 ± 6.00 | 47.60 ± 8.29 | 58.95 ± 9.33 |
| FedDrift | 54.08 ± 12.62 | 58.21 ± 8.00 | 48.36 ± 19.20 | 43.86 ± 12.37 | 47.16 ± 11.85 | 33.75 ± 5.75 | 36.44 ± 6.01 | 36.60 ± 9.05 | 42.36 ± 6.93 |
| ATP | 42.75 ± 25.29 | 76.00 ± 20.47 | 95.62 ± 2.20 | 38.30 ± 26.19 | 87.20 ± 8.07 | 84.64 ± 13.35 | 36.22 ± 21.01 | 83.91 ± 6.24 | 90.27 ± 6.21 |
| FEROMA | **96.50 ± 3.23** | **98.66 ± 0.44** | **99.36 ± 0.29** | **97.12 ± 1.75** | **97.94 ± 2.01** | **98.67 ± 1.08** | **96.91 ± 1.89** | **97.24 ± 2.96** | **99.32 ± 0.30** |

Table 14: Performance comparison across three different non-IID Levels of $P(Y)$ and three distribution drifting levels on the MNIST dataset.

| Non-IID Level | Low | | | Medium | | | High | | |
|---|---|---|---|---|---|---|---|---|---|
| # Drifting | 5 / 20 | 10 / 20 | 20 / 20 | 5 / 20 | 10 / 20 | 20 / 20 | 5 / 20 | 10 / 20 | 20 / 20 |
| FedAvg | 73.51 ± 1.57 | 72.91 ± 0.81 | 72.93 ± 0.95 | 64.88 ± 1.74 | 64.96 ± 2.08 | 64.24 ± 2.01 | 57.10 ± 2.07 | 57.19 ± 2.04 | 56.38 ± 1.82 |
| FedRC | 29.62 ± 4.95 | 28.79 ± 5.42 | 28.33 ± 8.02 | 13.97 ± 3.91 | 12.91 ± 2.29 | 14.78 ± 2.95 | 12.96 ± 3.62 | 12.12 ± 2.26 | 11.62 ± 1.95 |
| FedEM | 29.61 ± 4.94 | 28.79 ± 5.42 | 28.34 ± 8.03 | 13.96 ± 3.90 | 12.91 ± 2.29 | 14.77 ± 2.94 | 12.96 ± 3.61 | 12.12 ± 2.25 | 11.62 ± 1.94 |
| FeSEM | 78.69 ± 1.37 | 78.98 ± 0.74 | 79.27 ± 0.41 | 78.58 ± 1.34 | 75.47 ± 1.41 | 77.39 ± 1.88 | 75.80 ± 1.58 | 73.68 ± 1.93 | 73.66 ± 3.38 |
| CFL | 75.41 ± 0.94 | 75.01 ± 0.79 | 75.53 ± 0.62 | 66.82 ± 1.56 | 67.26 ± 1.22 | 66.31 ± 2.01 | 58.88 ± 1.40 | 59.95 ± 1.66 | 58.11 ± 1.78 |
| IFCA | 76.89 ± 2.60 | 77.15 ± 2.31 | 78.31 ± 3.64 | 71.64 ± 2.80 | 69.70 ± 4.04 | 69.28 ± 2.08 | 63.46 ± 3.02 | 60.51 ± 1.13 | 58.44 ± 2.45 |
| pFedMe | 91.42 ± 0.38 | 91.55 ± 0.42 | 91.28 ± 0.77 | 91.28 ± 0.37 | 91.01 ± 0.41 | 90.37 ± 0.68 | 90.99 ± 0.35 | 90.56 ± 0.41 | 89.32 ± 0.66 |
| APFL | **92.70 ± 0.56** | **92.58 ± 0.53** | **92.54 ± 0.68** | **92.03 ± 0.62** | **91.64 ± 0.69** | **91.22 ± 0.68** | **91.05 ± 0.49** | **90.80 ± 0.58** | **89.87 ± 0.85** |
| FedDrift | 91.20 ± 2.60 | 92.55 ± 0.42 | 92.50 ± 0.27 | 89.71 ± 3.43 | 89.71 ± 2.82 | 90.54 ± 1.03 | 87.51 ± 2.09 | 84.43 ± 2.29 | 83.87 ± 3.09 |
| ATP | 76.94 ± 0.72 | 75.79 ± 0.63 | 77.10 ± 1.13 | 68.32 ± 1.29 | 67.73 ± 2.30 | 68.14 ± 1.97 | 60.37 ± 1.45 | 60.14 ± 1.27 | 60.23 ± 1.01 |
| FEROMA | 89.58 ± 0.50 | 89.48 ± 0.45 | 89.80 ± 0.75 | 88.90 ± 0.53 | 88.87 ± 0.68 | 89.01 ± 0.64 | 88.15 ± 0.66 | 88.06 ± 0.51 | 88.41 ± 0.68 |

Table 15: Performance comparison across three different non-IID Levels of $P(Y|X)$ and three distribution drifting levels on the MNIST dataset.

| Non-IID Level | Low | | | Medium | | | High | | |
|---|---|---|---|---|---|---|---|---|---|
| # Drifting | 5 / 20 | 10 / 20 | 20 / 20 | 5 / 20 | 10 / 20 | 20 / 20 | 5 / 20 | 10 / 20 | 20 / 20 |
| FedAvg | 73.39 ± 9.75 | 81.57 ± 2.86 | 77.66 ± 3.30 | 73.28 ± 4.02 | 73.09 ± 4.06 | 74.97 ± 4.10 | 71.56 ± 5.11 | 74.30 ± 5.09 | 69.94 ± 9.80 |
| FedRC | 38.39 ± 13.46 | 34.60 ± 11.07 | 42.95 ± 15.57 | 13.55 ± 3.01 | 13.71 ± 3.44 | 14.24 ± 3.57 | 11.95 ± 1.19 | 12.03 ± 0.84 | 12.36 ± 2.03 |
| FedEM | 39.48 ± 16.08 | 40.06 ± 12.86 | 39.86 ± 19.18 | 12.87 ± 3.04 | 15.95 ± 7.59 | 14.70 ± 4.40 | 12.31 ± 1.92 | 12.01 ± 1.33 | 12.55 ± 2.39 |
| FeSEM | 70.07 ± 7.57 | 74.58 ± 5.34 | 74.80 ± 4.84 | 67.89 ± 8.47 | 62.18 ± 3.89 | 63.74 ± 10.06 | 62.85 ± 4.13 | 59.89 ± 7.05 | 56.57 ± 6.46 |
| CFL | 82.84 ± 4.32 | 78.01 ± 2.92 | 81.78 ± 4.99 | 79.30 ± 3.25 | 78.22 ± 3.23 | 77.96 ± 3.70 | 76.52 ± 1.45 | 77.83 ± 5.80 | 78.94 ± 2.82 |
| IFCA | 45.78 ± 12.55 | 49.72 ± 13.79 | 35.88 ± 11.68 | 33.13 ± 10.31 | 37.07 ± 13.08 | 30.66 ± 3.80 | 34.99 ± 12.31 | 29.25 ± 6.99 | 25.62 ± 7.72 |
| pFedMe | 47.31 ± 9.19 | 54.89 ± 7.12 | 53.18 ± 7.55 | 39.63 ± 10.46 | 42.60 ± 6.62 | 48.26 ± 7.80 | 36.97 ± 7.00 | 37.18 ± 3.43 | 45.86 ± 14.27 |
| APFL | 66.94 ± 5.64 | 71.04 ± 2.56 | 76.07 ± 9.07 | 65.29 ± 9.73 | 67.60 ± 8.09 | 69.64 ± 4.79 | 62.99 ± 4.48 | 60.84 ± 3.22 | 66.99 ± 5.20 |
| FedDrift | 48.43 ± 12.31 | 55.56 ± 5.44 | 46.14 ± 10.31 | 47.79 ± 6.16 | 51.62 ± 5.61 | 43.21 ± 9.51 | 45.44 ± 8.37 | 50.35 ± 5.78 | 46.22 ± 5.39 |
| ATP | 84.74 ± 3.67 | 79.95 ± 6.33 | 78.96 ± 12.67 | 73.81 ± 10.86 | 78.94 ± 5.89 | 78.62 ± 10.91 | 79.43 ± 4.01 | 76.84 ± 10.65 | 79.88 ± 6.86 |
| FEROMA | **87.26 ± 4.80** | **88.81 ± 1.94** | **89.87 ± 2.09** | **86.99 ± 4.52** | **89.02 ± 1.59** | **90.11 ± 0.79** | **89.44 ± 1.17** | **88.29 ± 1.12** | **89.71 ± 0.60** |

Table 16: Performance comparison across three different non-IID Levels of $P(X|Y)$ and three distribution drifting levels on the MNIST dataset.

| Non-IID Level | Low | | | Medium | | | High | | |
|---|---|---|---|---|---|---|---|---|---|
| # Drifting | 5 / 20 | 10 / 20 | 20 / 20 | 5 / 20 | 10 / 20 | 20 / 20 | 5 / 20 | 10 / 20 | 20 / 20 |
| FedAvg | 36.45 ± 0.83 | 37.34 ± 0.55 | 38.14 ± 0.49 | 20.46 ± 5.80 | 21.56 ± 3.40 | 26.80 ± 2.84 | 17.45 ± 7.18 | 17.09 ± 5.73 | 20.92 ± 2.85 |
| FedRC | 25.88 ± 0.24 | 26.17 ± 0.61 | 26.34 ± 0.70 | 13.40 ± 0.84 | 12.30 ± 1.09 | 15.55 ± 2.24 | 10.94 ± 0.73 | 10.66 ± 0.31 | 11.54 ± 0.78 |
| FedEM | 25.88 ± 0.26 | 26.18 ± 0.60 | 26.39 ± 0.71 | 13.39 ± 0.85 | 12.30 ± 1.08 | 15.55 ± 2.24 | 10.94 ± 0.74 | 10.66 ± 0.31 | 11.54 ± 0.78 |
| FeSEM | 35.03 ± 0.66 | 35.38 ± 1.03 | 34.87 ± 0.96 | 21.44 ± 1.93 | 20.73 ± 0.95 | 22.76 ± 4.76 | 19.28 ± 2.32 | 19.26 ± 1.55 | 18.99 ± 1.43 |
| CFL | 37.54 ± 0.67 | 38.29 ± 0.76 | 39.15 ± 1.14 | 22.69 ± 3.00 | 25.26 ± 0.96 | 27.67 ± 2.62 | 21.11 ± 4.86 | 16.93 ± 1.98 | 22.78 ± 1.96 |
| IFCA | 31.08 ± 1.99 | 33.18 ± 3.00 | 33.75 ± 2.12 | 17.33 ± 1.54 | 20.28 ± 2.92 | 21.82 ± 3.60 | 18.83 ± 3.16 | 18.45 ± 1.62 | 18.37 ± 1.40 |
| pFedMe | 23.54 ± 0.48 | 26.43 ± 2.35 | 27.51 ± 0.63 | 16.94 ± 0.47 | 18.12 ± 0.98 | 18.66 ± 1.59 | 16.55 ± 1.77 | 17.41 ± 1.72 | 15.25 ± 0.35 |
| APFL | 27.88 ± 0.47 | 31.59 ± 2.43 | 34.07 ± 0.58 | 20.04 ± 1.10 | 22.61 ± 1.16 | 24.73 ± 2.41 | 18.80 ± 2.98 | 19.53 ± 2.49 | 21.10 ± 1.12 |
| FedDrift | 31.10 ± 3.38 | 32.12 ± 4.98 | 32.25 ± 4.15 | 17.31 ± 1.42 | 20.50 ± 1.97 | 21.10 ± 1.15 | 17.86 ± 1.98 | 18.65 ± 1.86 | 17.92 ± 1.92 |
| ATP | 33.53 ± 1.17 | 32.15 ± 3.73 | 31.46 ± 2.78 | 28.27 ± 1.66 | 29.93 ± 1.77 | 28.17 ± 5.07 | 24.22 ± 2.87 | 24.39 ± 0.49 | 25.13 ± 0.38 |
| FEROMA | **40.38 ± 0.26** | **39.11 ± 2.14** | **40.30 ± 0.51** | **31.81 ± 0.60** | **31.71 ± 2.98** | **33.67 ± 0.63** | **28.78 ± 2.95** | **28.84 ± 1.06** | **29.96 ± 1.72** |

Table 17: Performance comparison across three different non-IID Levels of $P(X)$ and three distribution drifting levels on the CIFAR-10 dataset.

| Non-IID Level | Low | | | Medium | | | High | | |
|---|---|---|---|---|---|---|---|---|---|
| # Drifting | 5 / 20 | 10 / 20 | 20 / 20 | 5 / 20 | 10 / 20 | 20 / 20 | 5 / 20 | 10 / 20 | 20 / 20 |
| FedAvg | 44.72 ± 14.83 | 43.09 ± 3.40 | 56.75 ± 9.04 | 40.86 ± 9.09 | 45.75 ± 9.38 | 42.29 ± 2.39 | 38.33 ± 5.13 | 37.35 ± 9.07 | 44.29 ± 11.23 |
| FedRC | 43.49 ± 15.66 | 43.32 ± 4.50 | 54.57 ± 8.04 | 26.99 ± 5.28 | 43.05 ± 9.04 | 39.41 ± 5.05 | 27.16 ± 1.97 | 35.08 ± 9.91 | 44.42 ± 8.51 |
| FedEM | 42.95 ± 13.97 | 47.11 ± 6.56 | 50.36 ± 11.30 | 29.77 ± 6.50 | 39.36 ± 10.00 | 41.62 ± 4.74 | 26.85 ± 6.04 | 34.51 ± 10.46 | 44.16 ± 8.45 |
| FeSEM | 38.80 ± 7.70 | 40.48 ± 8.70 | 49.92 ± 8.95 | 31.75 ± 10.34 | 34.09 ± 10.76 | 31.37 ± 9.11 | 27.47 ± 7.95 | 33.42 ± 7.81 | 34.03 ± 6.70 |
| CFL | 48.02 ± 14.91 | 44.00 ± 6.16 | 55.99 ± 9.65 | 32.79 ± 5.51 | 48.37 ± 6.23 | 39.98 ± 3.12 | 39.13 ± 8.32 | 44.26 ± 9.04 | 41.43 ± 3.54 |
| IFCA | 36.96 ± 10.36 | 41.26 ± 6.62 | 32.50 ± 13.51 | 27.02 ± 5.27 | 27.12 ± 9.02 | 19.76 ± 6.24 | 23.55 ± 2.51 | 18.80 ± 2.99 | 20.10 ± 5.28 |
| pFedMe | 30.75 ± 8.65 | 29.95 ± 9.66 | 26.51 ± 11.54 | 27.57 ± 7.48 | 23.00 ± 8.06 | 16.86 ± 5.77 | 20.37 ± 4.17 | 20.03 ± 3.64 | 18.92 ± 6.83 |
| APFL | 42.48 ± 9.30 | 43.23 ± 5.90 | 39.24 ± 12.88 | 36.89 ± 6.09 | 34.81 ± 8.88 | 25.85 ± 4.95 | 30.50 ± 4.78 | 30.89 ± 8.94 | 29.99 ± 6.88 |
| FedDrift | 38.37 ± 10.39 | 41.35 ± 8.44 | 43.44 ± 12.65 | 33.52 ± 9.61 | 31.56 ± 9.90 | 26.85 ± 4.69 | 27.19 ± 5.98 | 27.96 ± 4.74 | 32.92 ± 4.20 |
| ATP | 30.51 ± 14.24 | 24.22 ± 6.96 | 34.46 ± 8.80 | 23.71 ± 7.36 | 24.68 ± 12.85 | 29.76 ± 13.14 | 18.05 ± 3.42 | 16.18 ± 5.04 | 35.29 ± 1.99 |
| FEROMA | **73.04 ± 8.46** | **73.15 ± 3.82** | **68.69 ± 5.26** | **58.31 ± 5.08** | **62.88 ± 6.31** | **58.60 ± 10.85** | **55.11 ± 3.82** | **67.47 ± 8.17** | **58.99 ± 6.57** |

Table 18: Performance comparison across three different non-IID Levels of $P(Y)$ and three distribution drifting levels on the CIFAR-10 dataset.

| Non-IID Level | Low | | | Medium | | | High | | |
|---|---|---|---|---|---|---|---|---|---|
| # Drifting | 5 / 20 | 10 / 20 | 20 / 20 | 5 / 20 | 10 / 20 | 20 / 20 | 5 / 20 | 10 / 20 | 20 / 20 |
| FedAvg | 40.05 ± 1.04 | 39.85 ± 1.44 | 39.65 ± 1.18 | 36.09 ± 0.97 | 36.12 ± 1.08 | 35.92 ± 1.07 | 32.56 ± 1.96 | 32.15 ± 2.25 | 32.05 ± 1.79 |
| FedRC | 27.06 ± 1.22 | 26.70 ± 1.53 | 27.26 ± 1.15 | 12.17 ± 1.64 | 11.16 ± 0.86 | 12.44 ± 1.75 | 11.31 ± 1.42 | 11.12 ± 0.84 | 10.95 ± 1.00 |
| FedEM | 27.03 ± 1.22 | 26.73 ± 1.51 | 27.25 ± 1.16 | 12.17 ± 1.64 | 11.15 ± 0.86 | 12.44 ± 1.76 | 11.31 ± 1.42 | 11.12 ± 0.84 | 10.95 ± 0.99 |
| FeSEM | 40.62 ± 0.65 | 40.94 ± 0.59 | 40.78 ± 0.84 | 38.36 ± 0.63 | 37.51 ± 1.02 | 38.51 ± 0.59 | 35.78 ± 1.17 | 36.26 ± 0.63 | 37.15 ± 0.66 |
| CFL | 41.17 ± 1.06 | 40.90 ± 1.30 | 40.91 ± 0.98 | 36.99 ± 1.06 | 37.09 ± 0.83 | 36.70 ± 1.19 | 33.39 ± 1.75 | 32.93 ± 2.00 | 32.86 ± 1.92 |
| IFCA | **42.36 ± 2.22** | 42.41 ± 2.42 | 42.09 ± 1.36 | 38.73 ± 1.97 | 38.79 ± 1.48 | 38.68 ± 0.54 | 35.81 ± 2.07 | 35.74 ± 1.57 | 34.98 ± 1.78 |
| pFedMe | 36.54 ± 0.51 | 36.83 ± 0.51 | 37.19 ± 0.47 | 36.26 ± 0.44 | 36.83 ± 0.51 | 37.57 ± 0.36 | 36.57 ± 0.65 | 37.27 ± 0.35 | 37.82 ± 0.44 |
| APFL | 41.78 ± 0.53 | 42.17 ± 0.31 | **42.59 ± 0.61** | 40.88 ± 0.40 | 41.66 ± 0.33 | 41.73 ± 0.32 | 40.44 ± 0.74 | 40.47 ± 0.58 | **41.32 ± 0.82** |
| FedDrift | 40.79 ± 0.29 | 41.04 ± 0.89 | 40.66 ± 0.58 | 37.13 ± 0.33 | 37.09 ± 0.23 | 37.13 ± 0.22 | 33.47 ± 1.77 | 34.07 ± 1.56 | 33.28 ± 1.48 |
| ATP | 39.37 ± 1.16 | 38.52 ± 0.73 | 39.58 ± 0.98 | 35.38 ± 0.82 | 36.13 ± 0.92 | 35.24 ± 0.97 | 31.67 ± 1.24 | 31.74 ± 1.19 | 31.71 ± 1.48 |
| FEROMA | 42.22 ± 0.54 | **42.47 ± 0.32** | 42.14 ± 0.32 | **41.18 ± 0.78** | **41.74 ± 0.25** | **41.47 ± 0.48** | **40.76 ± 0.61** | **41.01 ± 0.50** | 40.74 ± 0.64 |

Table 19: Performance comparison across three different non-IID Levels of $P(Y|X)$ and three distribution drifting levels on the CIFAR-10 dataset.

| Non-IID Level | Low | | | Medium | | | High | | |
|---|---|---|---|---|---|---|---|---|---|
| # Drifting | 5 / 20 | 10 / 20 | 20 / 20 | 5 / 20 | 10 / 20 | 20 / 20 | 5 / 20 | 10 / 20 | 20 / 20 |
| FedAvg | 32.17 ± 3.46 | 29.12 ± 3.97 | 26.62 ± 5.79 | 25.30 ± 3.66 | 25.37 ± 2.66 | 21.87 ± 3.09 | 26.09 ± 1.18 | 22.42 ± 2.15 | 23.92 ± 2.27 |
| FedRC | 22.01 ± 2.43 | 22.06 ± 1.52 | 25.18 ± 3.29 | 18.71 ± 2.85 | 18.01 ± 1.87 | 20.00 ± 3.76 | 16.28 ± 1.71 | 17.35 ± 1.83 | 15.62 ± 1.05 |
| FedEM | 20.87 ± 1.85 | 22.16 ± 4.10 | 22.98 ± 4.36 | 17.29 ± 1.90 | 18.33 ± 1.56 | 17.94 ± 1.78 | 15.71 ± 1.33 | 16.04 ± 2.11 | 16.27 ± 3.22 |
| FeSEM | 28.27 ± 2.44 | 29.70 ± 4.41 | 29.65 ± 4.45 | 25.57 ± 1.75 | 23.16 ± 3.36 | 22.54 ± 3.93 | 23.41 ± 2.41 | 21.74 ± 1.32 | 19.97 ± 2.78 |
| CFL | 29.29 ± 2.46 | 27.27 ± 4.88 | 29.32 ± 3.62 | 25.70 ± 3.39 | 27.69 ± 5.36 | 25.37 ± 4.29 | 22.85 ± 1.60 | 27.85 ± 2.50 | 25.02 ± 1.80 |
| IFCA | 23.48 ± 7.55 | 29.57 ± 3.28 | 22.87 ± 6.26 | 19.48 ± 6.33 | 18.58 ± 3.92 | 16.99 ± 2.15 | 17.32 ± 3.86 | 17.54 ± 3.90 | 10.61 ± 0.96 |
| pFedMe | 18.60 ± 3.04 | 22.47 ± 4.39 | 16.79 ± 3.10 | 16.65 ± 1.25 | 15.09 ± 2.31 | 12.75 ± 1.96 | 15.59 ± 1.87 | 14.09 ± 1.25 | 13.01 ± 0.94 |
| APFL | 25.02 ± 5.18 | 26.81 ± 3.06 | 29.18 ± 6.42 | 23.17 ± 3.43 | 24.37 ± 1.86 | 22.87 ± 2.75 | 22.45 ± 1.77 | 24.00 ± 2.79 | 24.82 ± 2.76 |
| FedDrift | 25.17 ± 7.09 | 28.72 ± 1.31 | 25.30 ± 5.45 | 19.48 ± 0.59 | 23.20 ± 3.84 | 20.53 ± 4.66 | 20.11 ± 3.67 | 22.37 ± 1.30 | 20.99 ± 2.88 |
| ATP | 27.81 ± 1.81 | 24.99 ± 6.25 | 26.08 ± 6.81 | 24.52 ± 1.81 | 26.54 ± 1.74 | 24.12 ± 2.24 | 22.49 ± 1.94 | 22.87 ± 1.43 | 21.48 ± 2.82 |
| FEROMA | **40.29 ± 3.29** | **38.32 ± 3.00** | **39.89 ± 2.23** | **38.54 ± 2.42** | **38.12 ± 2.48** | **36.96 ± 3.85** | **35.36 ± 1.86** | **35.83 ± 1.02** | **31.87 ± 1.76** |

Table 20: Performance comparison across three different non-IID Levels of $P(X|Y)$ and three distribution drifting levels on the CIFAR-10 dataset.

| Non-IID Level | Low | | | Medium | | | High | | |
|---|---|---|---|---|---|---|---|---|---|
| # Drifting | 5 / 20 | 10 / 20 | 20 / 20 | 5 / 20 | 10 / 20 | 20 / 20 | 5 / 20 | 10 / 20 | 20 / 20 |
| FedAvg | 61.86 ± 2.02 | 63.82 ± 1.54 | 64.19 ± 2.15 | 69.66 ± 1.31 | 69.87 ± 1.28 | 70.10 ± 0.46 | 55.85 ± 1.90 | 54.46 ± 2.30 | 55.94 ± 3.49 |
| FedRC | 46.24 ± 0.55 | 48.18 ± 2.34 | 50.42 ± 3.99 | 41.89 ± 3.53 | 44.61 ± 2.33 | 46.15 ± 5.89 | 12.55 ± 1.69 | 11.32 ± 0.82 | 11.65 ± 1.01 |
| FedEM | 46.08 ± 0.67 | 48.66 ± 2.49 | 50.10 ± 3.85 | 42.06 ± 3.52 | 44.80 ± 2.20 | 46.06 ± 5.96 | 12.56 ± 1.70 | 11.33 ± 0.84 | 11.64 ± 1.00 |
| FeSEM | 54.13 ± 4.07 | 54.84 ± 3.64 | 58.77 ± 5.76 | 65.00 ± 5.61 | 64.55 ± 4.52 | 66.59 ± 1.45 | 35.51 ± 6.56 | 38.13 ± 7.11 | 37.48 ± 5.87 |
| CFL | 65.82 ± 1.37 | 66.51 ± 0.99 | 67.89 ± 1.85 | 72.15 ± 1.41 | 71.89 ± 1.47 | 71.92 ± 0.39 | 59.43 ± 1.67 | 59.64 ± 1.10 | 58.13 ± 1.97 |
| IFCA | 27.72 ± 4.23 | 29.86 ± 7.74 | 33.61 ± 8.65 | 33.60 ± 7.54 | 35.74 ± 8.13 | 36.89 ± 7.38 | 19.16 ± 5.33 | 23.13 ± 12.49 | 10.32 ± 1.63 |
| pFedMe | 23.66 ± 1.94 | 28.71 ± 8.46 | 30.71 ± 3.25 | 41.81 ± 4.97 | 43.83 ± 4.32 | 47.87 ± 3.97 | 23.14 ± 7.06 | 24.29 ± 6.20 | 22.30 ± 4.67 |
| APFL | 41.28 ± 1.82 | 46.68 ± 5.77 | 49.51 ± 2.94 | 62.39 ± 3.02 | 64.52 ± 2.27 | 65.16 ± 2.21 | 38.03 ± 6.09 | 40.97 ± 5.78 | 41.25 ± 4.07 |
| FedDrift | 28.18 ± 2.48 | 33.83 ± 8.95 | 35.47 ± 3.53 | 40.77 ± 5.48 | 45.99 ± 6.78 | 48.31 ± 2.64 | 26.69 ± 8.62 | 28.36 ± 6.97 | 24.33 ± 4.42 |
| ATP | 60.87 ± 3.38 | 63.19 ± 2.03 | 62.96 ± 5.33 | 74.32 ± 0.48 | 72.71 ± 2.86 | 74.41 ± 0.98 | 54.80 ± 2.33 | 55.41 ± 1.17 | 59.67 ± 2.37 |
| FEROMA | **73.79 ± 0.41** | **73.83 ± 0.68** | **73.45 ± 0.76** | **74.84 ± 0.49** | **75.12 ± 0.60** | **75.03 ± 0.51** | **72.26 ± 0.37** | **71.62 ± 0.34** | **71.99 ± 0.65** |

Table 21: Performance comparison across three different non-IID Levels of $P(X)$ and three distribution drifting levels on the FMNIST dataset.

| Non-IID Level | Low | | | Medium | | | High | | |
|---|---|---|---|---|---|---|---|---|---|
| # Drifting | 5 / 20 | 10 / 20 | 20 / 20 | 5 / 20 | 10 / 20 | 20 / 20 | 5 / 20 | 10 / 20 | 20 / 20 |
| FedAvg | 77.11 ± 12.18 | 72.21 ± 4.40 | 83.51 ± 5.09 | 65.95 ± 6.90 | 68.24 ± 10.91 | 79.23 ± 5.01 | 66.09 ± 8.88 | 55.08 ± 16.79 | 66.47 ± 20.60 |
| FedRC | 73.33 ± 13.70 | 68.69 ± 6.19 | 73.02 ± 12.34 | 64.14 ± 3.60 | 65.21 ± 9.05 | 62.11 ± 12.16 | 56.67 ± 5.41 | 53.72 ± 5.36 | 66.30 ± 11.81 |
| FedEM | 74.14 ± 9.01 | 76.95 ± 5.14 | 76.74 ± 8.07 | 60.32 ± 12.02 | 64.75 ± 4.05 | 68.48 ± 12.17 | 51.09 ± 9.43 | 60.92 ± 10.01 | 66.79 ± 11.45 |
| FeSEM | 73.41 ± 11.73 | 69.42 ± 9.37 | 69.29 ± 8.17 | 62.55 ± 5.86 | 60.59 ± 9.05 | 67.36 ± 8.50 | 50.85 ± 6.67 | 44.75 ± 8.58 | 64.80 ± 6.96 |
| CFL | 73.96 ± 13.42 | 73.43 ± 7.40 | 77.72 ± 8.62 | 70.97 ± 4.60 | 71.86 ± 8.44 | 80.63 ± 5.11 | 65.63 ± 4.32 | 62.01 ± 12.34 | 72.70 ± 9.01 |
| IFCA | 39.34 ± 12.56 | 42.07 ± 11.56 | 37.18 ± 13.17 | 37.72 ± 15.67 | 37.57 ± 7.43 | 27.73 ± 9.95 | 25.37 ± 7.80 | 21.58 ± 3.61 | 24.19 ± 6.23 |
| pFedMe | 43.05 ± 12.37 | 44.31 ± 8.35 | 38.76 ± 17.94 | 38.08 ± 13.36 | 33.00 ± 11.55 | 22.77 ± 6.17 | 31.35 ± 5.91 | 24.93 ± 7.18 | 22.75 ± 5.67 |
| APFL | 58.41 ± 9.33 | 56.31 ± 5.97 | 56.14 ± 12.79 | 51.51 ± 7.18 | 49.41 ± 9.51 | 44.99 ± 5.45 | 42.94 ± 4.82 | 42.26 ± 8.01 | 48.67 ± 8.59 |
| FedDrift | 45.09 ± 12.05 | 44.59 ± 11.60 | 41.92 ± 17.64 | 34.30 ± 6.98 | 44.59 ± 11.60 | 29.84 ± 6.98 | 55.02 ± 10.12 | 34.61 ± 4.94 | 35.46 ± 6.49 |
| ATP | 65.72 ± 19.17 | 51.93 ± 14.65 | 43.36 ± 32.78 | 61.35 ± 17.68 | 67.94 ± 18.31 | 55.95 ± 26.14 | 48.92 ± 27.49 | 46.53 ± 24.37 | 46.90 ± 26.28 |
| FEROMA | **96.58 ± 1.81** | **97.82 ± 1.46** | **97.82 ± 2.50** | **93.48 ± 4.75** | **96.09 ± 2.78** | **97.30 ± 1.88** | **93.26 ± 5.43** | **95.04 ± 4.29** | **95.50 ± 2.87** |

Table 22: Performance comparison across three different non-IID Levels of $P(Y)$ and three distribution drifting levels on the FMNIST dataset.

| Non-IID Level | Low | | | Medium | | | High | | |
|---|---|---|---|---|---|---|---|---|---|
| # Drifting | 5 / 20 | 10 / 20 | 20 / 20 | 5 / 20 | 10 / 20 | 20 / 20 | 5 / 20 | 10 / 20 | 20 / 20 |
| FedAvg | 62.98 ± 1.74 | 61.83 ± 2.81 | 62.62 ± 2.74 | 54.90 ± 2.56 | 54.45 ± 2.28 | 54.32 ± 2.49 | 50.96 ± 2.68 | 50.88 ± 3.46 | 50.03 ± 3.31 |
| FedRC | 58.95 ± 3.00 | 58.25 ± 3.15 | 58.66 ± 2.79 | 26.85 ± 11.47 | 30.44 ± 9.46 | 28.24 ± 7.44 | 20.77 ± 4.16 | 20.39 ± 4.02 | 19.06 ± 4.58 |
| FedEM | 59.08 ± 3.05 | 58.23 ± 3.13 | 58.61 ± 2.71 | 26.83 ± 11.46 | 30.43 ± 9.45 | 28.25 ± 7.45 | 20.77 ± 4.16 | 20.40 ± 4.02 | 19.06 ± 4.59 |
| FeSEM | 66.49 ± 2.08 | 66.50 ± 2.45 | 67.38 ± 2.03 | 64.98 ± 1.60 | 66.24 ± 1.46 | 64.84 ± 1.59 | 63.65 ± 2.65 | 65.37 ± 1.68 | 64.42 ± 3.99 |
| CFL | 64.04 ± 1.44 | 63.18 ± 2.53 | 63.47 ± 2.20 | 56.21 ± 1.85 | 55.72 ± 2.18 | 55.63 ± 2.12 | 51.07 ± 3.35 | 52.08 ± 3.61 | 50.25 ± 3.32 |
| IFCA | 69.58 ± 3.35 | 68.59 ± 2.98 | 68.28 ± 4.08 | 61.89 ± 1.76 | 60.78 ± 3.52 | 61.45 ± 2.78 | 49.74 ± 14.68 | 50.01 ± 9.74 | 42.42 ± 16.44 |
| pFedMe | 76.73 ± 0.34 | 76.42 ± 0.30 | 76.30 ± 0.35 | 76.43 ± 0.68 | 75.74 ± 0.15 | 75.71 ± 0.38 | 75.91 ± 0.38 | **75.33 ± 0.24** | **74.70 ± 0.36** |
| APFL | **78.63 ± 0.54** | **78.06 ± 0.59** | **77.90 ± 0.68** | **77.12 ± 0.82** | **76.94 ± 0.69** | **75.82 ± 1.03** | **76.31 ± 0.52** | 75.21 ± 0.64 | 74.31 ± 0.78 |
| FedDrift | 77.56 ± 0.74 | 76.93 ± 1.55 | 77.77 ± 0.30 | 75.52 ± 1.28 | 76.00 ± 0.60 | 75.12 ± 0.83 | 74.75 ± 0.94 | 73.22 ± 0.50 | 71.89 ± 0.97 |
| ATP | 64.48 ± 1.70 | 63.69 ± 2.61 | 63.63 ± 2.53 | 56.88 ± 2.56 | 56.39 ± 1.96 | 56.51 ± 3.13 | 53.05 ± 3.06 | 53.50 ± 3.54 | 52.00 ± 3.23 |
| FEROMA | 75.37 ± 0.33 | 75.23 ± 0.42 | 75.25 ± 0.27 | 74.83 ± 0.38 | 74.64 ± 0.37 | 74.69 ± 0.36 | 74.29 ± 0.37 | 73.83 ± 0.42 | 73.93 ± 0.26 |

Table 23: Performance comparison across three different non-IID Levels of $P(Y|X)$ and three distribution drifting levels on the FMNIST dataset.

| Non-IID Level | Low | | | Medium | | | High | | |
|---|---|---|---|---|---|---|---|---|---|
| # Drifting | 5 / 20 | 10 / 20 | 20 / 20 | 5 / 20 | 10 / 20 | 20 / 20 | 5 / 20 | 10 / 20 | 20 / 20 |
| FedAvg | 65.15 ± 10.19 | 69.65 ± 3.73 | 67.79 ± 3.94 | 64.46 ± 1.14 | 66.62 ± 2.34 | 66.23 ± 4.61 | 64.09 ± 1.64 | 61.99 ± 5.29 | 64.42 ± 2.26 |
| FedRC | 48.93 ± 5.84 | 60.64 ± 5.33 | 52.11 ± 4.32 | 43.63 ± 4.19 | 48.53 ± 5.81 | 46.30 ± 9.01 | 34.81 ± 10.03 | 33.24 ± 9.71 | 36.21 ± 7.53 |
| FedEM | 56.66 ± 5.44 | 59.05 ± 2.30 | 62.28 ± 3.88 | 44.11 ± 5.97 | 43.58 ± 4.64 | 42.91 ± 11.92 | 44.28 ± 9.71 | 37.80 ± 10.92 | 34.75 ± 8.75 |
| FeSEM | 63.18 ± 2.08 | 63.19 ± 3.29 | 65.68 ± 1.07 | 55.72 ± 5.48 | 54.20 ± 4.28 | 54.19 ± 4.57 | 47.71 ± 4.15 | 44.83 ± 5.24 | 48.73 ± 7.02 |
| CFL | 71.20 ± 1.69 | 69.80 ± 2.05 | 72.93 ± 2.53 | 65.72 ± 4.45 | 66.24 ± 2.24 | 68.24 ± 1.58 | 63.27 ± 4.00 | 66.18 ± 4.57 | 65.05 ± 2.33 |
| IFCA | 34.03 ± 15.83 | 40.55 ± 10.38 | 23.44 ± 11.22 | 33.76 ± 12.59 | 30.71 ± 8.15 | 23.22 ± 6.64 | 24.94 ± 3.71 | 18.30 ± 6.85 | 23.63 ± 10.05 |
| pFedMe | 37.33 ± 7.78 | 41.24 ± 3.44 | 35.80 ± 10.07 | 35.75 ± 7.34 | 30.55 ± 5.25 | 28.78 ± 5.66 | 30.81 ± 7.53 | 28.17 ± 3.18 | 31.60 ± 5.50 |
| APFL | 54.85 ± 6.85 | 56.22 ± 4.92 | 56.31 ± 7.42 | 47.96 ± 6.79 | 50.92 ± 8.42 | 48.10 ± 5.86 | 49.11 ± 10.42 | 45.82 ± 5.65 | 46.84 ± 2.19 |
| FedDrift | 40.93 ± 7.29 | 46.06 ± 5.83 | 46.16 ± 8.85 | 38.03 ± 10.57 | 34.23 ± 6.04 | 38.54 ± 9.45 | 36.15 ± 6.39 | 37.69 ± 4.47 | 38.31 ± 8.71 |
| ATP | 70.44 ± 2.72 | 69.64 ± 1.43 | 71.45 ± 4.13 | 67.13 ± 4.32 | 67.65 ± 6.12 | 69.13 ± 1.35 | 66.85 ± 2.51 | 66.97 ± 4.65 | 62.36 ± 4.34 |
| FEROMA | **78.74 ± 1.78** | **75.17 ± 5.05** | **78.98 ± 1.49** | **75.15 ± 3.95** | **76.95 ± 3.40** | **73.12 ± 6.37** | **74.21 ± 2.44** | **76.79 ± 3.44** | **70.23 ± 7.08** |

Table 24: Performance comparison across three different non-IID Levels of $P(X|Y)$ and three distribution drifting levels on the FMNIST dataset.

| Non-IID Level | Low | | | Medium | | | High | | |
|---|---|---|---|---|---|---|---|---|---|
| # Drifting | 5 / 20 | 10 / 20 | 20 / 20 | 5 / 20 | 10 / 20 | 20 / 20 | 5 / 20 | 10 / 20 | 20 / 20 |
| FedAvg | 43.85 ± 0.80 | 45.13 ± 0.68 | 44.87 ± 0.87 | 7.47 ± 5.13 | 13.74 ± 4.59 | 12.84 ± 2.75 | 10.52 ± 8.51 | 8.58 ± 4.26 | 8.30 ± 3.61 |
| FedRC | **44.97 ± 0.73** | **46.90 ± 0.73** | **48.07 ± 0.83** | 13.08 ± 0.64 | 16.64 ± 4.18 | 15.70 ± 1.96 | 19.97 ± 5.97 | 11.93 ± 2.50 | 13.97 ± 2.67 |
| FedEM | **44.97 ± 0.73** | 46.89 ± 0.72 | **48.07 ± 0.83** | 13.08 ± 0.64 | 16.63 ± 4.19 | 15.69 ± 1.95 | 19.97 ± 5.98 | 11.93 ± 2.51 | 13.97 ± 2.68 |
| FeSEM | 37.76 ± 1.27 | 40.33 ± 1.92 | 43.35 ± 2.04 | 11.57 ± 0.56 | 18.50 ± 4.07 | 15.92 ± 0.40 | 15.91 ± 3.34 | 10.62 ± 2.23 | 12.31 ± 1.79 |
| CFL | 43.73 ± 0.59 | 45.90 ± 0.55 | 46.98 ± 0.68 | 13.04 ± 1.82 | 15.91 ± 4.00 | 14.72 ± 1.56 | 18.95 ± 5.76 | 11.94 ± 2.42 | 11.45 ± 2.66 |
| IFCA | 43.19 ± 0.45 | 44.95 ± 0.68 | 45.69 ± 0.59 | 6.85 ± 2.79 | 0.99 ± 0.05 | 8.68 ± 4.58 | 9.43 ± 5.60 | 4.16 ± 0.74 | 6.79 ± 4.52 |
| pFedMe | 18.15 ± 0.85 | 20.58 ± 1.55 | 22.31 ± 0.48 | 7.17 ± 1.20 | 9.66 ± 2.02 | 10.43 ± 0.20 | 8.49 ± 2.11 | 8.23 ± 1.32 | 8.71 ± 0.87 |
| APFL | 37.54 ± 0.29 | 39.47 ± 1.43 | 40.66 ± 0.73 | 12.85 ± 0.59 | 15.70 ± 3.41 | 17.01 ± 0.70 | 16.73 ± 5.16 | 12.96 ± 1.77 | 14.38 ± 2.67 |
| FedDrift | 21.36 ± 1.13 | 23.62 ± 4.06 | 33.93 ± 3.34 | 8.72 ± 2.36 | 13.81 ± 3.54 | 14.71 ± 1.15 | 10.16 ± 3.69 | 10.00 ± 2.13 | 11.87 ± 1.71 |
| ATP | 11.77 ± 5.80 | 19.05 ± 3.56 | 14.29 ± 6.41 | 3.13 ± 1.04 | 3.47 ± 1.65 | 5.28 ± 1.49 | 2.96 ± 0.36 | 3.65 ± 0.85 | 5.56 ± 1.57 |
| FEROMA | 39.83 ± 0.62 | 39.40 ± 0.79 | 39.37 ± 0.81 | **36.97 ± 0.97** | **36.73 ± 0.46** | **36.53 ± 0.74** | **32.74 ± 1.47** | **34.15 ± 0.44** | **33.99 ± 0.89** |

Table 25: Performance comparison across three different non-IID Levels of $P(X)$ and three distribution drifting levels on the CIFAR-100 dataset.

| Non-IID Level | Low | | | Medium | | | High | | |
|---|---|---|---|---|---|---|---|---|---|
| # Drifting | 5 / 20 | 10 / 20 | 20 / 20 | 5 / 20 | 10 / 20 | 20 / 20 | 5 / 20 | 10 / 20 | 20 / 20 |
| FedAvg | 35.10 ± 7.62 | 35.82 ± 5.54 | 41.28 ± 5.48 | 36.65 ± 3.67 | 40.69 ± 1.77 | 41.53 ± 3.70 | 33.17 ± 5.28 | 35.96 ± 3.40 | 37.42 ± 2.93 |
| FedRC | 34.83 ± 5.04 | 41.09 ± 6.33 | 38.67 ± 3.63 | 32.36 ± 5.43 | 36.92 ± 3.39 | 42.43 ± 3.39 | 40.45 ± 4.58 | 39.98 ± 2.23 | 42.35 ± 3.52 |
| FedEM | 36.03 ± 5.49 | 41.05 ± 6.33 | 38.40 ± 3.46 | 32.33 ± 5.33 | 36.80 ± 3.50 | 42.12 ± 3.38 | 40.32 ± 4.17 | 41.86 ± 3.48 | 42.17 ± 3.29 |
| FeSEM | 34.43 ± 7.45 | 35.61 ± 4.14 | 36.29 ± 3.80 | 29.09 ± 6.25 | 32.54 ± 3.23 | 30.65 ± 3.77 | 31.02 ± 4.26 | 32.24 ± 1.76 | 35.33 ± 2.50 |
| CFL | 33.89 ± 6.66 | 37.97 ± 5.32 | 33.51 ± 4.11 | 29.53 ± 3.35 | 37.14 ± 4.07 | 39.35 ± 4.70 | 35.67 ± 5.18 | 35.24 ± 2.77 | 38.48 ± 2.87 |
| IFCA | 11.28 ± 7.98 | 10.18 ± 2.18 | 12.97 ± 3.40 | 2.27 ± 1.28 | 8.85 ± 8.63 | 4.66 ± 1.67 | 2.45 ± 0.72 | 2.44 ± 0.36 | 2.65 ± 0.49 |
| pFedMe | 16.63 ± 5.20 | 13.85 ± 3.22 | 16.55 ± 1.22 | 11.88 ± 2.94 | 16.49 ± 4.73 | 11.26 ± 3.07 | 13.61 ± 2.90 | 11.87 ± 3.06 | 11.59 ± 3.26 |
| APFL | 35.11 ± 6.15 | 34.02 ± 1.18 | 34.25 ± 0.85 | 29.31 ± 4.51 | 31.13 ± 0.77 | 31.86 ± 4.09 | 30.90 ± 2.53 | 30.73 ± 2.87 | 34.72 ± 3.24 |
| FedDrift | 24.21 ± 7.61 | 21.91 ± 4.63 | 24.89 ± 2.87 | 18.08 ± 3.98 | 23.51 ± 6.41 | 17.09 ± 4.13 | 18.05 ± 3.02 | 17.42 ± 2.72 | 22.81 ± 4.11 |
| ATP | 15.65 ± 4.07 | 36.75 ± 6.32 | 40.72 ± 4.09 | 15.68 ± 5.18 | 36.71 ± 5.50 | 42.90 ± 6.47 | 18.53 ± 10.89 | 35.31 ± 4.91 | 40.81 ± 4.82 |
| FEROMA | **44.65 ± 4.42** | **51.03 ± 6.51** | **51.90 ± 2.13** | **48.43 ± 3.40** | **48.51 ± 2.59** | **52.90 ± 4.83** | **40.82 ± 5.57** | **47.56 ± 6.77** | **45.91 ± 3.54** |

Table 26: Performance comparison across three different non-IID Levels of $P(Y)$ and three distribution drifting levels on the CIFAR-100 dataset.

| Non-IID Level | Low | | | Medium | | | High | | |
|---|---|---|---|---|---|---|---|---|---|
| # Drifting | 5 / 20 | 10 / 20 | 20 / 20 | 5 / 20 | 10 / 20 | 20 / 20 | 5 / 20 | 10 / 20 | 20 / 20 |
| FedAvg | 37.69 ± 0.81 | 37.84 ± 0.79 | 38.16 ± 0.98 | 27.39 ± 1.10 | 27.44 ± 1.26 | 27.44 ± 1.20 | 15.83 ± 0.49 | 15.75 ± 0.31 | 15.90 ± 0.40 |
| FedRC | 38.94 ± 0.80 | 38.98 ± 0.96 | 39.23 ± 0.94 | 28.14 ± 1.12 | 28.14 ± 1.15 | 28.16 ± 1.33 | 16.24 ± 0.41 | 16.19 ± 0.40 | 16.09 ± 0.33 |
| FedEM | 38.94 ± 0.80 | 38.98 ± 0.96 | 39.23 ± 0.94 | 28.14 ± 1.12 | 28.14 ± 1.15 | 28.16 ± 1.33 | 16.24 ± 0.41 | 16.19 ± 0.40 | 16.09 ± 0.33 |
| FeSEM | 35.99 ± 0.89 | 34.00 ± 1.04 | 33.39 ± 1.35 | 26.45 ± 1.11 | 26.23 ± 1.16 | 25.84 ± 1.30 | 16.00 ± 0.58 | 16.61 ± 0.55 | 16.47 ± 0.51 |
| CFL | 38.28 ± 0.69 | 38.61 ± 0.84 | 38.57 ± 0.68 | 27.78 ± 1.13 | 27.69 ± 1.21 | 27.63 ± 1.31 | 16.03 ± 0.42 | 15.92 ± 0.39 | 15.90 ± 0.41 |
| IFCA | 38.17 ± 0.74 | 38.36 ± 0.87 | 38.54 ± 0.83 | 27.78 ± 1.25 | 27.62 ± 1.30 | 27.83 ± 1.31 | 0.97 ± 0.05 | 1.00 ± 0.06 | 0.97 ± 0.09 |
| pFedMe | 24.81 ± 0.19 | 24.88 ± 0.17 | 24.41 ± 0.24 | 24.28 ± 0.26 | 24.73 ± 0.25 | 23.89 ± 0.24 | 24.34 ± 0.65 | 24.32 ± 0.27 | 22.34 ± 0.14 |
| APFL | **41.52 ± 0.19** | **42.07 ± 0.85** | **42.25 ± 0.48** | 35.44 ± 0.49 | **35.79 ± 0.21** | **36.23 ± 0.28** | 29.43 ± 0.37 | 29.85 ± 0.38 | 29.88 ± 0.24 |
| FedDrift | 28.70 ± 0.09 | 26.42 ± 1.03 | 36.93 ± 1.26 | 26.56 ± 0.32 | 25.76 ± 0.35 | 26.87 ± 0.88 | 25.08 ± 0.32 | 19.23 ± 0.88 | 15.98 ± 0.60 |
| ATP | 19.63 ± 1.67 | 25.84 ± 1.81 | 20.18 ± 3.30 | 17.03 ± 2.41 | 19.95 ± 2.22 | 20.22 ± 1.81 | 12.41 ± 0.66 | 12.16 ± 0.53 | 15.54 ± 0.79 |
| FEROMA | 38.88 ± 0.28 | 39.62 ± 0.50 | 39.36 ± 0.30 | **35.63 ± 0.30** | 35.71 ± 0.29 | 35.72 ± 0.08 | **31.75 ± 0.14** | **32.46 ± 0.27** | **32.15 ± 0.25** |

Table 27: Performance comparison across three different non-IID Levels of $P(Y|X)$ and three distribution drifting levels on the CIFAR-100 dataset.

| Non-IID Level | Low | | | Medium | | | High | | |
|---|---|---|---|---|---|---|---|---|---|
| # Drifting | 5 / 20 | 10 / 20 | 20 / 20 | 5 / 20 | 10 / 20 | 20 / 20 | 5 / 20 | 10 / 20 | 20 / 20 |
| FedAvg | 33.07 ± 7.39 | 29.13 ± 8.76 | 28.00 ± 5.13 | 23.18 ± 7.07 | 28.56 ± 8.56 | 21.69 ± 1.68 | 25.20 ± 8.82 | 24.30 ± 6.84 | 27.08 ± 4.87 |
| FedRC | 25.46 ± 2.53 | 35.48 ± 11.24 | 29.95 ± 5.14 | 26.94 ± 2.37 | 32.09 ± 3.38 | 30.88 ± 9.80 | 31.46 ± 7.76 | 27.68 ± 0.91 | 30.58 ± 7.60 |
| FedEM | **45.53 ± 3.56** | 37.30 ± 6.95 | 38.41 ± 2.37 | 37.70 ± 6.61 | 31.42 ± 6.46 | 33.02 ± 5.90 | 26.20 ± 4.04 | 29.08 ± 2.66 | 29.85 ± 2.40 |
| FeSEM | 27.20 ± 5.57 | 29.57 ± 6.20 | 24.66 ± 4.30 | 22.86 ± 9.39 | 21.38 ± 3.72 | 23.60 ± 5.06 | 18.62 ± 4.58 | 21.19 ± 3.14 | 19.88 ± 2.59 |
| CFL | 28.45 ± 2.88 | 29.24 ± 6.12 | 34.74 ± 4.86 | 33.92 ± 4.22 | 29.60 ± 7.61 | 25.33 ± 2.70 | 23.74 ± 4.10 | 22.52 ± 3.53 | 24.20 ± 1.50 |
| IFCA | 10.90 ± 7.53 | 23.22 ± 13.20 | 22.02 ± 2.80 | 22.78 ± 8.67 | 14.63 ± 6.90 | 17.00 ± 7.77 | 7.53 ± 6.56 | 7.76 ± 4.82 | 7.68 ± 6.20 |
| pFedMe | 17.02 ± 2.99 | 11.58 ± 3.07 | 12.74 ± 0.32 | 10.83 ± 0.79 | 14.38 ± 4.82 | 10.37 ± 1.07 | 11.87 ± 1.52 | 10.72 ± 1.21 | 14.19 ± 0.71 |
| APFL | 27.51 ± 7.08 | 29.31 ± 6.17 | 27.94 ± 3.97 | 30.56 ± 5.32 | 29.41 ± 2.92 | 30.33 ± 3.73 | 20.23 ± 2.38 | 25.26 ± 5.16 | 26.20 ± 2.74 |
| FedDrift | 22.58 ± 3.57 | 19.13 ± 6.14 | 24.14 ± 0.75 | 15.26 ± 6.22 | 19.76 ± 3.65 | 12.51 ± 3.63 | 13.61 ± 2.02 | 14.84 ± 4.35 | 19.21 ± 4.50 |
| ATP | 6.38 ± 1.63 | 11.15 ± 1.96 | 12.95 ± 2.72 | 7.01 ± 0.73 | 9.56 ± 0.81 | 12.39 ± 1.99 | 5.55 ± 2.03 | 8.40 ± 2.85 | 13.70 ± 4.16 |
| FEROMA | 40.24 ± 1.79 | **40.64 ± 1.00** | **40.41 ± 2.24** | **39.53 ± 1.31** | **38.26 ± 1.94** | **39.73 ± 1.34** | **37.92 ± 0.86** | **39.25 ± 1.08** | **37.99 ± 0.87** |

Table 28: Performance comparison across three different non-IID Levels of $P(X|Y)$ and three distribution drifting levels on the CIFAR-100 dataset.

(subsection I.2); for 20 clients, we retain the same dataset size. For larger scales, we increase the dataset size via duplication: duplicating the dataset once for 50 clients, and twice for 100 clients.

Tables 29 to Table 33 provide detailed results for FLUX and baseline methods, evaluated across increasing numbers of clients and various distribution shift types and levels. We could not evaluate the performance of FedDrift under 50 and 100 clients due to prohibitive memory and computational costs (see Appendix B.2 for details). For the baselines that do not provide a solution for test-only clients, we weight all models by the number of clients in the cluster, and use the expectation that weights all model outputs as an estimation of the predicted labels.

| # Clients | 10 Clients | | 20 Clients | | 50 Clients | | 100 Clients | |
|---|---|---|---|---|---|---|---|---|
| Algorithm | Accuracy | Time | Accuracy | Time | Accuracy | Time | Accuracy | Time |
| FedAvg | 74.85 ± 5.71 | **3.77** | 73.98 ± 3.04 | **5.91** | 74.00 ± 3.33 | **7.16** | 73.27 ± 3.84 | **8.28** |
| FedRC | 29.84 ± 7.47 | 8.63 | 29.44 ± 5.35 | 9.83 | 25.02 ± 3.94 | 11.36 | 24.06 ± 3.44 | 12.38 |
| FedEM | 32.01 ± 8.56 | 8.53 | 26.78 ± 4.58 | 9.27 | 25.95 ± 4.28 | 11.27 | 24.77 ± 3.03 | 12.11 |
| FeSEM | 73.02 ± 6.63 | 7.11 | 72.58 ± 4.65 | 7.65 | 73.11 ± 3.93 | 9.12 | 74.30 ± 3.36 | 9.93 |
| CFL | 77.17 ± 6.96 | 6.32 | 78.22 ± 4.21 | 6.57 | 76.92 ± 4.18 | 8.52 | 76.25 ± 2.88 | 8.74 |
| IFCA | 37.16 ± 10.44 | 6.77 | 45.99 ± 9.90 | 8.14 | 48.40 ± 6.12 | 9.66 | 43.15 ± 5.22 | 10.66 |
| pFedMe | 55.20 ± 12.16 | 5.65 | 55.36 ± 8.79 | 6.60 | 54.40 ± 3.48 | 7.71 | 49.77 ± 5.08 | 9.01 |
| APFL | 71.43 ± 8.02 | 7.40 | 72.24 ± 5.31 | 8.38 | 72.99 ± 4.18 | 10.05 | 68.39 ± 4.96 | 10.23 |
| FedDrift | 63.34 ± 11.52 | 8.74 | 57.88 ± 9.40 | 9.56 | N/A | N/A | N/A | N/A |
| ATP | 71.39 ± 8.95 | 4.95 | 74.83 ± 4.37 | 6.51 | 72.46 ± 4.55 | 7.74 | 71.21 ± 5.45 | 8.68 |
| FEROMA | **91.27 ± 0.67** | 4.84 | **90.17 ± 1.96** | 6.43 | **90.16 ± 1.98** | 7.67 | **87.06 ± 2.70** | 8.61 |

Table 29: Performance comparison across the number of clients in 10, 20, 50, and 100, summarizing all four types of heterogeneity ($P(X)$, $P(Y)$, $P(Y|X)$, $P(X|Y)$) on the MNIST dataset. Time is reported in $\log_2$ seconds.

| # Clients | 10 Clients | | 20 Clients | | 50 Clients | | 100 Clients | |
|---|---|---|---|---|---|---|---|---|
| Algorithm | Accuracy | Time | Accuracy | Time | Accuracy | Time | Accuracy | Time |
| FedAvg | 74.20 ± 3.72 | **3.54** | 74.64 ± 1.50 | **5.74** | 74.50 ± 3.06 | **7.15** | 71.05 ± 3.41 | **8.16** |
| FedRC | 11.44 ± 0.18 | 8.34 | 11.35 ± 0.00 | 9.60 | 11.35 ± 0.00 | 10.95 | 11.35 ± 0.00 | 11.43 |
| FedEM | 11.44 ± 0.18 | 8.07 | 11.35 ± 0.00 | 8.80 | 11.35 ± 0.00 | 10.13 | 11.35 ± 0.00 | 11.49 |
| FeSEM | 69.43 ± 3.59 | 6.90 | 72.98 ± 4.49 | 7.30 | 74.54 ± 1.80 | 8.50 | 75.48 ± 1.34 | 9.38 |
| CFL | 79.66 ± 3.93 | 6.47 | 81.08 ± 2.36 | 6.38 | 78.73 ± 3.18 | 8.40 | 77.04 ± 1.24 | 8.52 |
| IFCA | 22.70 ± 6.75 | 6.51 | 39.79 ± 4.79 | 7.56 | 47.86 ± 9.25 | 9.43 | 42.23 ± 3.40 | 10.15 |
| pFedMe | 52.90 ± 1.71 | 5.57 | 46.24 ± 6.37 | 6.66 | 50.62 ± 3.74 | 7.63 | 47.22 ± 1.47 | 9.06 |
| APFL | 72.89 ± 2.14 | 7.01 | 71.85 ± 3.77 | 7.94 | 74.07 ± 0.96 | 9.97 | 72.40 ± 1.04 | 10.12 |
| FedDrift | 60.13 ± 5.80 | 8.74 | 46.86 ± 8.80 | 9.55 | N/A | N/A | N/A | N/A |
| ATP | 71.08 ± 7.72 | 3.95 | 75.81 ± 2.38 | 6.43 | 76.44 ± 0.56 | 7.79 | 74.96 ± 1.54 | 8.81 |
| FEROMA | **87.51 ± 1.19** | 4.00 | **86.45 ± 1.22** | 6.32 | **87.69 ± 0.91** | 7.88 | **83.09 ± 1.29** | 8.87 |

Table 30: Performance comparison across heterogeneity type $P(X)$ on the MNIST dataset. Time is reported in $\log_2$ seconds.

## I.4 RESULTS ON REAL-WORLD DATASETS

Following the setup in subsection I.1, we compare the performance of FEROMA against baseline methods on both the CheXpert and Office-Home datasets (baseline ATP cannot be adapted to the multi-label classification task of CheXpert dataset). As shown in Table 34 and Table 35, FEROMA consistently achieves top-tier performance across various settings of non-IID severity and degrees of distribution drift. Specifically, it either outperforms or closely matches the best-performing methods in nearly all configurations. While personalized methods such as ATP and APFL occasionally yield strong results—particularly in low-drift scenarios, their performance tends to degrade under increasing drift levels, indicating overfitting to local data. In contrast, FEROMA maintains stable and high performance even in highly drifted or non-IID environments, demonstrating strong robustness and generalization without relying on explicit personalization.

| # Clients | 10 Clients | | 20 Clients | | 50 Clients | | 100 Clients | |
|---|---|---|---|---|---|---|---|---|
| Algorithm | Accuracy | Time | Accuracy | Time | Accuracy | Time | Accuracy | Time |
| FedAvg | 85.98 ± 6.45 | **4.34** | 84.82 ± 4.58 | **6.19** | 80.64 ± 3.01 | **7.30** | 84.38 ± 6.68 | **8.41** |
| FedRC | 68.89 ± 10.89 | 8.88 | 75.08 ± 5.67 | 10.32 | 63.79 ± 5.65 | 11.52 | 60.82 ± 6.26 | 12.25 |
| FedEM | 72.31 ± 9.84 | 8.85 | 66.08 ± 5.81 | 9.42 | 67.11 ± 5.95 | 11.19 | 62.14 ± 4.71 | 11.87 |
| FeSEM | 78.62 ± 6.58 | 7.44 | 72.37 ± 3.53 | 7.97 | 75.11 ± 7.31 | 9.19 | 77.53 ± 5.33 | 9.88 |
| CFL | 81.56 ± 12.51 | 6.89 | 85.21 ± 5.86 | 7.03 | 81.90 ± 6.52 | 8.96 | 82.39 ± 5.34 | 9.05 |
| IFCA | 36.77 ± 17.81 | 7.10 | 35.27 ± 10.01 | 8.20 | 37.22 ± 3.98 | 10.05 | 26.81 ± 6.73 | 10.74 |
| pFedMe | 38.59 ± 18.23 | 6.06 | 39.32 ± 13.26 | 7.00 | 34.31 ± 4.61 | 8.10 | 25.16 ± 7.59 | 9.53 |
| APFL | 53.90 ± 13.88 | 7.68 | 56.56 ± 8.56 | 8.70 | 55.12 ± 6.09 | 10.60 | 47.32 ± 8.51 | 10.60 |
| FedDrift | 45.60 ± 16.15 | 8.87 | 43.46 ± 14.96 | 9.69 | N/A | N/A | N/A | N/A |
| ATP | 80.42 ± 10.16 | 4.93 | 86.25 ± 7.72 | 6.62 | 81.29 ± 8.67 | 7.91 | 74.85 ± 8.51 | 8.79 |
| FEROMA | **99.12 ± 0.37** | 4.84 | **95.75 ± 3.62** | 6.54 | **94.77 ± 3.74** | 7.73 | **89.01 ± 5.03** | 8.71 |

Table 31: Performance comparison across heterogeneity type $P(Y)$ on the MNIST dataset. Time is reported in $\log_2$ seconds.

| # Clients | 10 Clients | | 20 Clients | | 50 Clients | | 100 Clients | |
|---|---|---|---|---|---|---|---|---|
| Algorithm | Accuracy | Time | Accuracy | Time | Accuracy | Time | Accuracy | Time |
| FedAvg | 66.16 ± 1.87 | **3.45** | 65.75 ± 1.63 | **5.88** | 63.30 ± 0.91 | **7.20** | 62.92 ± 0.69 | 8.37 |
| FedRC | 13.84 ± 3.24 | 8.98 | 12.91 ± 2.29 | 9.87 | 10.50 ± 0.42 | 11.90 | 11.48 ± 1.49 | 13.45 |
| FedEM | 13.84 ± 3.24 | 8.85 | 12.91 ± 2.29 | 9.81 | 10.50 ± 0.42 | 12.22 | 11.49 ± 1.49 | 13.11 |
| FeSEM | 81.24 ± 2.17 | 7.17 | 75.43 ± 1.55 | 7.90 | 73.89 ± 1.31 | 9.84 | 70.97 ± 0.67 | 10.67 |
| CFL | 68.73 ± 1.90 | 5.35 | 67.26 ± 1.23 | 6.25 | 65.71 ± 0.75 | 8.14 | 64.87 ± 0.83 | 8.72 |
| IFCA | 69.66 ± 1.58 | 6.90 | 69.70 ± 4.02 | 8.84 | 70.14 ± 4.59 | 9.95 | 69.12 ± 5.36 | 11.32 |
| pFedMe | 91.31 ± 0.32 | 5.32 | 90.95 ± 0.47 | 6.17 | 90.01 ± 0.36 | 7.38 | 88.39 ± 0.46 | **8.32** |
| APFL | 91.85 ± 0.47 | 7.64 | 91.61 ± 0.68 | 8.36 | **91.05 ± 0.35** | 9.46 | **90.09 ± 0.62** | 9.99 |
| FedDrift | 95.56 ± 1.48 | 8.53 | **96.59 ± 0.26** | 9.41 | N/A | N/A | N/A | N/A |
| ATP | 66.40 ± 1.23 | 5.39 | 64.67 ± 2.82 | 6.28 | 63.24 ± 1.45 | 7.50 | 61.57 ± 1.53 | 8.56 |
| FEROMA | 88.81 ± 0.40 | 5.74 | 88.59 ± 0.62 | 6.24 | 88.40 ± 0.47 | 7.44 | 88.10 ± 0.57 | 8.44 |

Table 32: Performance comparison across heterogeneity type $P(Y|X)$ on the MNIST dataset. Time is reported in $\log_2$ seconds.

| # Clients | 10 Clients | | 20 Clients | | 50 Clients | | 100 Clients | |
|---|---|---|---|---|---|---|---|---|
| Algorithm | Accuracy | Time | Accuracy | Time | Accuracy | Time | Accuracy | Time |
| FedAvg | 73.04 ± 8.44 | **3.57** | 70.71 ± 3.32 | **5.76** | 77.57 ± 5.02 | **6.98** | 74.72 ± 1.54 | **8.13** |
| FedRC | 25.20 ± 9.70 | 8.13 | 18.43 ± 8.77 | 9.34 | 14.43 ± 5.46 | 10.80 | 12.57 ± 2.44 | 11.29 |
| FedEM | 30.45 ± 13.63 | 8.16 | 16.79 ± 6.72 | 8.77 | 14.85 ± 6.14 | 10.71 | 14.10 ± 3.52 | 11.10 |
| FeSEM | 62.78 ± 10.71 | 6.84 | 69.53 ± 7.16 | 7.26 | 68.90 ± 1.84 | 8.53 | 73.22 ± 3.80 | 9.39 |
| CFL | 78.74 ± 4.27 | 6.16 | 79.31 ± 5.43 | 6.49 | 81.35 ± 4.08 | 8.46 | 80.69 ± 1.53 | 8.64 |
| IFCA | 19.49 ± 8.37 | 6.47 | 39.20 ± 15.89 | 7.55 | 38.37 ± 5.22 | 8.98 | 34.44 ± 4.84 | 10.04 |
| pFedMe | 38.00 ± 16.00 | 5.57 | 44.92 ± 9.61 | 6.44 | 42.67 ± 3.61 | 7.64 | 38.31 ± 6.57 | 8.89 |
| APFL | 67.09 ± 7.75 | 7.16 | 68.95 ± 4.98 | 8.41 | 71.71 ± 5.63 | 9.94 | 63.73 ± 4.93 | 10.12 |
| FedDrift | 52.08 ± 15.30 | 8.79 | 44.62 ± 7.23 | 9.58 | N/A | N/A | N/A | N/A |
| ATP | 67.67 ± 12.49 | 5.17 | 72.62 ± 1.79 | 6.67 | 68.87 ± 2.28 | 7.73 | 73.47 ± 6.46 | 8.53 |
| FEROMA | **89.64 ± 0.29** | 4.05 | **89.90 ± 0.63** | 6.60 | **89.77 ± 0.82** | 7.60 | **88.04 ± 1.41** | 8.37 |

Table 33: Performance comparison across heterogeneity type $P(X|Y)$ on the MNIST dataset. Time is reported in $\log_2$ seconds.

| Non-IID Level | Low | | | Medium | | | High | | |
|---|---|---|---|---|---|---|---|---|---|
| # Drifting | 5 / 20 | 10 / 20 | 20 / 20 | 5 / 20 | 10 / 20 | 20 / 20 | 5 / 20 | 10 / 20 | 20 / 20 |
| FedAvg | 59.50 ± 1.04 | 62.07 ± 6.04 | 57.68 ± 2.05 | 60.91 ± 1.19 | 59.34 ± 5.33 | 61.26 ± 1.25 | 57.38 ± 1.17 | 56.76 ± 2.00 | 56.62 ± 1.31 |
| FedRC | 57.07 ± 3.12 | 58.80 ± 0.15 | 56.77 ± 0.19 | 54.38 ± 0.32 | 55.23 ± 2.43 | 55.79 ± 3.43 | 52.76 ± 1.24 | 53.55 ± 1.54 | 51.46 ± 1.23 |
| FedEM | 56.31 ± 1.27 | 55.12 ± 0.24 | 56.54 ± 3.42 | 53.58 ± 0.32 | 53.45 ± 2.34 | 52.44 ± 1.23 | 50.45 ± 2.34 | 51.34 ± 3.34 | 50.44 ± 2.32 |
| FeSEM | 64.09 ± 3.63 | 63.76 ± 0.05 | 64.45 ± 0.18 | 59.33 ± 0.29 | 60.76 ± 0.18 | 61.32 ± 0.14 | 59.23 ± 2.43 | 58.55 ± 1.35 | 57.54 ± 3.44 |
| CFL | 65.09 ± 5.21 | 64.85 ± 0.07 | 63.73 ± 0.24 | 62.34 ± 0.29 | 62.21 ± 0.22 | 60.31 ± 0.16 | 60.44 ± 0.45 | 61.43 ± 0.83 | 60.43 ± 3.91 |
| IFCA | 55.94 ± 0.74 | 55.12 ± 1.32 | 54.31 ± 4.35 | 52.54 ± 1.54 | 52.00 ± 2.43 | 51.33 ± 2.78 | 51.14 ± 0.92 | 50.88 ± 3.44 | 51.54 ± 4.53 |
| pFedMe | 61.70 ± 0.64 | 61.12 ± 2.78 | 58.68 ± 1.46 | 59.92 ± 0.31 | 57.56 ± 0.14 | 57.88 ± 0.11 | 55.54 ± 0.04 | 55.48 ± 0.03 | 56.07 ± 0.28 |
| APFL | 56.35 ± 1.23 | 56.90 ± 0.06 | 55.90 ± 0.18 | 54.78 ± 1.72 | 54.83 ± 0.53 | 53.55 ± 1.35 | 53.78 ± 0.28 | 52.83 ± 0.12 | 51.00 ± 1.02 |
| FedDrift | 75.79 ± 0.24 | 74.86 ± 0.20 | 74.03 ± 0.60 | 71.70 ± 0.64 | 72.02 ± 0.48 | 71.91 ± 0.20 | 70.47 ± 0.94 | 70.48 ± 1.59 | 69.11 ± 1.31 |
| ATP | N/A | N/A | N/A | N/A | N/A | N/A | N/A | N/A | N/A |
| FEROMA | **76.32 ± 0.05** | **75.99 ± 0.14** | **76.64 ± 0.72** | **72.16 ± 0.67** | 71.96 ± 0.15 | 71.82 ± 0.69 | 69.35 ± 0.85 | 68.90 ± 0.70 | 68.59 ± 0.75 |

Table 34: Performance comparison across three different non-IID Levels and three distribution drifting levels on the CheXpert dataset.

| # Drifting | 5 / 20 | 10 / 20 | 20 / 20 |
|---|---|---|---|
| FedAvg | 41.83 ± 1.44 | 40.11 ± 1.42 | 41.02 ± 0.89 |
| FedRC | 13.83 ± 5.30 | 13.32 ± 1.44 | 16.63 ± 2.50 |
| FedEM | 14.32 ± 4.30 | 14.86 ± 0.50 | 17.43 ± 2.50 |
| FeSEM | 35.47 ± 0.17 | 33.60 ± 0.86 | 32.40 ± 1.31 |
| CFL | 36.27 ± 0.33 | 33.93 ± 0.77 | 34.33 ± 1.60 |
| IFCA | 40.40 ± 4.54 | 37.17 ± 2.12 | 22.83 ± 5.90 |
| pFedMe | 37.70 ± 1.04 | 34.53 ± 2.05 | 31.70 ± 1.53 |
| APFL | **44.19 ± 2.98** | 39.99 ± 0.50 | 34.08 ± 2.20 |
| FedDrift | 42.08 ± 3.94 | 42.34 ± 0.43 | **41.94 ± 1.12** |
| ATP | 41.90 ± 3.13 | 40.88 ± 4.29 | 39.55 ± 5.26 |
| FEROMA | 43.43 ± 0.86 | **42.75 ± 1.21** | 41.11 ± 1.95 |

Table 35: Performance comparison across three distribution drifting levels on the Office-Home dataset.

## I.5 COMPARISON WITH NON-FL TEST-TIME ADAPTATION APPROACH

In this section, we compare FEROMA with CoLA (Chen et al., 2024a), which performs collaborative test-time adaptation by sharing compact domain-knowledge vectors across devices and updating each client using lightweight similarity-based or reprogramming-based adjustments to handle domain shifts during inference. Despite that CoLA is not originally proposed for FL, we include it as a strong baseline as its design aligns with FEROMA's ability to handle test-time distribution changes.

| Non-IID Level | Low | | | Medium | | | High | | |
|---|---|---|---|---|---|---|---|---|---|
| # Drifting | 5 / 20 | 10 / 20 | 20 / 20 | 5 / 20 | 10 / 20 | 20 / 20 | 5 / 20 | 10 / 20 | 20 / 20 |
| CoLA | 89.0 ± 0.2 | 88.3 ± 0.3 | 88.5 ± 0.5 | 82.6 ± 4.1 | 82.2 ± 1.9 | 84.9 ± 0.3 | 80.4 ± 4.1 | 84.2 ± 0.3 | 84.6 ± 0.6 |
| FedAvg | 66.5 ± 1.2 | 67.9 ± 1.5 | 72.1 ± 3.5 | 73.2 ± 4.0 | 73.7 ± 2.3 | 73.2 ± 1.8 | 54.8 ± 1.8 | 54.3 ± 2.7 | 58.5 ± 5.9 |
| FEROMA | 88.9 ± 0.3 | 88.7 ± 0.3 | 89.5 ± 0.4 | 87.1 ± 2.3 | 86.5 ± 2.5 | 86.1 ± 3.4 | 86.4 ± 0.2 | 85.8 ± 0.4 | 85.7 ± 1.2 |

Table 36: Performance comparison among CoLA, FedAvg, and FEROMA with heterogeneity type $P(X)$, across three different non-IID Levels and three distribution drifting levels.

The experiment settings are the same as reported in our main experiments. As shown from Table 36 to Table 39, CoLA provides strong test-time robustness due to its domain-adaptation mechanism, but it does not address training-time distribution drift; consequently, while its performance is competitive, it does not surpass FEROMA, which handles both drift and shift during training.

## I.6 GENERALIZABILITY OF DPE TO TIME-SERIES DATA

FEROMA does not make any assumption about the underlying data modality. Instead, it relies on the DPE as an abstraction layer that compresses potentially high-dimensional, multi-modal, or long-tailed client data into a compact distributional representation. As long as a suitable feature extractor exists

| Non-IID Level | Low | | | Medium | | | High | | |
|---|---|---|---|---|---|---|---|---|---|
| # Drifting | 5 / 20 | 10 / 20 | 20 / 20 | 5 / 20 | 10 / 20 | 20 / 20 | 5 / 20 | 10 / 20 | 20 / 20 |
| CoLA | 94.7 ± 3.9 | 98.7 ± 0.5 | 99.2 ± 1.5 | 94.2 ± 4.7 | 96.3 ± 2.6 | 99.0 ± 0.0 | 94.0 ± 0.3 | 90.6 ± 7.6 | 96.7 ± 2.4 |
| FedAvg | 75.1 ± 12.4 | 83.8 ± 8.8 | 94.6 ± 1.6 | 74.5 ± 5.8 | 82.8 ± 9.4 | 90.8 ± 1.4 | 70.4 ± 11.9 | 74.7 ± 11.6 | 89.4 ± 5.8 |
| FEROMA | 96.5 ± 3.2 | 98.7 ± 0.4 | 99.4 ± 0.3 | 97.1 ± 1.8 | 97.9 ± 2.0 | 98.7 ± 1.1 | 96.9 ± 1.9 | 97.2 ± 3.0 | 99.3 ± 0.3 |

Table 37: Performance comparison among CoLA, FedAvg, and FEROMA with heterogeneity type $P(Y)$, across three different non-IID Levels and three distribution drifting levels.

| Non-IID Level | Low | | | Medium | | | High | | |
|---|---|---|---|---|---|---|---|---|---|
| # Drifting | 5 / 20 | 10 / 20 | 20 / 20 | 5 / 20 | 10 / 20 | 20 / 20 | 5 / 20 | 10 / 20 | 20 / 20 |
| CoLA | 84.5 ± 2.6 | 85.4 ± 1.5 | 82.9 ± 0.4 | 76.5 ± 3.8 | 80.2 ± 1.3 | 77.1 ± 4.0 | 78.3 ± 1.8 | 78.9 ± 0.4 | 77.8 ± 1.3 |
| FedAvg | 73.5 ± 1.6 | 72.9 ± 0.8 | 72.9 ± 1.0 | 64.9 ± 1.7 | 65.0 ± 2.1 | 64.2 ± 2.0 | 57.1 ± 2.1 | 57.2 ± 2.0 | 56.4 ± 1.1 |
| FEROMA | 89.6 ± 0.5 | 89.5 ± 0.5 | 89.8 ± 0.8 | 88.9 ± 0.5 | 88.9 ± 0.7 | 89.0 ± 0.6 | 88.2 ± 0.7 | 88.1 ± 0.5 | 88.4 ± 0.7 |

Table 38: Performance comparison among CoLA, FedAvg, and FEROMA with heterogeneity type $P(Y \mid X)$, across three different non-IID Levels and three distribution drifting levels.

for a given modality, FEROMA can operate on top of it without modification to the aggregation or mapping mechanisms.

In the main experiments, we instantiate the DPE using few-round trained models for image-based tasks, as this provides a practical and lightweight way to capture distributional structure while remaining easy to integrate into standard FL pipelines. Importantly, this choice is not intrinsic to FEROMA. The DPE design is intentionally model-agnostic and can be instantiated with alternative feature extractors tailored to different data types.

To evaluate the generalizability of the DPE abstraction beyond image data, we adapt FEROMA to a time-series classification task using the UCI Human Activity Recognition (UCI-HAR) (Reyes-Ortiz et al., 2013) dataset. This dataset consists of multi-dimensional sensor signals and exhibits strong temporal dependencies and multi-modal feature characteristics. Due to its relatively small scale, we set the number of clients to 10 and avoid data augmentation in order to preserve its real-world structure. Client heterogeneity is introduced via Dirichlet sampling over label distributions with concentration parameter $\alpha$, where smaller $\alpha$ corresponds to stronger non-IIDness in $P(Y)$.

Table 40 reports the results across different heterogeneity levels. FEROMA consistently outperforms all baselines by a large margin under severe heterogeneity, and remains competitive as heterogeneity decreases. These results indicate that distribution-profile-based aggregation remains effective even when client feature spaces are time-dependent and multi-modal, and that the DPE abstraction is not limited to image-based representations.

## I.7 FEROMA ROBUSTNESS AGAINST UNSEEN OR OUT-OF-SUPPORT DISTRIBUTIONS

FEROMA is not explicitly designed to handle completely unseen or out-of-support data distributions. Nevertheless, robustness to such scenarios is implicitly evaluated throughout our experimental setup. In the drifting dataset generation process, test-time distributions may differ from those observed during training, particularly under high non-IID severity and low drifting frequency (e.g., non-IID

| Non-IID Level | Low | | | Medium | | | High | | |
|---|---|---|---|---|---|---|---|---|---|
| # Drifting | 5 / 20 | 10 / 20 | 20 / 20 | 5 / 20 | 10 / 20 | 20 / 20 | 5 / 20 | 10 / 20 | 20 / 20 |
| CoLA | 86.7 ± 0.2 | 89.0 ± 1.7 | 85.9 ± 3.3 | 86.1 ± 1.3 | 82.6 ± 7.2 | 84.3 ± 0.4 | 88.1 ± 0.9 | 79.2 ± 6.9 | 88.9 ± 0.6 |
| FedAvg | 73.4 ± 9.8 | 81.6 ± 2.9 | 77.7 ± 3.3 | 73.3 ± 4.0 | 73.1 ± 4.1 | 75.0 ± 4.1 | 71.6 ± 5.1 | 74.3 ± 5.1 | 69.9 ± 9.8 |
| FEROMA | 87.3 ± 4.8 | 88.8 ± 1.9 | 89.9 ± 2.1 | 87.0 ± 4.5 | 89.0 ± 1.6 | 90.1 ± 0.8 | 89.4 ± 1.2 | 88.3 ± 1.1 | 89.7 ± 0.6 |

Table 39: Performance comparison among CoLA, FedAvg, and FEROMA with heterogeneity type $P(X \mid Y)$, across three different non-IID Levels and three distribution drifting levels.

| $\alpha$ | FedAvg | FEROMA | FedRC | FedEM | FeSEM | CFL | IFCA | pFedMe | APFL | FedDrift | ATP |
|---|---|---|---|---|---|---|---|---|---|---|---|
| 0.4 | $37.3 \pm 4.4$ | $61.5 \pm 3.3$ | $39.9 \pm 4.8$ | $39.9 \pm 4.9$ | $24.2 \pm 6.4$ | $38.5 \pm 7.8$ | $34.0 \pm 3.7$ | $26.5 \pm 9.1$ | $35.5 \pm 3.5$ | $32.0 \pm 2.8$ | $38.3 \pm 4.3$ |
| 0.5 | $35.1 \pm 3.5$ | $53.9 \pm 2.0$ | $39.1 \pm 5.2$ | $39.1 \pm 5.3$ | $35.5 \pm 11.2$ | $39.3 \pm 7.1$ | $37.3 \pm 3.2$ | $25.6 \pm 7.5$ | $42.4 \pm 8.5$ | $37.9 \pm 29.8$ | $40.4 \pm 5.5$ |
| 0.6 | $36.9 \pm 5.0$ | $58.0 \pm 5.2$ | $40.5 \pm 4.2$ | $40.5 \pm 4.3$ | $34.8 \pm 7.5$ | $40.1 \pm 3.6$ | $39.0 \pm 5.3$ | $30.7 \pm 8.6$ | $46.2 \pm 9.6$ | $38.9 \pm 7.7$ | $41.3 \pm 8.3$ |
| 0.7 | $37.7 \pm 1.6$ | $50.9 \pm 7.9$ | $41.1 \pm 4.5$ | $41.2 \pm 4.5$ | $38.1 \pm 5.0$ | $44.6 \pm 7.1$ | $40.4 \pm 5.6$ | $30.9 \pm 5.3$ | $46.6 \pm 5.5$ | $38.3 \pm 10.6$ | $43.9 \pm 6.5$ |
| 0.8 | $43.0 \pm 3.5$ | $53.4 \pm 2.5$ | $42.9 \pm 3.7$ | $43.0 \pm 3.8$ | $40.1 \pm 6.7$ | $41.5 \pm 3.3$ | $42.0 \pm 3.3$ | $36.2 \pm 7.9$ | $50.0 \pm 7.6$ | $39.1 \pm 8.2$ | $45.1 \pm 3.3$ |

Table 40: Performance on the UCI-HAR time-series dataset under varying label distribution heterogeneity. Smaller $\alpha$ indicates stronger non-IIDness in $P(Y)$.

level *High* with drifting $5/20$ in Table 2). These cases, analyzed in Appendix B, naturally include test distributions that are not present in recent training rounds.

To more explicitly evaluate robustness against truly unseen distributions, we conduct an additional experiment on MNIST with 20 clients under medium non-IID conditions. During testing, we vary the proportion of clients whose data distributions are unseen during training, and report the final test accuracy averaged over drifting frequencies 5, 10, and 20 (out of 20 total rounds).

Table 41 reports the results. When no clients encounter unseen distributions, FEROMA achieves its standard baseline performance. As the unseen rate increases, FedAvg exhibits a steady degradation in accuracy. In contrast, FEROMA maintains strong performance even in extreme settings where all clients face unseen test distributions, indicating that profile-based model assignment and aggregation generalize beyond the training support.

| Unseen Rate | 0% | 20% | 40% | 60% | 80% | 100% |
|---|---|---|---|---|---|---|
| FedAvg | $73.25 \pm 4.45$ | $69.12 \pm 2.84$ | $65.20 \pm 0.76$ | $62.60 \pm 0.31$ | $59.59 \pm 0.78$ | $53.05 \pm 4.17$ |
| FEROMA | $86.70 \pm 1.56$ | $86.12 \pm 1.28$ | $85.79 \pm 1.39$ | $85.10 \pm 0.09$ | $84.98 \pm 0.05$ | $84.80 \pm 0.12$ |

Table 41: Robustness to unseen test distributions on MNIST with 20 clients under medium non-IID conditions. Results are averaged over drifting frequencies 5, 10, and 20 (out of 20 rounds).

Overall, these results indicate that although FEROMA does not explicitly model out-of-support distributions, its distribution-profile-based aggregation and model assignment mechanisms provide strong empirical robustness under severe test-time distribution shifts.

## I.8 ABLATION STUDY ON MONTE-CARLO SUBSAMPLING

The Distribution Profile Extractor (DPE) must satisfy Requirement (R3) on controlled stochasticity. As described in Section B.4.1, our implementation enforces this property via a Monte Carlo subsampling strategy. Specifically, for each client, we apply $M$ independent Bernoulli masks with sampling probability $\gamma = 0.5$, and compute distribution profiles over the resulting subsets. The variance introduced by this process is governed by the number of subsamples $M$, the number of used data points $v^{(k)}$, and the proxy variance $\tau^2$ of the latent coordinates, as derived in Equation 2. Among these parameters, $M$ and $\gamma$ are design choices under our control. In this ablation, we study the impact of varying $M \in \{1, 2, 3\}$, keeping $\gamma = 0.5$ fixed, across all four types of distribution shifts ($P(X)$, $P(Y)$, $P(Y|X)$, $P(X|Y)$) and three heterogeneity levels (low, medium, high), on both MNIST and CIFAR-10.

The results, reported in Tables 42–45, show that FEROMA is robust to the choice of $M$, with only minor accuracy differences across values. Nevertheless, slightly better performance is observed with larger $M$, suggesting that reducing the variance of the extracted profiles improves model association and final accuracy. This supports the intuition that while stochasticity is essential for privacy and regularization, excessive noise may degrade the reliability of distribution similarity computations.

## I.9 ABLATION STUDY ON THRESHOLD PROFILE ASSOCIATION

To prevent noisy or weakly related distributions from influencing model aggregation, we apply a thresholding mechanism to the profile similarity weights. Specifically, a threshold $\tau$ is introduced to discard low-similarity associations, thereby promoting aggregation only among clients with suffi-

ciently aligned distributions. This design is motivated by clustered FL principles, where collaboration is restricted to similar clients, but implemented here in a soft and data-driven manner. The thresholding step is applied after computing the similarity-based weights $w_t^{(k,j)}$, as described in Equation 4. In the ablation study, we set the threshold $\tau$ to the average value of the similarity weights (e.g., 0.05 in the case of 20 clients), providing a simple heuristic to evaluate the effect of thresholding.

From Table 42 to Table 45, the results show that accuracy remains largely stable with thresholding enabled or disabled. In future work, we expect that more principled or adaptive strategies for selecting the threshold $\tau$ could further enhance aggregation quality—particularly in highly imbalanced or noisy distribution settings.

| Non-IID Type | $P(X)$ | | | $P(Y)$ | | |
|---|---|---|---|---|---|---|
| Non-IID Level | Low | Medium | High | Low | Medium | High |
| # $M = 1$, threshold off | 87.88 ± 1.07 | 87.29 ± 1.48 | 85.30 ± 1.34 | 84.56 ± 9.61 | 79.69 ± 10.61 | 79.85 ± 10.41 |
| # $M = 2$, threshold off | 88.10 ± 0.88 | 87.55 ± 1.61 | 85.89 ± 0.30 | 87.61 ± 6.53 | 83.81 ± 8.62 | 78.91 ± 10.86 |
| # $M = 3$, threshold off | 88.67 ± 0.30 | 86.50 ± 2.48 | 85.81 ± 0.36 | 86.24 ± 8.27 | 85.94 ± 5.23 | 80.11 ± 9.65 |
| # $M = 1$, threshold on | 87.62 ± 0.98 | 87.21 ± 1.40 | 85.71 ± 0.52 | 86.05 ± 8.60 | 76.37 ± 10.30 | 73.26 ± 11.61 |
| # $M = 2$, threshold on | 87.53 ± 1.00 | 85.61 ± 4.75 | 85.48 ± 0.46 | 86.95 ± 7.53 | 81.59 ± 8.67 | 75.75 ± 7.90 |
| # $M = 3$, threshold on | 88.20 ± 0.42 | 85.98 ± 2.10 | 85.43 ± 0.27 | 86.80 ± 9.44 | 79.89 ± 8.26 | 68.67 ± 12.67 |

Table 42: Test accuracy on MNIST under varying numbers of Monte Carlo masks ($M = 1, 2, 3$) and with thresholding enabled or disabled, across different non-IID levels in $P(X)$ and $P(Y)$.

| Non-IID Type | $P(Y|X)$ | | | $P(X|Y)$ | | |
|---|---|---|---|---|---|---|
| Non-IID Level | Low | Medium | High | Low | Medium | High |
| # $M = 1$, threshold off | 84.10 ± 2.90 | 76.40 ± 3.70 | 73.88 ± 1.75 | 90.02 ± 0.47 | 88.64 ± 1.21 | 89.67 ± 1.14 |
| # $M = 2$, threshold off | 84.40 ± 3.36 | 78.71 ± 3.78 | 77.12 ± 3.07 | 88.70 ± 2.12 | 90.06 ± 0.23 | 88.53 ± 0.96 |
| # $M = 3$, threshold off | 85.04 ± 2.44 | 80.09 ± 2.14 | 78.38 ± 2.62 | 88.81 ± 1.94 | 89.02 ± 1.59 | 88.29 ± 1.12 |
| # $M = 1$, threshold on | 84.09 ± 2.87 | 76.54 ± 3.68 | 73.76 ± 1.75 | 89.38 ± 0.87 | 88.24 ± 2.52 | 88.88 ± 0.93 |
| # $M = 2$, threshold on | 84.50 ± 3.25 | 78.59 ± 3.68 | 76.99 ± 3.10 | 89.80 ± 0.41 | 88.26 ± 2.04 | 86.03 ± 3.24 |
| # $M = 3$, threshold on | 85.02 ± 2.43 | 79.90 ± 2.03 | 78.05 ± 2.57 | 90.05 ± 0.69 | 89.47 ± 1.35 | 88.86 ± 1.23 |

Table 43: Test accuracy on MNIST under varying numbers of Monte Carlo masks ($M = 1, 2, 3$) and with thresholding enabled or disabled, across different non-IID levels in $P(Y|X)$ and $P(X|Y)$.

| Non-IID Type | $P(X)$ | | | $P(Y)$ | | |
|---|---|---|---|---|---|---|
| Non-IID Level | Low | Medium | High | Low | Medium | High |
| # $M = 1$, threshold off | 40.25 ± 0.79 | 32.76 ± 2.43 | 30.97 ± 1.05 | 39.54 ± 5.29 | 34.73 ± 5.06 | 31.16 ± 3.31 |
| # $M = 2$, threshold off | 40.11 ± 0.57 | 32.41 ± 2.78 | 31.05 ± 1.73 | 38.73 ± 6.90 | 35.62 ± 5.37 | 32.24 ± 9.06 |
| # $M = 3$, threshold off | 40.13 ± 0.73 | 32.94 ± 1.78 | 31.61 ± 1.69 | 41.01 ± 7.51 | 41.33 ± 4.71 | 33.94 ± 7.60 |
| # $M = 1$, threshold on | 39.98 ± 0.58 | 31.48 ± 2.45 | 30.62 ± 1.00 | 38.81 ± 5.73 | 38.91 ± 1.38 | 30.24 ± 3.50 |
| # $M = 2$, threshold on | 40.06 ± 0.68 | 32.23 ± 2.54 | 30.54 ± 1.93 | 39.01 ± 8.81 | 35.74 ± 5.19 | 29.95 ± 8.30 |
| # $M = 3$, threshold on | 40.19 ± 0.60 | 32.10 ± 1.57 | 30.76 ± 1.28 | 39.98 ± 8.64 | 41.48 ± 3.04 | 31.64 ± 5.89 |

Table 44: Test accuracy on CIFAR-10 under varying numbers of Monte Carlo masks ($M = 1, 2, 3$) and with thresholding enabled or disabled, across different non-IID levels in $P(X)$ and $P(Y)$.

## I.10 ABLATION STUDY ON DISTANCE FUNCTION

We analyze the impact of the distance function $\mathcal{D}(\cdot, \cdot)$, introduced in Equation 3, on the overall performance of FEROMA. Specifically, we compare two commonly used similarity measures: Euclidean distance and cosine distance. To isolate their effects, we evaluate four combinations by varying the choice of $\mathcal{D}$ during training and test-time profile matching.

The evaluated settings are as follows:

• **E.E.:** Euclidean distance used in both training and testing.

| Non-IID Type | $P(Y|X)$ | | | $P(X|Y)$ | | |
|---|---|---|---|---|---|---|
| **Non-IID Level** | **Low** | **Medium** | **High** | **Low** | **Medium** | **High** |
| # $M = 1$, threshold off | 36.83 ± 1.91 | 34.16 ± 1.30 | 29.47 ± 0.50 | 41.56 ± 1.72 | 39.77 ± 1.30 | 38.94 ± 2.85 |
| # $M = 2$, threshold off | 36.24 ± 0.36 | 34.20 ± 1.47 | 30.01 ± 0.99 | 43.91 ± 1.36 | 37.93 ± 1.41 | 37.59 ± 3.65 |
| # $M = 3$, threshold off | 36.78 ± 1.74 | 34.48 ± 1.85 | 29.91 ± 1.00 | 41.25 ± 3.77 | 42.38 ± 1.82 | 38.91 ± 1.89 |
| # $M = 1$, threshold on | 36.83 ± 1.53 | 34.23 ± 1.22 | 29.64 ± 0.55 | 40.22 ± 3.31 | 40.30 ± 0.99 | 37.93 ± 1.56 |
| # $M = 2$, threshold on | 36.32 ± 0.67 | 34.49 ± 1.51 | 30.07 ± 1.22 | 41.62 ± 3.37 | 38.71 ± 2.85 | 34.21 ± 2.71 |
| # $M = 3$, threshold on | 36.94 ± 1.42 | 34.55 ± 1.72 | 29.94 ± 0.93 | 38.17 ± 2.85 | 39.04 ± 2.68 | 37.25 ± 1.51 |

Table 45: Test accuracy on CIFAR-10 under varying numbers of Monte Carlo masks ($M = 1, 2, 3$) and with thresholding enabled or disabled, across different non-IID levels in $P(Y|X)$ and $P(X|Y)$.

- **C.C.**: Cosine distance used in both training and testing.
- **E.C.**: Euclidean distance used during training; cosine distance during testing.
- **C.E.**: Cosine distance used during training; Euclidean distance during testing.

These combinations allow us to assess whether consistency between training and testing distance functions is important, and whether certain metrics generalize better under mismatch. The ablation study is conducted on the FMNIST dataset using all four non-IID types described in the main experiments. For each type, we evaluate three non-IID levels (Low, Medium, High) and three distribution drift levels (5/20, 10/20, 20/20), following the same settings as detailed in subsection I.2. Table 46 summarizes the overall performance across all combinations, and the result shows FEROMA is robust despite the choices of $\mathcal{D}$. We adopt the **C.E.** configuration (cosine distance for training and Euclidean distance for testing) for the remainder of our experiments, as it consistently achieves slightly better performance. Detailed results for each non-IID type are provided in Table 47 through Table 50.

| # Drifting | 5 / 20 | 10 / 20 | 20 / 20 |
|---|---|---|---|
| E.C. | 75.911 ± 3.12 | 76.419 ± 3.94 | 75.930 ± 5.75 |
| E.E. | 77.833 ± 3.43 | 77.079 ± 3.76 | 76.603 ± 4.59 |
| C.E. | 77.904 ± 3.42 | 77.357 ± 3.93 | 76.891 ± 4.73 |
| C.C. | 75.864 ± 3.01 | 76.302 ± 3.78 | 75.809 ± 5.75 |

Table 46: Performance comparison among different distance functions across all non-IID types and levels on the FMNIST dataset.

| Non-IID Level | Low | | | Medium | | | High | | |
|---|---|---|---|---|---|---|---|---|---|
| **# Drifting** | **5 / 20** | **10 / 20** | **20 / 20** | **5 / 20** | **10 / 20** | **20 / 20** | **5 / 20** | **10 / 20** | **20 / 20** |
| E.C. | 73.73 ± 0.48 | 74.89 ± 0.43 | 72.13 ± 0.43 | 73.76 ± 0.73 | 75.11 ± 0.67 | 71.58 ± 0.35 | 73.43 ± 0.75 | 75.09 ± 0.48 | 72.08 ± 0.62 |
| E.E. | 73.63 ± 0.53 | 74.87 ± 0.57 | 71.98 ± 0.25 | 73.69 ± 0.70 | 75.01 ± 0.63 | 71.30 ± 0.52 | 73.65 ± 0.69 | 75.01 ± 0.50 | 71.56 ± 0.77 |
| C.E. | 73.69 ± 0.46 | 74.87 ± 0.57 | 72.95 ± 0.25 | 73.70 ± 0.60 | 75.04 ± 0.70 | 71.23 ± 0.59 | 73.55 ± 0.72 | 75.01 ± 0.47 | 71.71 ± 0.62 |
| C.C. | 73.79 ± 0.41 | 74.84 ± 0.49 | 72.26 ± 0.37 | 73.83 ± 0.68 | 75.12 ± 0.60 | 71.62 ± 0.34 | 73.45 ± 0.76 | 75.03 ± 0.51 | 71.99 ± 0.65 |

Table 47: Performance comparison among different distance functions across non-IID type $P(X)$ on the FMNIST dataset.

| Non-IID Level | Low | | | Medium | | | High | | |
|---|---|---|---|---|---|---|---|---|---|
| **# Drifting** | **5 / 20** | **10 / 20** | **20 / 20** | **5 / 20** | **10 / 20** | **20 / 20** | **5 / 20** | **10 / 20** | **20 / 20** |
| E.C. | 94.11 ± 2.60 | 82.89 ± 7.27 | 83.09 ± 4.43 | 95.05 ± 3.45 | 90.20 ± 6.38 | 84.29 ± 8.43 | 94.07 ± 6.25 | 90.46 ± 6.10 | 85.09 ± 14.84 |
| E.E. | 94.49 ± 2.70 | 88.30 ± 6.23 | 86.22 ± 9.08 | 94.44 ± 3.67 | 90.22 ± 6.84 | 78.27 ± 9.74 | 93.99 ± 6.29 | 90.00 ± 5.63 | 81.07 ± 12.77 |
| C.E. | 94.46 ± 2.65 | 88.38 ± 6.34 | 86.22 ± 9.08 | 94.43 ± 3.63 | 90.33 ± 6.92 | 81.61 ± 10.57 | 94.11 ± 6.08 | 90.05 ± 5.73 | 83.92 ± 13.43 |
| C.C. | 93.82 ± 2.98 | 82.93 ± 7.32 | 82.98 ± 3.56 | 95.01 ± 3.48 | 90.10 ± 6.27 | 83.51 ± 7.60 | 93.88 ± 6.57 | 90.34 ± 5.91 | 83.97 ± 14.66 |

Table 48: Performance comparison among different distance functions across non-IID type $P(Y)$ on the FMNIST dataset.

| Non-IID Level | Low | | | Medium | | | High | | |
|---|---|---|---|---|---|---|---|---|---|
| # Drifting | 5 / 20 | 10 / 20 | 20 / 20 | 5 / 20 | 10 / 20 | 20 / 20 | 5 / 20 | 10 / 20 | 20 / 20 |
| E.C. | 72.52 ± 1.69 | 66.83 ± 2.61 | 65.28 ± 1.21 | 71.00 ± 2.71 | 69.14 ± 1.41 | 63.29 ± 1.54 | 72.41 ± 1.63 | 67.01 ± 2.00 | 63.03 ± 1.81 |
| E.E. | 73.68 ± 0.57 | 72.26 ± 1.37 | 71.24 ± 1.31 | 71.62 ± 1.63 | 72.10 ± 1.01 | 70.21 ± 2.00 | 72.55 ± 1.72 | 66.59 ± 1.82 | 62.99 ± 2.07 |
| C.E. | 73.68 ± 0.80 | 72.44 ± 1.18 | 71.54 ± 0.85 | 71.83 ± 1.70 | 72.38 ± 1.03 | 70.63 ± 1.45 | 72.46 ± 1.72 | 66.87 ± 2.14 | 64.00 ± 1.92 |
| C.C. | 72.49 ± 1.69 | 67.14 ± 1.95 | 65.11 ± 1.15 | 70.62 ± 2.79 | 68.61 ± 1.40 | 63.47 ± 1.57 | 72.48 ± 1.64 | 66.74 ± 1.66 | 63.08 ± 2.01 |

Table 49: Performance comparison among different distance functions across non-IID type $P(Y|X)$ on the FMNIST dataset.

| Non-IID Level | Low | | | Medium | | | High | | |
|---|---|---|---|---|---|---|---|---|---|
| # Drifting | 5 / 20 | 10 / 20 | 20 / 20 | 5 / 20 | 10 / 20 | 20 / 20 | 5 / 20 | 10 / 20 | 20 / 20 |
| E.C. | 78.55 ± 1.92 | 75.55 ± 3.96 | 74.62 ± 2.53 | 75.14 ± 5.05 | 77.27 ± 3.61 | 76.74 ± 3.40 | 79.02 ± 1.49 | 73.06 ± 6.13 | 70.34 ± 7.06 |
| E.E. | 78.81 ± 1.67 | 79.22 ± 1.98 | 78.65 ± 1.03 | 78.82 ± 1.43 | 78.38 ± 1.14 | 78.32 ± 1.46 | 79.33 ± 1.53 | 79.84 ± 1.76 | 78.92 ± 0.84 |
| C.E. | 78.88 ± 1.68 | 79.11 ± 1.99 | 78.22 ± 0.88 | 78.81 ± 1.39 | 78.42 ± 1.19 | 78.14 ± 1.56 | 79.55 ± 1.41 | 79.75 ± 1.83 | 78.86 ± 0.94 |
| C.C. | 78.74 ± 1.78 | 75.15 ± 3.95 | 74.21 ± 2.44 | 75.17 ± 5.05 | 76.95 ± 3.40 | 76.79 ± 3.44 | 78.98 ± 1.49 | 73.12 ± 6.37 | 70.23 ± 7.08 |

Table 50: Performance comparison among different distance functions across non-IID type $P(X|Y)$ on the FMNIST dataset.

## J ILLUSTRATIONS OF THE DYNAMIC AGGREGATION STRATEGY

This appendix visualizes the dynamic aggregation behavior of FEROMA using 20 heatmaps, each corresponding to a single client. Each heatmap shows the evolution of profile distances to other clients across training rounds, illustrating how aggregation weights adapt over time. Dynamic aggregation begins at round 6; in this round, all distances are initialized to a uniform value (0.05 for 20 clients) due to the absence of previous-round profiles for comparison.

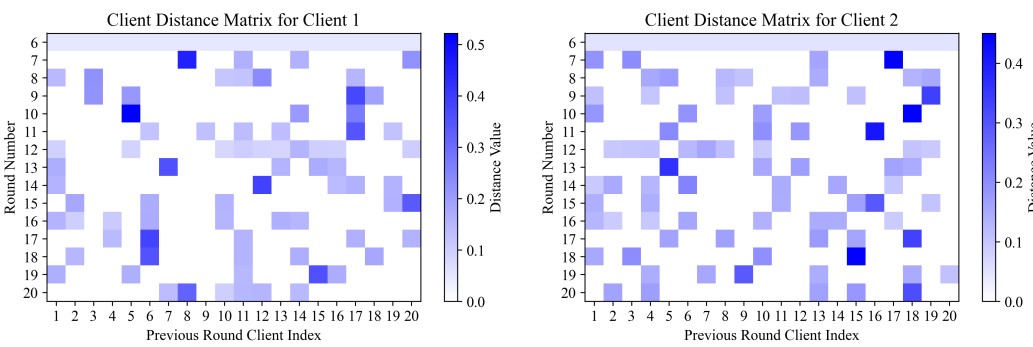

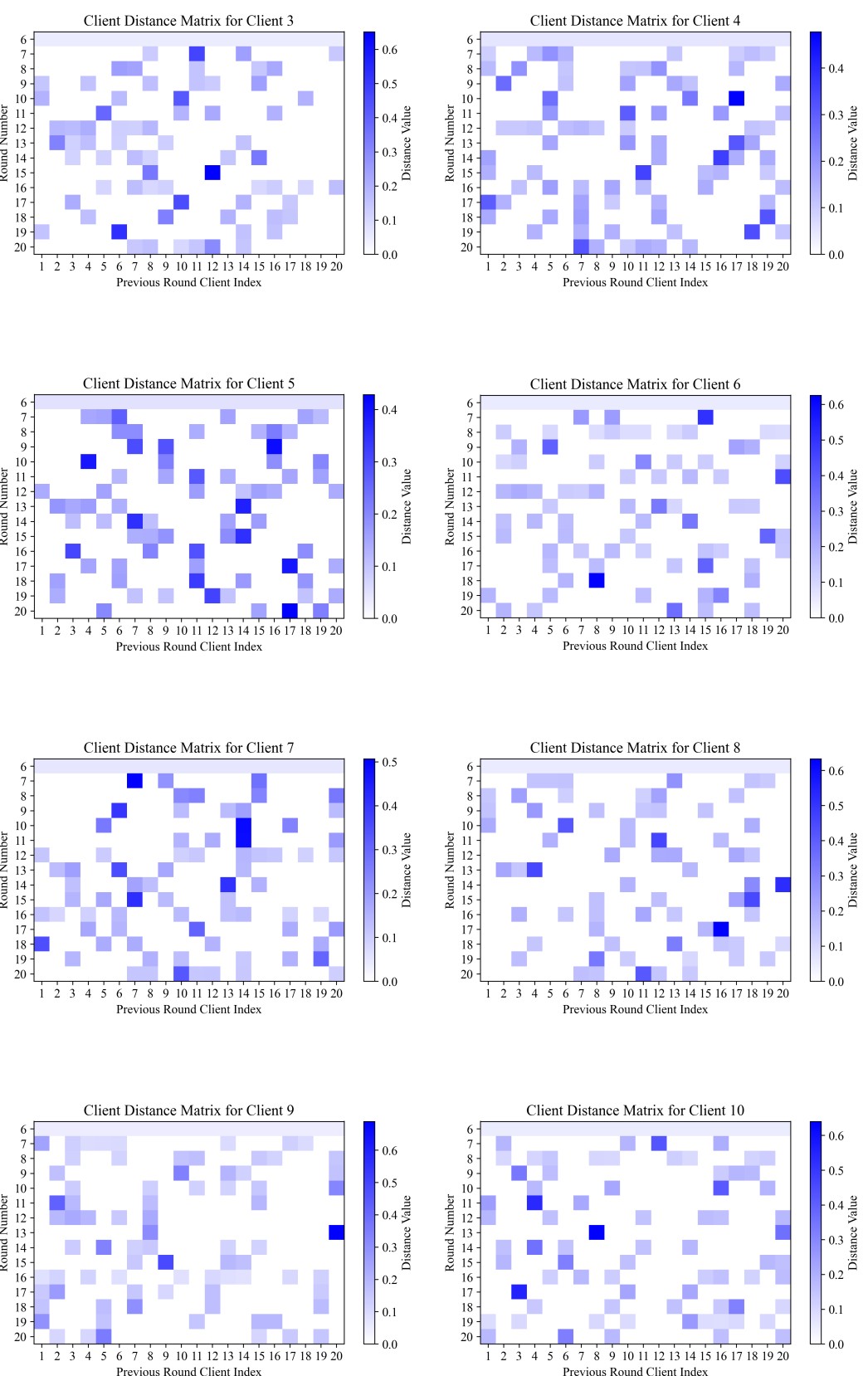

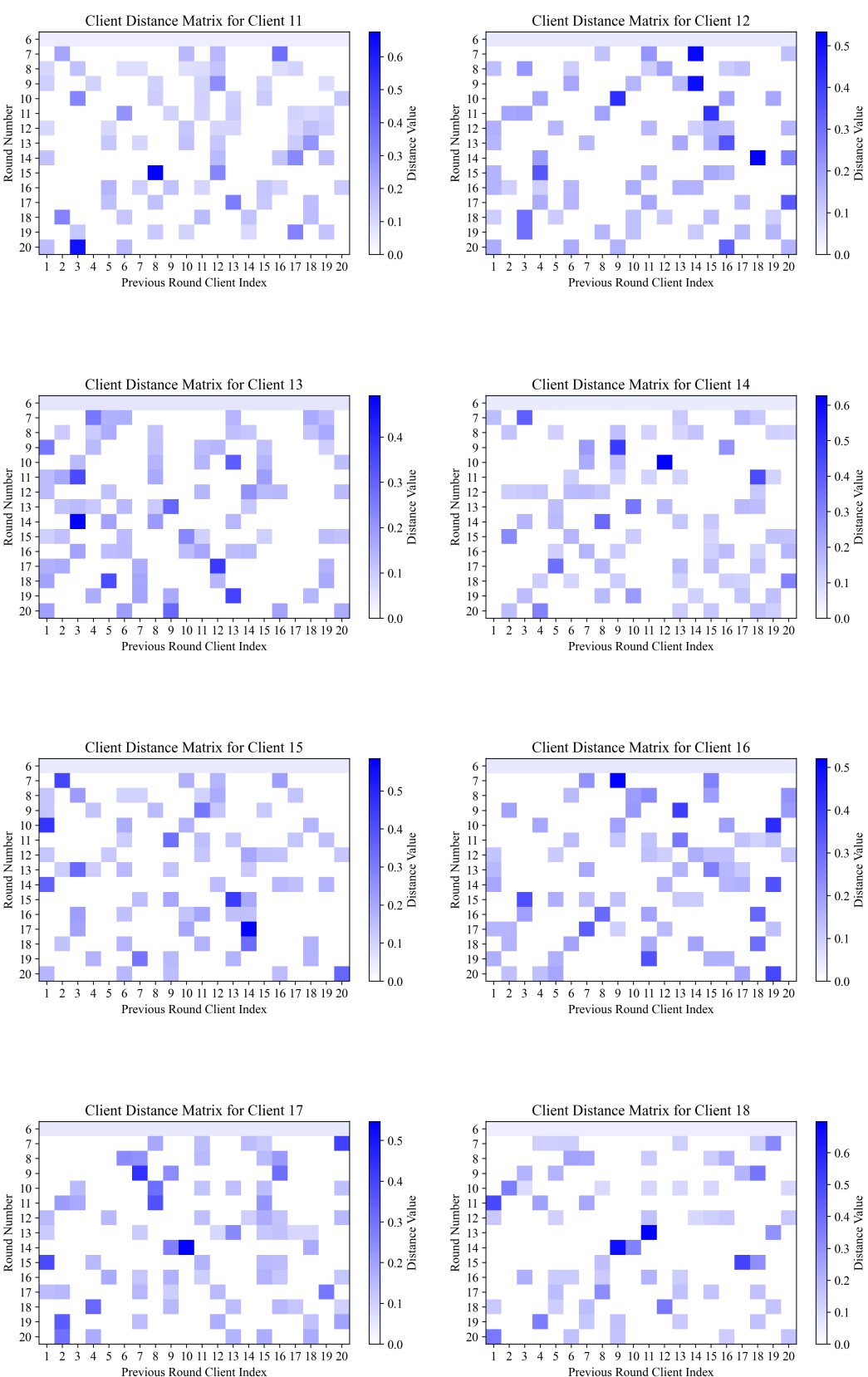

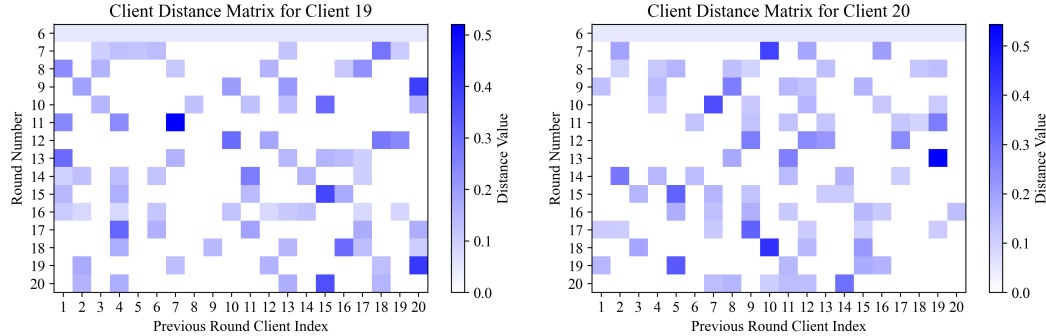

## K    EVALUATION WITH HIGHER NUMBER OF COMMUNICATION ROUNDS

This appendix reports additional experiments on CIFAR-100 with 20 clients under medium non-IID heterogeneity of type $P(X)$, using 100 and 200 communication rounds with drift introduced every 10 rounds. We plot per-client test accuracy over time to examine long-horizon training behavior. While FedAvg exhibits large performance fluctuations and unstable convergence at each drift event, FEROMA maintains steady accuracy improvements with only minor transient variations, demonstrating stable adaptation under prolonged dynamic non-IID conditions.

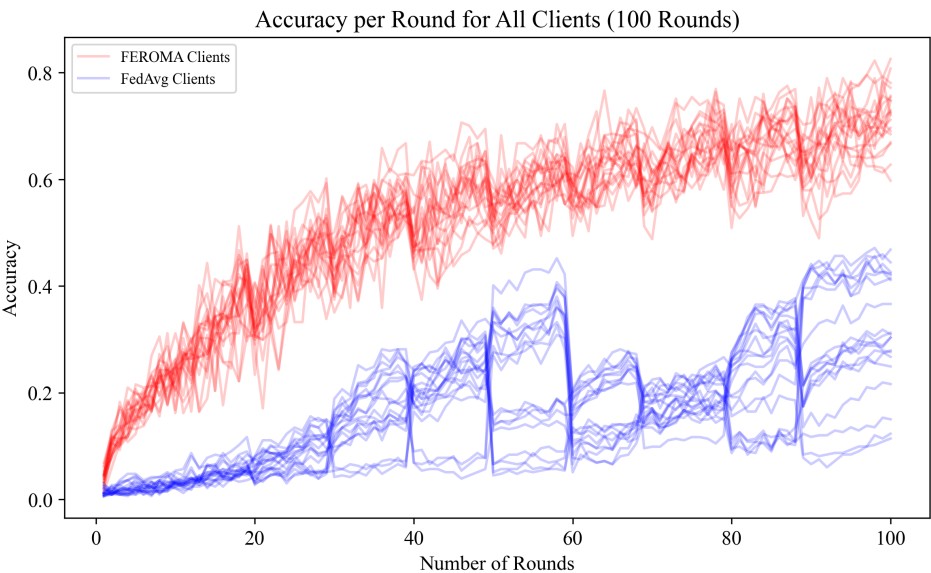

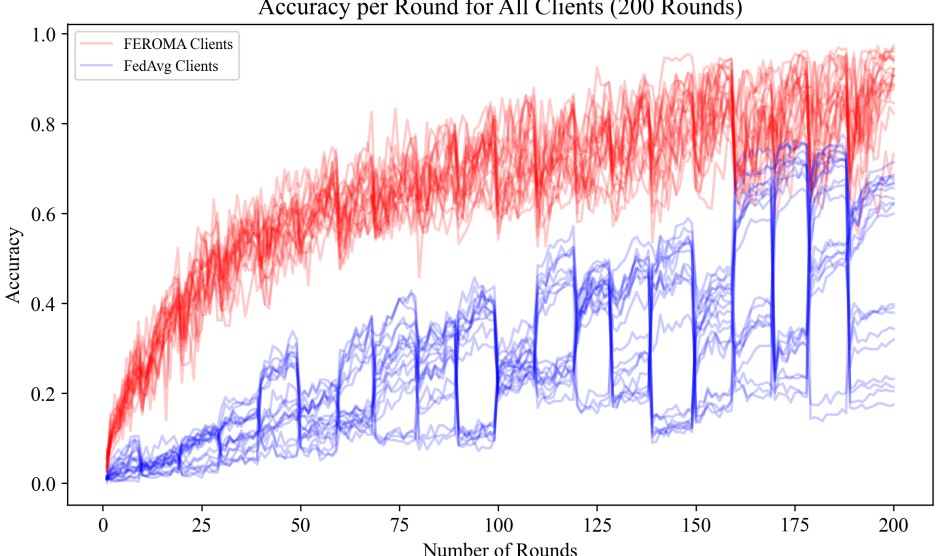

# L LARGE LANGUAGE MODEL USAGE DISCLOSURE

We disclose the use of Large Language Models (LLMs) in the preparation of this manuscript. Specifically, we used Claude (Anthropic) and GPT-4o solely for writing assistance and polishing. LLMs were used exclusively for:

1. grammar correction and sentence structure improvement,
2. clarity enhancement and readability optimization,
3. consistency in technical terminology and notation, and
4. general writing style refinement.

No content, ideas, analyses, or experimental results were generated by LLMs. All suggestions were carefully reviewed, edited, and approved by the authors before incorporation. The authors retain full responsibility for the entire content, including any errors or inaccuracies.

