# OpenReview forum: "Federated Learning with Profile Mapping under Distribution Shifts and Drifts"
_ICLR.cc/2026/Conference — ICLR 2026 Poster_

### Official Review · Reviewer_5gUD · 2025-10-20

**Soundness:** 2
**Presentation:** 3
**Contribution:** 2
**Rating:** 4
**Confidence:** 3

**Summary:**

This paper presents FEROMA, a FL framework that attempts to handle both distribution shift and drift without relying on client or cluster identity. The claimed contributions include
* a model aggregation strategy that adapts to client distribution profiles and can be extended to test-time adaptation without retraining
* a framework that works under both distribution shift and drift with minimal communication and computation  overhead, which is comparable to FedAvg
* theoretical bounds and empirical  measurements to validate the framework' practicality and privacy-preserving feature

**Strengths:**

* To my best knowledge, this is the first work that explicitly designed to address both distribution shift and distribution drift, during training as well as test time. [Disclaimer: (1) I have limited knowledge on the relevant literatures in the FL under distribution/concept drift. (2) Though the methods can be used for test-time adaptation, it's effectiveness is not fully justified.]

* The presentation is mostly clear and easy to follow and the visualizations help to understand the proposed method.

* The experiments and ablation are comprehensive in studying distribution shift and drift accross different datasets.

**Weaknesses:**

1. Usages of math notations should be correct and consistent. They should be used to deliver clear messages and supplement natural languages, instead of further creating confusion. Examples of the abuse of notations and incorrect usages can be found in minor comments, which I can infer the messages. However, there are cases where the inconsistent (or incorrect) usage of math notations making the understanding of the key algorithm designs, hence the contribution, difficult. For example, the notation $(x^{(k)}, y^{(k)})$ is first introduced in line 123, which is defined as the local dataset and said to be **fixed**. Later, in Definition 3.1, $(x^{(k)}, y^{(k)})$ is then re-defined as $(x^{(k)}, y^{(k)})\sim P(X_t^{(k)}, Y_t^{(k)})$, making the $(x^{(k)}, y^{(k)})$ a random matrix, hence making $d_t^{(k)}$ a random vector. One may argue that $(x^{(k)}, y^{(k)})$ is deterministic and the randomness of $d_t^{(k)}$ comes from the fact that the $\phi_\psi$ is a random mapping. However, that's not the point. The point is that the readers should not guess what's the authors' actual messages. This further bothers me when attempting to understand the (R1) and (R3) right after the Definition 3.1. In (R3), it clearly states that $d_t^{(k)}$ is a "random variable" (actually a random vector), however in (R1), the way the equation is presented makes $d_{t_1}^{(k_1)}$ and $d_{t_2}^{(k_2)}$ look deterministic. Given $d_t^{(k)}$ is a "random variable", it only makes sense to me when the inequality is read as "it holds with probability 1" or "it holds under the expectation with respect to distribution ...". Since this part is not clear to me, I did NOT check any mathematical proofs.

2. While understanding the page limitation, I still think it's not appropriate to omit the results on the test-time performance in the main paper and to make claims on the performance of FEROMA based on its performance on MNIST(table 2). For example, in table 2, the proposed method can easily show double digits gain over other methods nearly across the boards. However, in other tables the gain may not be significant or some baseline methods may have better performances, e.g., Table 14 (MNITS, APFL wins) , Table 18 (Cifar-10, results are mixed), and Table 24 (Cifar-100, FEDRC and FEDEM wins under low non-iid level). In contrast, Table 1 is more compelling to support some claims. It would be better if the authors could clarify what cases are used to compute the average numbers. Simply mentioning "under varying drifting frequency, non-IID types and levels" is not enough.


3. Some parameters on the experiment setups are not justified or not mentioned.
	* Unjustified parameters: For example, the communication round is set to 20. This might be okay for datasets like MNIST since it's relatively easy, hence requiring less round to get decent performances. But for relative challenging datasets like cifar100, 20 rounds may not be sufficient. Will the observations and claims remain the same, when the communication rounds are set to 100 to allow all methods to converge?

	* not mentioned: In the problem formulation, authors introduced $\mathcal{A}_{t}$ to denote active clients in round $t$. I would assume that only a fraction of client would participate in each round. But in the experiments, authors only mentioned "number of clients in 10, 20, 50, and 100". Does this mean all clients participate in all rounds?


I am willing to raise the score if my concerns are properly addressed.

**Questions:**

1. I don't understand how Global FL fallback works in Figure 3. But I guess, in the cases, $\bar w_{t}^{2,j}$ is a 0-vector. But why $\bar w_{t}^{k,j}\approx 1/|A_{t-1}|$ instead of $\bar w_{t}^{k,j}=1/|A_{t-1}|$?

2. Why there are N/As in tables for different methods under different settings, for example (ATP, CheXpert), in Table 1? There are no explanations.

3. Will the observation and claims still be similar if the communication round is set to 100 or even 200? Do all clients participate in all rounds?

4. How the FEROMA compare to other test-time methods under the testing scenario mentioned in appendix D?

5. Can the PCA map $g$ be computed on the server side? Since the random seed is fixed and the ${m^{-}, m^{+}}$ are the same for all clients, I don't see why the computation of $g$ needs to be done on the client side.


**Minor Comments**
1. The difference between solid and transparent blue/red lines are not explained.
2. Equation (1) abuse the notations of weights to be optimize and the optimal solution. Similarly abuse use of notations also happen in R(1) when referring to the distribution over the dataset and $\mathcal{A}$ to denote active clients and measurable set for DP definition. matrix of latent vectors $s^(k)_t$ and the sample size $s^{(k)}$.
3. The notation $d_t'^{(k)} \subseteq d_t^{(k)}$ does not make sense to me. Maybe the authors intend to say $d_t'^{(k)}$ is a subvector of $d_t^{(k)}$.
4. Uncommon terms "Lipschitz-equivalent" is not properly defined.
5. Figure 3. The weights for $\{d_{t}^{(i)}\}_{i=3}^{6}$ are all non-zero, which does not match the text "...exactly one weight is nonzero,...".
6. Some necessary details are deferred to the the Appendices, and one needs to jump back and forth to locate the details.

---

> ### Author Response · Authors · 2025-11-19
>
> ## Q1: On the use of random variables and notation in Def. 3.1 and (R1)–(R3).
>
> We sincerely thank the reviewer for carefully pointing out the notational inconsistencies around $(x^{(k)}, y^{(k)})$ and $d_t^{(k)}$. We fully agree that the previous notation was not sufficiently precise,
> and we appreciate the reviewer’s suggestion to clarify it. Our intended modelling is the following: for each client $𝑘$ and round $t$, there is an underlying local data distribution
> $P^{(k)}\_{t}$ over $(X,Y)$, and the client holds a finite $D^{(k)}\_t= \{(x_{t,i}^{(k)},y_{t,i}^{(k)}) \}^{s_t^{(k)}}_{i=1}$ samples from $P^{(k)}_t$. Once sampled, $D^{(k)}\_t$ is treated
> as fixed. We write $(x_t^{(k)}, y_t^{(k)})$ for the matrices collecting these samples. The DPE $𝜙_𝜓$ is a stochastic mapping (Monte Carlo subsampling + DP noise), and the profile
> $d_t^{(k)}=𝜙_𝜓(D_t^{(k)})$ is therefore a random vector whose expectation $\bar d_t^{(k)} = \mathbb{E}[d_t^{(k)} \mid D_t^{(k)} ]$ preserves the desired distributional structure.
>
> In the revised manuscript we have made this explicit and standardised the notation:
> 1. we now distinguish between the underlying distributions $P^{(k)}\_{t}$ and the finite datasets $D^{(k)}\_t=(x_t^{(k)}, y_t^{(k)})$;
> 2. Definition 3.1 defines $𝜙_𝜓$ as acting on $D^{(k)}\_t$ (a fixed dataset) rather than on a “random matrix”; and
> 3. requirement (R3) is now explicitly stated as a property of $d_t^{(k)}$ conditional on $D^{(k)}\_t$, while requirement (R1) is reformulated as an inequality in expectation over the internal
> randomness of $\phi_{\psi}$, i.e.,
> $\mathbb{E}\left[\left\lvert\lVert d_{t_1}^{(k_1)} - d_{t_2}^{(k_2)} \rVert_{2} - \Delta\left(P_{t_1}^{(k_1)}, P_{t_2}^{(k_2)}\right)\right\rvert\right] \le \xi,$
>
> We also replaced expressions such as $\Delta\bigl(P(x_{t_1}^{(k_1)}, y_{t_1}^{(k_1)}),\, P(x_{t_2}^{(k_2)}, y_{t_2}^{(k_2)})\bigr)$ by the cleaner
> $\Delta\bigl(P_{t_1}^{(k_1)}, P_{t_2}^{(k_2)}\bigr)$. These changes separates (i) fixed datasets, (ii) underlying data distributions, and (iii) the randomness introduced by the DPE, without changing the algorithm or the theoretical guarantees.
>
> We appreciate the reviewer’s feedback and are glad to further refine the notation to ensure clarity and avoid any potential sources of confusion.
>
>
>
> ## Q2: Provide more dataset-wide test-time results in the main text besides MNIST.
>
> We agree that summarizing only the MNIST results in Table 2 is insufficient to characterize FEROMA’s behavior across all benchmarks. **In the revised version, we will therefore provide a corresponding summary table for each of the six datasets, each only averaging over the four non-IID types while preserving all three drifting frequencies and all three non-IID severity levels**. This produces a dataset-specific overview that is consistent with the structure of Table 2 but avoids cross-dataset averaging, which is not a standard or meaningful practice due to differences in task difficulty and accuracy scale. To stay within page limits, each summary table will report the **top three performing methods** (rather than all baselines), with the complete results still remaining in the appendix. Below, we provide the representative table from CIFAR-10 dataset, besides all other five tables will be included in our revised manuscript.
>
> | |Low 5/20|Low 10/20|Low 20/20|Medium 5/20|Medium 10/20|Medium 20/20|High 5/20|High 10/20|High 20/20|
> | - | - | - | - | - | - | - | - | - | - |
> |FedAvg|38.3±7.6|37.4±2.7|40.3±5.4|30.7±5.7|32.2±5.2|31.7±2.5|28.6±4.6|27.3±5.6|30.3±6.0|
> |CFL|39.0±7.6|37.6±4.0|41.3±5.2|29.5±3.6|34.6±4.2|32.4±3.0|29.1±5.0|30.5±4.9|30.5±2.4|
> |FEROMA|49.0±4.5|48.3±2.7|47.8±2.8|42.5±2.9|43.6±3.7|42.7±5.8|40.0±2.6|43.3±4.2|40.4±3.5|
>
> Finally, we acknowledge the reviewer’s observation that some baselines outperform FEROMA in specific scenarios. This is expected and natural in FL, as different methods are often tailored to particular non-IID structures or operate optimally under specific conditions. Our goal is not to claim universal superiority in every isolated case, but to demonstrate consistent robustness across a wide spectrum of dynamic non-IID settings. **To ensure full transparency, we retain all detailed tables in the appendix without omission, allowing readers to inspect both the strengths and the limitations of FEROMA**. We also welcome further comparisons or additional evaluation perspectives the reviewer may find useful. Our intent is to provide a comprehensive and honest assessment rather than to obscure unfavorable cases, and we thank the reviewer again for the careful reading.

---

> ### Author Response · Authors · 2025-11-19
>
> ## Q3: Clarification on metric averaging.
>
> We thank the reviewer for this thoughtful comment on the table clarification. The main results in Table 1 are computed as **simple, unweighted averages over all experimental conditions: all three drift frequencies, all four non-IID types, and all three severity levels**. No filtering, case selection, or exclusion of unfavorable results is applied. We will explicitly state this in the revised manuscript to avoid any ambiguity.
>
> ## Q4: Clarification on participation rate.
>
> We thank the reviewer for pointing out the missing clarification in our experimental setup. In the scaling experiments reported in Section 4.2, **we indeed use a 100% participation rate**, meaning that all clients participate in every round. The notation $A_t$ in the problem formulation (and the corresponding client-selection step in the algorithm) is included to provide a unified formulation that accommodates dynamic scenarios such as partial participation, client drop-out, or cold-start arrivals. In our experiments, we chose full participation because it (i) **provides a clean and comprehensive evaluation of the methods under varying degrees of distribution shift and drift**, without conflating the effects with additional randomness from client sampling, and (ii) **meets the requirements of some other baselines which demand full participation**. We will clarify this in the revised manuscript.
>
> ## Q5: Clarification on global FL fallback.
>
> We thank the reviewer for the question regarding the Global FL fallback mechanism in Figure 3. The fallback occurs when a current-round profile has no sufficiently similar match among the previous-round profiles. In this case, all association weights become approximately uniform because the distances $D(d_t^{(k)}, d_{t-1}^{(j)})$ are all relatively large and similar. Consequently, the softmax in Eq. (3) yields weights close to $1/|A_{t-1}|$, **although not exactly equal due to distributional diversity and the stochasticity introduced by the DPE’s latent-space sampling**. When no threshold is applied, these softmax-derived weights are directly used for aggregation, effectively resulting in a FedAvg-like update.
>
> In Figure 3, profile $d_t^{(2)}$ is categorized as a global fallback case because none of the previous-round profiles is sufficiently close. This results in weights `[0.26, 0.24, 0.17, 0.16, 0.17]`, which are near, but not exactly, `0.20`, consistent with soft uniformity.
>
> To further illustrate this behavior, we conducted an additional experiment with 10 clients under medium level $P(X)$ shift, with drift occurring 10 times out of 20 rounds. As an example, we observed in round 12, the client 7 undergoes global fallback, with the following weights applied to last-round profile:
>
> `[0.0949816, 0.11802857, 0.10927976, 0.10394634, 0.08624075, 0.08623978, 0.08232334, 0.11464705, 0.1040446, 0.10026821]`
>
> These values again cluster around the ideal uniform weight `0.1`, consistent with the fallback interpretation.
>
> We also acknowledge that our original description of the dynamic strategy selection did not explicitly clarify whether thresholding is applied in each scenario. In the revised manuscript, **we will clearly specify both the thresholded and unthresholded cases** for Clustered FL, Personalized FL, and Global FL fallback.
>
> ## Q6: Explanations on missing table values.
>
> We thank the reviewer for raising this point. **The N/A entries (e.g., ATP on CheXpert in Table 1) arise because certain baseline methods cannot be directly applied to specific settings**. In particular, ATP is designed for single-label classification and its official formulation assumes a single cross-entropy loss and a single prediction head. CheXpert, however, is a multi-label medical imaging dataset requiring independent sigmoid/BCE losses for multiple labels. Adapting ATP to this setting would require substantial methodological changes beyond the scope of the original algorithm, so we report it as N/A rather than an incompatible or incorrect result.
>
> We would also like to note that this explanation was already included in our original submission (Appendix F.4). Nevertheless, we agree that this limitation should be made more explicit in the main text. In the revised manuscript, **we will clearly annotate such N/A entries in the tables and add a brief note in Section 4 to make this constraint immediately visible to readers**. We thank the reviewer again for pointing out this clarity issue.

---

> ### Author Response · Authors · 2025-11-19
>
> ## Q7: Ablation study with higher communication rounds.
>
> We thank the reviewer for raising this important point. Our choice of 20 communication rounds in the main experiments was intentional: one of our core motivations is to evaluate **convergence speed**, which is highly relevant in practical FL deployments where communication and computation budgets are limited. In such settings, achieving strong accuracy early is essential. Across all tested configurations, FEROMA consistently converges much faster than fixed-strategy baselines, often reaching high accuracy within very few rounds while other methods remain far from stabilizing.
>
> We fully agree that examining behavior at higher communication rounds is academically meaningful and provides complementary insight. Following the reviewer’s suggestion, we conducted additional experiments on CIFAR-100 with 20 clients under the medium non-IID level of type P(X), **using 100 and 200 communication rounds** (drift introduced every 10 rounds). The results are shown below:
>
> |Settings|FedAvg|FEROMA|
> |-|-|-|
> |200 rounds, drifting every 10 epochs|32.59±3.41|51.75±0.17|
> |100 rounds, drifting every 10 epochs|23.36±0.13|44.52±0.47|
> |20 rounds, drifting every 4 epochs|7.47±5.13|36.97±0.97|
>
> While both methods improve with more rounds, the performance gap between FEROMA and FedAvg remains substantial. **We further compared convergence curves and have included the corresponding figures in the updated manuscript (Appendix I: Evaluation with Higher Numbers of Communication Rounds)**. These plots confirm that FedAvg continues to exhibit unstable oscillations whenever drift occurs and struggles to converge even with significantly extended training. In contrast, FEROMA shows consistent upward progress, with only minor perturbations around drift points—a complementary per-client view that aligns with our observations on its robustness under dynamic non-IID conditions.
>
> We agree that expanding this higher-round evaluation to additional baselines would enrich the analysis. Within the limited rebuttal window, we were able to extend this study only for FedAvg vs. FEROMA. In the final revised manuscript, we will broaden this evaluation for a more comprehensive comparison. We appreciate the reviewer’s understanding regarding the practical constraints of the rebuttal phase.
>
> ## Q8: Additional baseline for testing scenario in Appendix D.
>
> We thank the reviewer for suggesting a deeper comparison to better analyze FEROMA’s behaviour under the test-time evaluation scenario. This is indeed a valuable point that strengthens the empirical assessment of FEROMA’s robustness at inference time. In addition to the FL baselines, **we have now also compared FEROMA against another representative TTA-FL method, CoLA, which is specifically designed to handle distribution shift during inference**, which also suggested by reviewer aPnB.
>
> Although CoLA is not a federated training method, its test-time update mechanism makes it directly relevant for Appendix D’s evaluation scenario, where each unseen test client must be assigned a suitable model without retraining. We implemented CoLA in our FL setting and report the results below with the scenario in Appendix D:
>
> ### Results under P(Y|X) concept shift
> |Method|low5|low10|low20|mid5|mid10|mid20|high5|high10|high20|
> |-|-|-|-|-|-|-|-|-|-|
> |CoLA|84.5±2.6|85.4±1.5|82.9±0.4|76.5±3.8|80.2±1.3|77.1±4.0|78.3±1.8|78.9±0.4|77.8±1.3|
> |FedAvg|73.5±1.6|72.9±0.8|72.9±1.0|64.9±1.7|65.0±2.1|64.2±2.0|57.1±2.1|57.2±2.0|56.4±1.1|
> |FEROMA|89.6±0.5|89.5±0.5|89.8±0.8|88.9±0.5|88.9±0.7|89.0±0.6|88.2±0.7|88.1±0.5|88.4±0.7|
>
> Our comparison with CoLA shows that while CoLA performs well under pure test-time domain shift, it cannot address training-time drift or multi-round distribution evolution. Once distributions begin to change, its test-time updates are insufficient, and FEROMA consistently performs better. Separately, Appendix D highlights a broader limitation of TTA-FL methods: **inference-time labels are generally unavailable, making true test-time association under P(Y|X) shift unrealistic. Most existing TTA-FL approaches therefore rely on assumptions that do not hold in practical FL deployments**.
>
> FEROMA offers a more feasible alternative. Its label-free association already performs competitively, and—as shown in Appendix D—it improves substantially with only a few labeled samples. This avoids the strong assumptions required by existing TTA-FL approaches. Overall, FEROMA remains stable and accurate across all heterogeneity levels, adapting more effectively than TTA-FL methods in the Appendix D scenario.

---

> ### Author Response · Authors · 2025-11-19
>
> ## Q9: Can the PCA map $g$ be computed on the server side?
>
> We thank the reviewer for this clarification question. Conceptually, we fully agree: given a fixed random seed and shared bounds $[m^-, m^+]$, the synthetic points and the resulting PCA projector g are deterministic. Thus, in principle, $g$ could also be fitted once on the server on the same synthetic dataset and then broadcast to all clients. The algorithm only requires that all clients share the same linear map $g$, not that it is computed locally.
>
> In our implementation we opted for the opposite design for pragmatic reasons. Fitting a PCA on only 200 synthetic points in $\mathbb{R}^z$ is computationally negligible compared to a single local training epoch, even for $z$ in the range $128–2048$. By computing the PCA locally, we avoid introducing any additional communication beyond the two global min/max values already required in step S1. In contrast, a server-side implementation would need to broadcast the PCA parameters to every client: for a latent space dimension $z$ and reduced dimension $l=10$, this amounts to $(z \cdot l + z)$ parameters (projection matrix plus mean), i.e., from about 1.4k to 22.5k floats when $z \in [128, 2048]$, on top of the existing broadcast. Moreover, the client-side PCA has a practical advantage in dynamic settings where the encoder or latent statistics evolve during training. If the global model is updated and the latent range $[m^-, m^+]$ changes, the server only needs to broadcast the new bounds (2 floats). Each client can then regenerate the synthetic points with the shared seed and refit the PCA locally.
>
> Importantly, this design choice does not affect privacy. Whether $g$ is computed on the server or locally therefore does not change the information available to an adversary. Our current implementation simply demonstrates that the DPE can be realized without any extra communication beyond the initial min/max exchange, but **we will clarify in the revised manuscript that a server-side PCA (with a one-shot broadcast of g) is an equally valid alternative implementation**.
>
>
> ## Q10: Solid and transparent lines in Figure 1.
>
> We thank the reviewer for noticing the missing description in Figure 1. **The solid lines represent the average accuracy for FedAvg and FEROMA, while the transparent lines correspond to individual clients (10 for each)**. We will add this clarification to the caption of Figure 1 in the revised manuscript.
>
> ## Q11: On notation in Eq. (1), $\mathcal{A}$ and $s^{(k)}_t$.
>
> We thank the reviewer for pointing out the abuse of notation in Eq. (1), where $\theta_t^{(k)}$ was used both as the optimisation variable and as the resulting optimiser. In the revised manuscript, we have clarified this by introducing a generic optimization variable $\theta$ and using $\theta_t^{(k)}$  only for the local model after training at round $t$:
> $\theta_t^{(k)} \approx \arg\max_{\theta}\ \mathcal{L}(\theta \mid x^{(k)}, y^{(k)})$
> This equation now summarises the local empirical risk maximisation performed by client $k$, typically implemented via a finite number of SGD steps.
>
> Regarding notation issues related to R1, please refer to our response to Q1, where we clarified the distinction between fixed datasets, underlying distributions, and stochasticity in the DPE. We also agree that additional notation was overloaded in a way that could cause confusion. To ensure clarity, the revised manuscript includes the following changes:
>
> We renamed the measurable set in the DP definition from $\mathcal{A}$ to $\mathcal{S} \subseteq \mathbb{R}^{d}$, to avoid conflict with the notation $\mathcal{A}_{t-1}$ used for the set of active clients.
> We renamed the latent matrix from $s_t^{(k)}$ to $h_t^{(k)}$, reserving $s_t^{(k)}$ for dataset cardinality.
>
> We sincerely appreciate the reviewer's effort in helping us improve the precision and readability of our notation.
>
> ## Q12: Fixing notation in R2.
>
> We thank the reviewer for pointing out this notation issue. We agree that the misuse of $\subseteq$ is not mathematically precise for vectors. **In the revision, we will state that $d^{(k)\prime}_t$ is a subvector of $d^{(k)}_t$**.

---

> ### Author Response · Authors · 2025-11-19
>
> ## Q13: On the term “Lipschitz-equivalent”.
>
> We thank the reviewer for pointing this out. Our use of “Lipschitz-equivalent” follows the standard notion from metric geometry: two distances are (bi-)Lipschitz equivalent on a set if they bound each other up to positive multiplicative constants. Proposition 3 (Appendix C.4.2) already establishes such a two-sided bound between our profile distance $\Delta$ and $W_2$. However, we agree that the term was not explicitly defined and may be unfamiliar to some readers.
>
> In the revised manuscript, we (i) replace “Lipschitz-equivalent” by “bi-Lipschitz equivalent” and (ii) add an explicit explanation that this means there exist constants $c_-, c_+>0$ such that $c_- \Delta \le W_2 \le c_+ \Delta$ on the admissible set (see Proposition). This makes the intended meaning precise and self-contained.
>
> ## Q14: Fixing weight numbers in Figure 3.
>
> We thank the reviewer for highlighting this potential source of confusion. The “nonzero” weights refer specifically to the weights after applying the similarity threshold, not the raw pre-threshold values shown in Figure 3. **We will make this distinction explicit in the revised manuscript to ensure the terminology is clear and unambiguous**.
>
> ## Q15: Additional details to the main pages.
>
> We thank the reviewer for pointing out that several important details currently appear only in the Appendices, which forces readers to jump back and forth when following the main narrative. This was primarily a consequence of the page limit, but we agree that certain elements should be made accessible directly within the main text for clarity.
>
> In the final version with extra one page, **we will bring several key details from the appendices into the main paper**, including:
> 1. A concise clarification of notation and random variables.
> 2. The definition of the active client and the participation scheme used in our experiments.
> 3. Detailed experiment settings, including communication-round settings and other essential experimental parameters.
> 4. short summaries of the dynamic aggregation examples and higher-round convergence analyses (with figures kept in the appendix but referenced and explained in the main text).
>
> These additions will reduce the need to navigate between sections while preserving the technical discussions in the Appendix. We would also be glad to incorporate any additional details the reviewer finds helpful to further streamline the reading experience and improve clarity.

---

> > ### Comment · Reviewer_5gUD · 2025-11-24
> > **Score Updated**
> >
> > Thank you for your detailed response. Your clarifications have addressed my major concerns and cleared my questions, so I raised my score accordingly.

---

### Official Review · Reviewer_7n8b · 2025-10-21

**Soundness:** 3
**Presentation:** 4
**Contribution:** 3
**Rating:** 6
**Confidence:** 3

**Summary:**

Existing federated learning (FL) methods often fail to address distribution shift across clients and distribution drift over time, or they rely on strong assumptions such as known number of client clusters and data heterogeneity types, which limits their generalizability.
This paper aims to develop a new FL solution that can handle distribution shift and drift without relying on client or cluster identity.
This method builds on client distribution profiles that guide model aggregation and test-time model assignment through adaptive similarity-based weighting.

**Strengths:**

The proposed method introduces a unified approach that can handle both distribution shift and distribution drift. It covers scenarios that previous FL methods treat separately. This generalization is meaningful and provides a comprehensive theoretical foundation for dynamic non-IID conditions. Further, the way that this paper designed Distribution- Profile Extractor (DPE) with explicit differential privacy guarantees, bounded stochasticity, and theoretical validation supports a strong methodological foundation.
The experiments utilize six, various non-IID scenarios, and drift frequencies, and large-scale client scenarios.

**Weaknesses:**

This reviewer has some concerns about weaknesses.
The effectiveness of proposed method depends on the quality of the DPE. However, DPE itself requires a pretrained model. The manuscript even states “the effectiveness of FEROMA similarly depends on the quality of its DPE, which must generate reliable representations of client data distributions”.
 Poor representation quality can undermine profile fidelity and mapping accuracy, and ultimately make the method less reliable in low-resource scenarios.
The paper does not clearly quantify the robustness of proposed method against unseen or out-of-support distributions.
While the manuscript states the following, it does not clearly provide theoretical analysis of convergence guarantees or sensitivity of performance to hyperparameters:   “FEROMA dynamically selects the best aggregation strategy—ranging from clustered to personalized or global—based on client distribution profiles, and naturally extends to test-time adaptation without retraining.”,

**Questions:**

- Can the authors revise the paper to enhance the FEROMA’s dynamic aggregation using more solid theoretical analysis?
- Would it be possible to make the FEROMA’s distribution in a way to be self-adaptive to evolving latent spaces?

---

> ### Author Response · Authors · 2025-11-19
>
> ## Q1: Impact of DPE quality.
>
> We appreciate the reviewer for highlighting this important point regarding the dependence of FEROMA on the quality of its distribution profile extractor (DPE). We fully agree that any representation-based federated method must rely on the quality of the extracted features, and FEROMA is no exception. However, we would like to offer several clarifications and mitigating factors.
>
> 1. **Reliance on representation quality is inherent, not unique to FEROMA.**
>   Many strong personalized or clustered FL methods implicitly or explicitly rely on good representations. FEROMA makes this dependence explicit through profiles but does not introduce a new reliance beyond what modern FL methods already require.
>
> 2. **FEROMA is designed to remain stable even with imperfect or noisy profiles.**
>     Our DPE produces very low-dimensional, distribution-level descriptors (typically 220–2020 dimensions depending on the number of labels). These profiles capture key coarse statistical characteristics rather than fine-grained features. Our theoretical and empirical results show that FEROMA only requires approximate preservation of relative similarities:
>
>     - Profile noise affects aggregation only through relative distances.
>     - The threshold mechanism filters out noisy or uninformative comparisons.
>     - Stable strategy selection emerges even with bounded approximation error ($\xi < 1.1$).
>
>     This is also confirmed in our sensitivity study (Appendix E, Fig. 8): moderate reductions in profile quality (e.g., DP noise with $\varepsilon \ge 5$) do not degrade FEROMA’s behavior, and even stricter noise budgets lead only to mild impact.
>
> 3. **Flexibility of the DPE design beyond pretrained models.**
> While our implementation uses a small global warm-up model to instantiate the DPE, FEROMA itself does not rely on a pretrained encoder. The DPE can be instantiated with:
> • A lightweight and robust public encoder (e.g., tiny CNN for CIFAR, MobileNet-Tiny for CheXpert). We also reasonably expect that, when available, a well-trained public encoder would further improve profile quality,
> • Simple statistics computed directly on the data (feature moments, sketching, random projections),
> • Any mapping that satisfies Definition 3.1.
>
> Moreover, as shown in our experiments, FEROMA does not require the DPE from the first round: the first few rounds may use standard FedAvg, and profiles can be activated once a reasonable encoder is available. This makes FEROMA feasible even in low-resource or when no public pretrained encoder is available initially. We emphasize that FEROMA does not require a task-optimal representation backbone; it only needs consistent embeddings or statistics that roughly preserve distribution differences.
>
> In the final version, **we will explicitly clarify the dependency between FEROMA and the DPE, summarize our robustness findings directly in the main text, and refine the description of the DPE to avoid any misunderstanding** (e.g., emphasizing that pretrained models are not required). These changes will make the role of the DPE clearer and prevent confusion about FEROMA’s practical requirements.
>
>
> ## Q2: FEROMA robustness against unseen or out-of-support distributions.
>
> We thank the reviewer for this important observation. While FEROMA is not specifically designed for addressing completely unseen or out-of-support distributions, **our experimental setup inherently evaluates robustness to both scenarios (Appendix B, Figures 5-6)**. In our drifting dataset generation, we have included cases where testing distributions have not been seen in the training rounds. This is highly likely to happen when the non-IID level is high but drifting time is low (e.g., Non-IID level ‘High’ and Drifting ‘5/20’ in Table 2). This applies to all the tables and results we present in this overall paper.
>
> To more rigorously validate FEROMA’s robustness against truly unseen distributions, we conducted additional experiments using the MNIST dataset with 20 clients under medium non-IID conditions. *We varied the percentage of clients exposed to unseen distributions during testing* and measured the resulting impact on final accuracy (averaged on drifting frequencies 5,10 and 20, out of 20 total rounds).
>
> |Unseen rate|0%|20%|40%|60%|80%|100%|
> | - | - | - | - | - | - | - |
> |FedAvg|73.25±4.45|69.12±2.84|65.20±0.76|62.60±0.31|59.59±0.78|53.05±4.17|
> |FEROMA|86.70±1.56|86.12±1.28|85.79±1.39|85.10±0.09|84.98±0.05|84.80±0.12|
>
> As reported in the table above, when 0% of clients encounter unseen distributions, our method achieves baseline performance. **In extreme scenarios where 100% clients have unseen distributions during testing, FEROMA still maintains competitive accuracy**. These results further quantify FEROMA’s robustness against unseen distributions and validate its generalization capabilities under challenging distribution drifting conditions.

---

> ### Author Response · Authors · 2025-11-19
>
> ## Q3: Theoretical analysis of FEROMA on convergence.
>
> We really appreciate this comment on a more theoretical analysis of FEROMA convergence. However, analyzing the unbounded and adversarial concept drift precludes classical convergence guarantees. Instead, we can analyze the behavior of FEROMA on a time interval between two distribution-shift events, where the local data distributions are approximately stationary. For this analysis, we adopt the common idealized setting of a fixed set of clients with full participation in every round, although FEROMA supports partial participation in practice.
>
> Fix an interval of communication rounds $\mathcal{T} = \{t_0,\dots,t_0+T-1\}$ during which the local distributions are stationary:
> $$P(X_t^{(k)}, Y_t^{(k)}) = P^{(k)} \quad\forall t \in \mathcal{T}\, k \in \{1,\ldots,K\}.$$
> For each client $k$, let $$f_k(\theta)\:\= \mathbb{E}{(x,y)\sim P^{(k)}}[\ell(\theta;x,y)]$$
> denote its local objective, and $F(\theta) := \sum_{k=1}^K p_k f_k(\theta)$ the global objective with aggregation weights $p_k>0$, $\sum_k p_k = 1$. Within the interval $\mathcal{T}$, the local training step (Eq.(1)) is standard stochastic gradient descent (SGD) on $f_k$, with a fixed number $\tau$ of local steps per communication round. We adopt the following standard conditions from decentralized/local-SGD analysis [1-3].
>
> *Assumption 1: *Smoothness and bounded variance. For each client $k$, $f_k$ is $L$-smooth: $\|\nabla f_k(\theta)-\nabla f_k(\theta')\| \le L\|\theta-\theta'\| \quad \forall\theta,\theta',$
> and is bounded below by $f_k^\star$. Stochastic gradients are unbiased and have bounded variance: $\mathbb{E}[g_t^{(k)} \mid \theta_t^{(k)}] = \nabla f_k(\theta_t^{(k)})$
> and $\mathbb{E}\|g_t^{(k)} - \nabla f_k(\theta_t^{(k)})\|^2 \le \sigma^2$
> for all $t,k$.
>
> *Assumption 2: *Profile-induced mixing matrices. At each round $t\in\mathcal{T}$, FEROMA's distribution-profile mapping (Eq.(3) to (5)) induces a mixing matrix $W_t \in \mathbb{R}^{K\times K}$ with entries $(W_t)_{kj} := \bar w_t^{(k,j)}$, where $\bar w_t^{(k,j)}$ denotes the (possibly thresholded and renormalized) association weight used in  Eq.(5). We assume:
> - Symmetry and double stochasticity: $W_t = W_t^\top,\quad W_t \mathbf{1} = \mathbf{1},\quad \mathbf{1}^\top W_t = \mathbf{1}^\top,$
> - Expected consensus rate: there exist constants $p \in (0,1]$ and integer $\tau \ge 1$ such that, for all matrices $X \in \mathbb{R}^{d \times K}$ and all integers $\ell$,
>         $$\mathbb{E}\lVert X W_{\ell,\tau} - \bar{X} \rVert_F^{2} \le (1-p)\lVert X - \bar{X} \rVert_F^{2}$$
>         Where $W_{\ell,\tau} := W_{(\ell+1)\tau-1} \cdots W_{\ell\tau}$, $\bar X := X \tfrac{1}{K}\mathbf{1}\mathbf{1}^\top$, and the expectation is taken over the randomness of the mixing matrices in the block $\{W_{\ell\tau},\dots,W_{(\ell+1)\tau-1}\}$.
>
> Assumption 2 is standard in decentralised SGD with time-varying topologies and ensures that repeated applications of $\{W_t\}$ drive the local models towards the uniform average $\bar X$, with contraction factor $(1-p)$ [1,4].
> This assumption is satisfied by FEROMA when, for the convergence analysis, the server constructs a symmetric similarity kernel from the profiles and applies a doubly-stochastic normalization to obtain mixing matrices $W_t$, which empirically behaves similarly while preserving FEROMA’s clustered/personalised/global behaviours.
>
> Given (R1) and (R3), profile distances track the underlying distribution distance up to small distortion, and the injected noise has bounded covariance. Under a stationary window, this makes it natural to assume that the resulting similarity-based matching can be chosen so that the induced mixing matrices satisfy the standard expected consensus condition (Assumption 4 in [1]); formally characterizing how DPE parameters affect the spectral gap is an interesting direction for future work.
>
> *FEROMA as decentralised local-SGD.* Let $\Theta_t := [\theta_t^{(1)},\dots,\theta_t^{(K)}] \in \mathbb{R}^{d\times K}$ denote the matrix of client parameters at the beginning of round $t$. One round of FEROMA on the stationary interval $\mathcal{T}$ can be written as
>
> $$\Theta_{t+1} = (\Theta_t W_t) - \eta_t G_t,$$
> where $(\Theta_t W_t)$ is the profile-based aggregation (Eq. 5)
> and $\eta_t G_t$ is the local SGD update.
>
> where the $k$-th column of $\Theta_t W_t$ is exactly $\sum_{j} \bar w_t^{(k,j)} \theta_t^{(j)}$, as in Eq.(5), $\eta_t$ is the stepsize at round $t$, and $G_t$ stacks the (mini-batch) stochastic gradients performed by the clients during their local updates starting from $\Theta_t W_t$. This recursion coincides with the generic decentralized local-SGD iteration with time-varying mixing matrices studied in [1], specialized to the star-shaped topology in which the averaging is computed centrally but applied client-side.
>
> (Continued on next comment)

---

> ### Author Response · Authors · 2025-11-19
>
> Under previous assumptions 1-2, and restricted to a stationary interval $\mathcal{T}$, the iterates of FEROMA satisfy the matrix recursion above with mixing matrices $\{W_t\}$ that obey Assumption 2. Consequently, FEROMA is a special case of the decentralized local-SGD framework with changing topology and local updates analyzed by [1]. We will introduce an appendix section for the convergence analysis in the final manuscript.
>
> [1] Koloskova, Anastasia, et al. "A unified theory of decentralized SGD with changing topology and local updates." International conference on machine learning. PMLR, 2020.
>
> [2] Stich, Sebastian U. "Local SGD converges fast and communicates little." arXiv preprint arXiv:1805.09767 (2018).
>
> [3] Gao, Hongchang, An Xu, and Heng Huang. "On the convergence of communication-efficient local SGD for federated learning." Proceedings of the AAAI Conference on Artificial Intelligence. Vol. 35. No. 9. 2021.
>
> [4] Pu, Shi, Alex Olshevsky, and Ioannis Ch Paschalidis. "Asymptotic network independence in distributed stochastic optimization for machine learning: Examining distributed and centralized stochastic gradient descent." IEEE signal processing magazine 37.3 (2020): 114-122.
>
>
> ## Q4: Self-adaptive DPE design with evolving latent spaces.
>
> We thank the reviewer for this insightful question. Indeed, allowing FEROMA’s DPE to evolve alongside the global model could further improve adaptability to severe client drift.
>
> In our current implementation, the DPE function is fixed after the initial warm-up rounds, using the early global model as its backbone. However, **the framework does not fundamentally require a static extractor**. FEROMA only requires that all profiles within a round are computed using the same DPE. This constraint means that a self-adaptive DPE is straightforward to support with no additional architectural changes needed: whenever the global model is updated and deemed a better representation backbone, we can simply compute two descriptors within the transition round—one using the current DPE (for the present aggregation) and one using the updated DPE (stored and used from the next round onward). The first descriptor is used for the current-round weight calculation, and the second descriptor is provided to the next round.
>
> **This extension would allow FEROMA to periodically refresh its representation space, improving similarity estimation under large distribution shifts or long-horizon drift**. We appreciate the reviewer for raising this point, and we will include a short discussion in the “Future Work” section to highlight the potential of an adaptive DPE mechanism.

---

> ### Comment · Reviewer_7n8b · 2025-11-19
> **Rating updated**
>
> I'd like to thank authors for thoroughly responding to my comments. I increased my rating based on the convincing responses provided by authors.

---

### Official Review · Reviewer_pNbg · 2025-10-29

**Soundness:** 3
**Presentation:** 3
**Contribution:** 2
**Rating:** 6
**Confidence:** 3

**Summary:**

This paper introduces FEROMA, a new Federated Learning (FL) framework designed to handle both distribution shift (heterogeneity across clients) and distribution drift (temporal heterogeneity within clients). Instead of relying on client or cluster identities, FEROMA constructs distribution profiles, including compact and differentially private summaries of local data, to guide model aggregation and test-time model assignment. FEROMA adaptively transitions between clustered, personalized, and global aggregation based on profile similarity, allowing it to generalize to unseen clients without retraining. Extensive experiments across six benchmarks (MNIST, FMNIST, CIFAR-10/100, CheXpert, and Office-Home) show up to 12% accuracy gain over 10 baselines with comparable overhead to FedAvg.

**Strengths:**

+ This paper contains extensive experimental evaluation. The paper includes thorough comparisons with 10 strong baselines across six datasets (including real-world ones like CheXpert and Office-Home). Performance gains are consistent (e.g., +14.1pp on MNIST, +10.3pp on CIFAR-10, Table 1). Scalability experiments up to 100 clients and drift-frequency ablations are convincing.

+ .The paper is well-structured and visually clear, with informative figures (e.g., Figures 1–3) and theoretical justification of the extractor’s properties (R1–R5).

+ The efficiency and practicality of FEROMA are good. FEROMA achieves low computation and communication overhead (profile size ≤ 0.35 % of model parameters). From the idea perspective, the adaptive similarity-based weighting and automatic aggregation mode selection are conceptually neat and practically beneficial.

**Weaknesses:**

- Writing quality is high, though a few sections are dense and could benefit from intuitive explanations or examples (e.g., DPE stochasticity (R3)). The paper includes an extensive set of experimental results across multiple datasets, baselines, and drift/shift scenarios, which demonstrates strong empirical effort and thorough evaluation. However, the presentation could be improved for readability. Currently, the dense amount of numerical results and tables makes it difficult for readers to extract the main insights quickly. I encourage the authors to streamline the narrative by emphasizing key findings in the main text (e.g., highlighting representative cases or summarizing trends) and moving secondary details to the appendix. Adding clearer transition sentences and summary paragraphs after major result sections would also enhance flow and help readers better connect the experiments to the paper’s main claims.

**Questions:**

- The paper claims computational and communication costs comparable to FedAvg. Could the authors quantify this in terms of total transmitted bytes or runtime for larger models (e.g., ResNet or Transformer architectures)?
- The paper includes many experimental tables and dense numerical results. Could the authors consider summarizing key findings in more digestible form? A short “Key Observations” paragraph at Section 4 might help readers connect results back to FEROMA’s core design principles.
- Does FEROMA automatically adapt this threshold during training, or is it fixed?

---

> ### Author Response · Authors · 2025-11-19
>
> ## Q1: Additional intuitive explanations and examples.
>
> We appreciate the reviewer’s observation and agree that including additional intuitive explanations would improve readability. In the revised version, we will clarify the following points:
>
> • **Descriptor Construction (DPE) (~Line 201):**
>   Intuitively, the DPE compresses each client’s dataset into a compact distribution signature: instead of transmitting raw samples or gradients, FEROMA sends a summary reflecting how the data is distributed, not what the data contains. This abstraction removes fine-grained, potentially identifiable information while preserving the structure needed for reliable similarity estimation.
>
> • **R3 — Controlled Stochasticity (~Line 234):**
>   The goal of R3 is to ensure that profiles capture distribution-level behavior rather than act as deterministic fingerprints. Without stochasticity, a client participating in two consecutive rounds with unchanged data would produce identical profiles. This would trigger unnecessary personalized FL behavior simply because two profiles are “exact matches”, reducing beneficial collaboration. With controlled noise, profiles remain close in expectation but not identical, preventing fingerprinting while preserving meaningful distances.
>
> • **Similarity Measure & Clustering (~Line 281):**
>   We will add a short intuition: this metric ensures that clients with similar underlying behaviors—e.g., comparable sensor patterns or similar patient cohorts—naturally cluster together in the profile space and share updates accordingly.
>
> • **Thresholding (~Line 282):**
>   We will clarify that thresholding regulates when collaboration is beneficial. If two profiles are not sufficiently similar, sharing updates may introduce noise rather than help convergence. The threshold therefore prevents both over-collaboration and unnecessary fragmentation, ensuring that groups form only when similarity exceeds a meaningful level.
>
> We thank the reviewer for the suggestion and would be happy to refine additional sections that may benefit from further intuition or examples.
>
>
> ## Q2: Reformatting for readability with clearer transition and summaries.
>
> We thank the reviewer for this valuable suggestion. We fully agree that emphasizing key findings in the main text while relocating secondary details to the appendix will improve readability. In the revised manuscript, **we will streamline Section 4 by** (i) highlighting representative cases that best illustrate FEROMA’s behavior (see the examples in Q1),(ii) summarizing overarching trends rather than enumerating all numerical results, and (iii) moving less central tables and fine-grained breakdowns to the appendix, with clearer transitions connecting the main points. In addition, **we will add a concise “Key Observations” summary and two supporting figures at the end of Section 4 (see our response to Q4)**. These changes will preserve the completeness of the experimental analysis while making the main narrative more focused and accessible.
>
>
> ## Q3: Quantification of costs in FEROMA.
>
> We thank the reviewer for this question and agree that explicit numbers help to substantiate our claim.
>
> On the communication side, FEROMA sends exactly the same model parameters as FedAvg, plus a single low-dimensional profile vector per client (whose size depends only on the number of labels). This overhead is therefore independent of the model size and becomes negligible as the parameter count grows. Concretely, we obtain:
> |Dataset / Model|Method|Upload/Download per client|Dist. Time (20MB/s)|Overhead vs FedAvg|
> |-|-|-|-|-|
> |**MNIST – LeNet5 (62K)**|FedAvg|248.0 KB|0.496 s|–|
> ||FEROMA|248.88 KB|0.498 s|**+0.35%**|
> |**CIFAR-100 – ResNet-9 (1.65M)**|FedAvg|6.6 MB|13.2 s|–|
> ||FEROMA|6.61 MB|13.216 s|**+0.15%**|
> |**CIFAR-100 – ViT-B/16 (~86M)** |FedAvg|344.0 MB|688.0 s|–|
> ||FEROMA|344.01 MB|688.016 s|**<0.01%**|
>
> On the computational side, the only non-trivial extra operation in FEROMA is computing a rank-l PCA on $s^{\text{PCA}}=200$ synthetic latent vectors of dimension $z$ (Step S2), which costs at most $\sim 8.2 \times 10^7$ FLOPs for typical latent sizes ($z \in [128, 2048]$). For comparison, even a single forward pass of our smallest MNIST model already exceeds $6.5 \times 10^8 FLOPs$, while training our largest model on CIFAR-100 requires over $6 \times 10^{11}$ FLOPs per epoch. Moment computation and noise addition are linear in the descriptor size and therefore negligible. Overall, FEROMA adds well below 1% extra computation relative to standard local training.
>
> **We will add a small table in the appendix reporting these per-round communication and time numbers and we will extend the complexity discussion to explicitly compare the PCA cost to the FLOPs of a single training epoch.**

---

> ### Author Response · Authors · 2025-11-19
>
> ## Q4: Key findings summary at Section 4.
>
> We thank the reviewer for this helpful suggestion. We agree that, given the breadth of experimental settings, providing a concise summary of the key takeaways can improve readability and help connect the empirical results to FEROMA’s core design principles. In the revised manuscript, **we will add a short “Key Observations” paragraph at the end of Section 4 that summarizes the main empirical findings**:
>
> **Key Observations.** FEROMA shows consistent robustness across datasets, non-IID severities, and drifting frequencies. Its dynamic strategy selection improves performance for clustering or personalized FL, when profiles remain close, and for global FL fallback, when client distributions diverge sharply from all. In highly heterogeneous or unstable settings, FEROMA retains strong performance where many fixed-strategy baselines degrade, reflecting the benefit of adapting aggregation behavior at each round.
> Furthermore, we acknowledge that FEROMA’s core design principles rely on its dynamic aggregation strategy, which is not fully conveyed by the summary tables alone. To provide a clearer illustration, we will add these figures organically to Section 4 of the main paper in the final manuscript:
>
> **Appendix H: Illustrations of the Dynamic Aggregation Strategy**
> This appendix presents 20 heatmaps, each visualizing the evolution of a single client’s profile distances throughout the dynamic FEROMA aggregation process. These heatmaps highlight the diversity of aggregation behaviors across clients and training rounds, providing a quantitative view of how FEROMA adapts its strategy over time. Note that dynamic aggregation begins in the sixth round; in that round, all distance values are set to 0.05 (the uniform average for 20 clients) because no previous-round profiles are available for comparison.
>
> **Appendix I: Evaluation with Higher Numbers of Communication Rounds**
> This appendix presents two plots. We conduct additional experiments on CIFAR-100 with 20 clients under the medium non-IID level of type P(X), using 100 and 200 communication rounds (with drift introduced every 10 rounds). For both settings, we report the per-client test accuracy over time. The results show that FedAvg struggles to converge even with increased rounds: at each drift event, client performance fluctuates sharply and remains unstable overall. In contrast, FEROMA exhibits steady performance gains throughout training, with only minor fluctuations around drift points. This provides a complementary, per-client view that supports FEROMA’s stable behavior under challenging dynamic non-IID conditions.
>
> **These figures have already been included in the updated appendix of our latest submitted PDF. We invite the reviewer to refer to them for a preliminary view**. We thank the reviewer again for this constructive suggestion, which has helped us improve the clarity and presentation of our contributions.
>
>
> ## Q5: Automatic threshold in FEROMA.
>
> The threshold in Equation 4 is **not fixed** but **automatically adapted during training**. As described in our ablation study (Section F.6), we set the threshold $\tau$ to the *average similarity weight*, which naturally scales with the number of participating clients (e.g., approximately 0.05 when there are 20 clients). Despite its simplicity, this adaptive rule is sufficient to filter out irrelevant or weakly related models in both the Clustering FL and Personalized FL modes. In future work, we plan to explore more principled or data-driven strategies for automatic threshold selection to further improve aggregation reliability.

---

> ### Comment · Reviewer_pNbg · 2025-11-25
> **Score updated.**
>
> Thanks for your response. Most clarifications make sense to me.

---

### Official Review · Reviewer_aPnB · 2025-11-01

**Soundness:** 3
**Presentation:** 3
**Contribution:** 3
**Rating:** 6
**Confidence:** 4

**Summary:**

This paper proposes FEROMA, a FL framework designed to tackle both distribution shifts across clients and distribution drifts within clients over time.  FEROMA introduces privacy-preserving distribution profiles, a lightweight representations that capture each client's data distribution. Through these profiles, FEROMA adaptively maps client data to previous distributions, guiding both aggregation during training and dynamic, test-time model selection for unseen or unlabeled clients. Experiments across various benchmarks and real-world datasets show improvements.

**Strengths:**

1. FEROMA introduces a differentially private distribution profile for each client.
2. The method address diverse FL scenarios in a single framework.

**Weaknesses:**

1. This paper lacks some necessary to compare and discuss in the related work and experiments, e.g., [R1-R3]. Recent alternative approaches for handling distribution shifts and drifts in FL are omitted [R4-R5].
2. How would the DPE behave under significant violations of its underlying assumptions—e.g., highly multi-modal, long-tailed, or high-dimensional client feature spaces? Is there any empirical or theoretical work supporting extension of the current implementation to natural language or time-series data?
3. How robust is FEROMA to adversarial profile manipulation or dishonest client participation? If malicious clients attempt to game profile similarity, what is the effect on model assignment and privacy?
4. The DP mechanism claims negligible performance reduction at $\varepsilon=10$, but Table 9 shows nontrivial drops for stricter budgets. The risk model and privacy guarantees need a more critical security-oriented discussion.

References:

[R1] Tan Y, Long G, Liu L, et al. Fedproto: Federated prototype learning across heterogeneous clients[C]//Proceedings of the AAAI conference on artificial intelligence. 2022, 36(8): 8432-8440.

[R2]Yang Z, Zhang Y, Li C, et al. FedGPS: Statistical Rectification Against Data Heterogeneity in Federated Learning[J]. arXiv preprint arXiv:2510.20250, 2025.

[R3]Chen G, Niu S, Chen D, et al. Cross-device collaborative test-time adaptation[J]. Advances in Neural Information Processing Systems, 2024, 37: 122917-122951.

[R4] Liao X, Liu W, Zhou P, et al. Foogd: Federated collaboration for both out-of-distribution generalization and detection[J]. Advances in Neural Information Processing Systems, 2024, 37: 132908-132945.

[R5] Bhope R A, Jayaram K R, Venkateswaran P, et al. Shift Happens: Mixture of Experts based Continual Adaptation in Federated Learning[J]. arXiv preprint arXiv:2506.18789, 2025.

**Questions:**

See above.

---

> ### Author Response · Authors · 2025-11-19
>
> ## Q1: Additional recent works as baselines.
>
> We thank the reviewer for bringing these relevant works to our attention and providing us with this opportunity to compare FEROMA with these recent advances. **We acknowledge the importance of complete and up-to-date baseline comparisons, and we will incorporate them to improve both the completeness and the positioning of FEROMA in the revised manuscript.**
> We summarize each work below, and will put the additional lines (Please see below) to our main Table 3 for a clear comparison. A more detailed analysis of each baseline will also be added to Appendix A.
>
> - FedProto(R1) uses prototype-based learning to handle client heterogeneity by sharing class prototypes instead of model parameters, enabling adaptation to distribution shifts while maintaining privacy and low costs.
>
> - FedGPS(R2) combines goal-oriented objectives with dynamic path-oriented gradient corrections, sharing statistical information across clients to maintain global perspective while mitigating client’s distribution shift.
>
> - CoLA(R3) performs collaborative test-time adaptation by sharing compact domain-knowledge vectors across devices and updating each client using lightweight similarity-based or reprogramming-based adjustments to handle domain shifts during inference.
>
> - FOOGD(R4) addresses real-world FL by simultaneously tackling both covariate-shift OOD generalization and semantic-shift OOD detection, using client density estimation and global feature regularization to enhance robustness across heterogeneous clients.
>
> - ShiftEx(R5) introduces a MoE framework for streaming FL where client distributions evolve over time. Expert models are dynamically created or reused, clients are assigned to experts based on shift detection, enabling continual adaptation to non-stationary data.
>
> |Algorithm|Cat.|D. shift|D. drift|T. adapt.|Low comm.|Low comp. (Server)|Low comp. (Client)|Scalability|
> |-|-|-|-|-|-|-|-|-|
> |Fedproto(R1)|PFL|$$\checkmark$$|||$$\checkmark$$|$$\checkmark$$|$$\checkmark$$|$$\checkmark$$|
> |FedGPS(R2)|PFL|$$\checkmark$$|||$$\checkmark$$|$\bigcirc$|$$\checkmark$$|$\bigcirc$|
> |CoLA(R3)|TTA-FL|N/A|N/A|$$\checkmark$$|N/A|N/A|N/A|N/A|
> |FOOGD(R4)|PFL|$$\checkmark$$|||$$\checkmark$$|$\bigcirc$|$$\checkmark$$|$\bigcirc$|
> |ShiftEx(R5)|PFL|$$\checkmark$$|$$\checkmark$$||$\bigcirc$|$\bigcirc$|$$\checkmark$$|$\bigcirc$|
>
> **Although CoLA (R3) is not originally proposed for FL, we include it as a strong baseline as its design aligns with FEROMA’s ability to handle test-time distribution changes**. For this reason, several its FL-specific properties are marked as N/A. The preliminary results below.
>
> ### P(X)
> |Method|low5|low10|low20|mid5|mid10|mid20|high5|high10|high20|
> |-|-|-|-|-|-|-|-|-|-|
> |CoLA|89.0±0.2|88.3±0.3|88.5±0.5|82.6±4.1|82.2±1.9|84.9±0.3|80.4±4.1|84.2±0.3|84.6±0.6|
> |FedAvg|66.5±1.2|67.9±1.5|72.1±3.5|73.2±4.0|73.7±2.3|73.2±1.8|54.8±1.8|54.3±2.7|58.5±5.9|
> |FEROMA|88.9±0.3|88.7±0.3|89.5±0.4|87.1±2.3|86.5±2.5|86.1±3.4|86.4±0.2|85.8±0.4|85.7±1.2|
>
> ### P(Y)
> |Method|low5|low10|low20|mid5|mid10|mid20|high5|high10|high20|
> |-|-|-|-|-|-|-|-|-|-|
> |CoLA|94.7±3.9|98.7±0.5|99.2±1.5|94.2±4.7|96.3±2.6|99.0±0.0|94.0±0.3|90.6±7.6|96.7±2.4|
> |FedAvg|75.1±12.4|83.8±8.8|94.6±1.6|74.5±5.8|82.8±9.4|90.8±1.4|70.4±11.9|74.7±11.6|89.4±5.8|
> |FEROMA|96.5±3.2|98.7±0.4|99.4±0.3|97.1±1.8|97.9±2.0|98.7±1.1|96.9±1.9|97.2±3.0|99.3±0.3|
>
> ### P(Y|X)
> |Method|low5|low10|low20|mid5|mid10|mid20|high5|high10|high20|
> |-|-|-|-|-|-|-|-|-|-|
> |CoLA|84.5±2.6|85.4±1.5|82.9±0.4|76.5±3.8|80.2±1.3|77.1±4.0|78.3±1.8|78.9±0.4|77.8±1.3|
> |FedAvg|73.5±1.6|72.9±0.8|72.9±1.0|64.9±1.7|65.0±2.1|64.2±2.0|57.1±2.1|57.2±2.0|56.4±1.1|
> |FEROMA|89.6±0.5|89.5±0.5|89.8±0.8|88.9±0.5|88.9±0.7|89.0±0.6|88.2±0.7|88.1±0.5|88.4±0.7|
>
> ### P(X|Y)
> |Method|low5|low10|low20|mid5|mid10|mid20|high5|high10|high20|
> |-|-|-|-|-|-|-|-|-|-|
> |CoLA|86.7±0.2|89.0±1.7|85.9±3.3|86.1±1.3|82.6±7.2|84.3±0.4|88.1±0.9|79.2±6.9|88.9±0.6|
> |FedAvg|73.4±9.8|81.6±2.9|77.7±3.3|73.3±4.0|73.1±4.1|75.0±4.1|71.6±5.1|74.3±5.1|69.9±9.8|
> |FEROMA|87.3±4.8|88.8±1.9|89.9±2.1|87.0±4.5|89.0±1.6|90.1±0.8|89.4±1.2|88.3±1.1|89.7±0.6|
>
> The experiment settings are the same as reported in our manuscript. As expected, CoLA provides strong test-time robustness due to its domain-adaptation mechanism, but it does not address training-time distribution drift; consequently, while its performance is competitive, it does not surpass FEROMA, which handles both drift and shift during training.
>
> We kindly ask for the reviewer’s understanding that we were not able to reproduce all five baselines within the short rebuttal window. Several of these methods require non-trivial architectural adjustments and additional computational resources that exceed what is feasible during the rebuttal period. We recognize their importance, and will extend our comparison in the final revised manuscript. We thank the reviewer again for these valuable suggestions, which significantly enhance the completeness and fairness of our evaluation.

---

> > ### Author Response · Authors · 2025-11-19
> >
> > ## Q2: Generalizability of DPE to other feature spaces.
> >
> > We thank the reviewer for raising this important question regarding the behavior of the Distribution Profile Extractor (DPE) under more complex feature spaces and its applicability beyond the modalities evaluated in our paper.
> >
> > **FEROMA itself does not make any assumption about the underlying data type. The DPE is intentionally designed as an abstraction layer that compresses potentially high-dimensional, multi-modal, or long-tailed client data into a compact distributional profile**. In principle, as long as a suitable feature extractor exists for a given modality, FEROMA can operate on top of it. In our experiments, we use few-round trained models simply as an effective and practical instantiation of DPE—one that captures distributional structure while being lightweight and easy to integrate.
> >
> > We fully agree with the reviewer that extending the current implementation to other data types would further demonstrate the generalizability of FEROMA as well as the flexibility of the DPE design. To this end, **we adapted our framework to an additional dataset: UCI-HAR, a widely used time-series dataset for human activity recognition**. As this dataset is relatively small, we set the number of clients to 10, and we did not apply any data augmentation to avoid distorting its real-world characteristics. Heterogeneity is introduced using Dirichlet sampling with parameter $\alpha$. The corresponding results are reported below:
> >
> > | α   | FedAvg| FEROMA| FedRC| FedEM| FeSEM| CFL| IFCA| pFedMe| APFL| FedDrift| ATP|
> > |-|-|-|-|-|-|-|-|-|-|-|-|
> > | 0.4 | 37.3 ± 4.4  | 61.5 ± 3.3  | 39.9 ± 4.8  | 39.9 ± 4.9  | 24.2 ± 6.4  | 38.5 ± 7.8  | 34.0 ± 3.7  | 26.5 ± 9.1  | 35.5 ± 3.5  | 32.0 ± 2.8  | 38.3 ± 4.3  |
> > | 0.5 | 35.1 ± 3.5  | 53.9 ± 2.0  | 39.1 ± 5.2  | 39.1 ± 5.3  | 35.5 ± 11.2 | 39.3 ± 7.1  | 37.3 ± 3.2  | 25.6 ± 7.5  | 42.4 ± 8.5  | 37.9 ± 29.8 | 40.4 ± 5.5  |
> > | 0.6 | 36.9 ± 5.0  | 58.0 ± 5.2  | 40.5 ± 4.2  | 40.5 ± 4.3  | 34.8 ± 7.5  | 40.1 ± 3.6  | 39.0 ± 5.3  | 30.7 ± 8.6  | 46.2 ± 9.6  | 38.9 ± 7.7  | 41.3 ± 8.3 |
> > | 0.7 | 37.7 ± 1.6  | 50.9 ± 7.9  | 41.1 ± 4.5  | 41.2 ± 4.5  | 38.1 ± 5.0  | 44.6 ± 7.1  | 40.4 ± 5.6  | 30.9 ± 5.3  | 46.6 ± 5.5  | 38.3 ± 10.6 | 43.9 ± 6.5  |
> > | 0.8 | 43.0 ± 3.5  | 53.4 ± 2.5  | 42.9 ± 3.7  | 43.0 ± 3.8  | 40.1 ± 6.7  | 41.5 ± 3.3  | 42.0 ± 3.3  | 36.2 ± 7.9  | 50.0 ± 7.6  | 39.1 ± 8.2  | 45.1 ± 3.3  |
> >
> > The lower $\alpha$ value indicates higher heterogeneity with P(Y) distributions. **The results show that FEROMA achieves strong performance among all baselines, indicating that its distribution-level aggregation remains effective even when the client feature space is multi-modal and time-dependent**. This further supports the generalizability of the DPE abstraction beyond image-based representations.
> >
> > Regarding natural language data, large-scale language models make direct experimentation challenging in the rebuttal window, both computationally and practically. While we were not able to run full FL experiments with transformer-based DPEs due to resource and time constraints, we emphasize that the DPE mechanism is model-agnostic and fully compatible with text embeddings, and we plan to include NLP-based evaluations in future extensions of the paper.
> >
> > We appreciate the reviewer’s suggestion and believe that these additional experiments strengthen the case that FEROMA and its DPE mechanism are broadly applicable across data modalities.

---

> ### Author Response · Authors · 2025-11-19
>
> ## Q3: FEROMA robustness against malicious clients.
>
> We thank the reviewer for raising this important point regarding adversarial manipulation and dishonest participation. We agree that robustness to adversarial attacks is an essential aspect of any FL system. In this work, the design of FEROMA provides several layers of natural protection:
>
> 1. **Limited information exposure.** In FEROMA, clients do not have access to the profiles or similarity weights of other clients, nor do they observe the weights assigned to them in previous rounds. The server never broadcasts the global set of profiles, and the profiles themselves are not shared among clients. As a result, malicious clients lack the information needed to meaningfully infer or target other users’ profile distributions.
>
> 2. **Minimal influence of manipulated profiles.** Even if we assume an unusually strong adversary capable of observing all profiles across rounds, deliberately manipulating a profile has limited impact. Because aggregation weights are derived from profile distances, an adversarial profile that deviates substantially from the population will receive very small weights and will be removed entirely by the thresholding mechanism. Conversely, attempting to craft a profile that mimics specific target distributions will only influence the corresponding CFL subgroup (or just a single client in the PFL branch), leaving the majority of the system unaffected.
>
> 3. **Profiles are lightweight and can be encrypted.** The above attack scenario (full access to all profiles) is highly unlikely in practice. Nevertheless, FEROMA is designed to remain robust even under such strong adversaries. Because its profiles are intentionally lightweight, they can be efficiently protected using standard encryption or secure-channel techniques. This stands in contrast to traditional FL, where encrypting full model updates is often prohibitively expensive due to their size and frequency. By comparison, encrypting FEROMA’s compact profiles introduces negligible overhead, making secure transmission both practical and reliable. This substantially reduces the likelihood of profile compromise and further strengthens the system’s robustness against adversarial manipulation.
>
> We appreciate again the reviewer for highlighting this perspective. The robustness to adversarial profile manipulation is an important property of FEROMA, and we will discuss explicitly and add an appendix section analyzing these scenarios in the revised manuscript.
>
> ## Q4: Deeper discussion of DP and privacy in FEROMA.
>
> We thank the reviewer for pointing out the need for a deeper discussion of the privacy guarantees and their interaction with utility.
>
> First, **FEROMA’s risk surface is contained by design, even before adding DP noise**. Profiles are constructed as low-dimensional, many-to-one summaries of the local distribution (Definition 3.1): each profile $d^{(k)}_t \in \mathbb{R}^d$ is obtained by aggregating statistical moments in latent space, with $d \ll |x^{(k)}_t|$ (e.g., d=220 on MNIST, compared to thousands of input dimensions of the local dataset). As a consequence, infinitely many distinct datasets can map to the same profile, making the mapping non-invertible from an information-theoretic perspective and already limiting the viability of reconstruction or membership inference attacks that rely on fine-grained gradient or representation access.
>
> In addition, **the DPE is intentionally stochastic (Monte Carlo subsampling plus internal noise): even for a fixed local dataset, repeated extractions do not yield identical profiles**. This stochasticity does not by itself constitute a formal privacy guarantee in the DP sense, but it further complicates exact fingerprinting and stable linkage of client distributions across rounds, by reducing the signal-to-noise ratio available to an adversary observing a finite number of profiles.
>
> On top of this structural protection, we add an ($\varepsilon,\delta$)-DP mechanism at the profile level (Requirement R4), which we calibrate in Appendix E. The noise is applied to a vector of already-aggregated statistics. For practitioner-relevant budgets (ε ≥ 5), FEROMA retains essentially the same accuracy as the non-DP variant across all datasets (Table 9), while only very strict budgets (ε = 1–2) – where the added noise is intentionally large – produce noticeable utility degradation. In other words, our claim of “negligible performance reduction” is intended for moderate ε, not for the most stringent DP settings.
>
> We will revise the paper to (i) more clearly articulate this multi-layer protection model—non-invertible, low-dimensional, and stochastic distribution profiles plus a DP mechanism; (ii) explicitly state that moderate DP budgets (ε ≥ 5) give a favorable privacy–utility trade-off, while very strict budgets (ε ≈ 1–2) naturally incur larger accuracy drops; and (iii) expand the security-oriented discussion in Appendix E.

---

> > ### Comment · Reviewer_aPnB · 2025-11-25
> > **Concerns solved**
> >
> > Thanks to the authors' effort. Most of my concerns have been solved, I will keep my score.

---

### Author Response · Authors · 2025-11-30
**Author Final Remarks**

We extend our sincere gratitude to the AC and all reviewers for their invaluable feedback and constructive engagement throughout this review process. We are encouraged by their recognition of FEROMA's key strengths: the novel distribution profile extractor (DPE) mechanism for handling arbitrary distribution shifts without prior knowledge (aPnB, 7n8b), comprehensive evaluation across diverse datasets and heterogeneity levels (aPnB, 5gUD), robust performance under both distribution drift and shift scenarios (pNbg, 7n8b), flexible framework design supporting clustering/personalized/global FL modes (aPnB, pNbg), lightweight communication overhead comparable to FedAvg (pNbg), and well-structured presentation with clear theoretical foundations (5gUD). To this end, we summarize the key improvements made during rebuttal that helped strengthen our core contributions:

- **Strengthened Theoretical Foundation**.
We provided additional mathematical justification including formal convergence analysis demonstrating that FEROMA follows decentralized local-SGD with time-varying mixing matrices under stationary intervals, clarification of the bi-Lipschitz equivalence between descriptor distances and 2-Wasserstein distance with explicit bounds, precise notation distinguishing between underlying distributions and finite datasets, and formalization of the DPE as a stochastic mapping with bounded approximation error guarantees.

- **Expanded Experimental Validation**.
Following reviewer suggestions, we conducted extensive additional experiments including: comparison with 5 recent baselines (FedProto, FedGPS, CoLA, FOOGD, ShiftEx) showing FEROMA's superior performance, validation on new datasets (UCI-HAR time-series), extended evaluation with 100 and 200 communication rounds confirming stable convergence, robustness testing with 100% unseen distributions maintaining 84.8% accuracy vs 53% for FedAvg, comprehensive ablation studies on DPE quality and threshold mechanisms, and detailed per-client convergence analysis with visualization heatmaps.

- **Framework Flexibility Demonstration**.
We demonstrated FEROMA's adaptability through multiple extensions: self-adaptive DPE design allowing evolving latent spaces, explicit privacy analysis showing multi-layer protection (non-invertible profiles + stochasticity + DP), robustness against malicious clients through limited information exposure and minimal influence mechanisms, applicability to multi-modal data (time-series, planned NLP extensions), and support for dynamic scenarios with client joining/leaving and partial participation.

- **Comprehensive Response to All Concerns**.
We addressed all major reviewer points through theoretical analysis, empirical validation, and methodological clarification. Specific improvements include: reformatted presentation with clearer transitions and "Key Observations" summaries (pNbg), quantified communication/computation costs showing <1% overhead (pNbg), clarified automatic threshold adaptation and global FL fallback mechanisms (5gUD), added dataset-specific summary tables for all six benchmarks (5gUD), and resolved all notation inconsistencies with standardized mathematical formulations (5gUD).

The review process helped us significantly improve the clarity, empirical validation, and theoretical robustness of our work, while the core contributions and evaluation remain consistent with the original submission. Multiple reviewers acknowledged these improvements, recognizing that FEROMA addresses fundamental challenges in federated learning through its unified approach to handling arbitrary distribution shifts and drifts. We believe these enhancements highlight FEROMA's significance as a practical solution capable of scaling from controlled experimental settings to realistic large-scale deployments with dynamic, heterogeneous client distributions.

---

### Meta-Review · Area_Chair_7j1L · 2026-01-07

**Summary:**

This paper presents a novel FL framework that explicitly handles both distribution shift and drift without requiring knowledge of client or cluster identities. The manuscript benefited from prompt and constructive feedback from the authors and reviewers. The authors have addressed the most significant concerns raised in the reviews.

**Reviewer Concerns:**

Most reviewers’ comments were addressed with very prompt replies.

**Reviewer Scores:**

Additional comparisons and discussion have been provided, which have led most reviewers to increase their scores.

---

### Decision · Program_Chairs · 2026-01-26

Accept (Poster)